# How much information do extinction and backscattering measurements contain about the chemical composition of atmospheric aerosol?

Michael Kahnert[1,2] and Emma Andersson[2]

[1]Research Department, Swedish Meteorological and Hydrological Institute, Folkborgsvägen 17, SE-601 76 Norrköping, Sweden
[2]Department of Earth and Space Science, Chalmers University of Technology, SE-412 96 Gothenburg, Sweden

*Correspondence to:* Michael Kahnert (michael.kahnert@smhi.se)

**Abstract.**

We theoretically and numerically investigate the problem of assimilating multiwavelength lidar observations of extinction and backscattering coefficients of aerosols into a chemical transport model. More specifically, we consider the inverse problem of determining the chemical composition of aerosols from these observations. The main questions are how much information the observations contain to determine the particles' chemical composition, and how one can optimise a chemical data assimilation system to make maximum use of the available information. We first quantify the information content of the measurements by computing the singular values of the scaled observation operator. From the singular values we can compute the number of signal degrees of freedom, $N_s$, and the reduction in Shannon entropy, $H$. As expected, the information content as expressed by either $N_s$ or $H$ grows as one increases the number of observational parameters and/or wavelengths. However, the information content is strongly sensitive to the observation error. The larger the observation error variance, the lower the growth rate of $N_s$ or $H$ with increasing number of observations. The right singular vectors of the scaled observation operator can be employed to transform the model variables into a new basis in which the components of the state vector can be partitioned into signal-related and noise-related components. We incorporate these results in a chemical data assimilation algorithm by introducing weak constraints that restrict the assimilation algorithm to acting on the signal-related model variables only. This ensures that the information contained in the measurements is fully exploited, but not over-used. Numerical tests show that the constrained data assimilation algorithm provides a solution to the inverse problem that is considerably less noisy than the corresponding unconstrained algorithm. This suggests that the restriction of the algorithm to the signal-related model variables suppresses the assimilation of noise in the observations.

## 1 Introduction

Atmospheric aerosols have a substantial, yet highly uncertain impact on climate, they can cause respiratory health problems, degrade visibility, and even compromise air-traffic safety. The physical and chemical properties of aerosols play a key role in understanding these effects. The aerosol properties are determined by a complex interplay of different chemical, microphysi-

cal, and meteorological processes. These processes are investigated in environmental modelling by use of chemical transport models (CTMs). However, modelling aerosol processes is plagued by substantial biases and errors (McKeen et al., 2007). It is, therefore, fundamentally important to evaluate and constrain CTMs by use of measurements.

Measurements from satellite instruments provide consistent long-term data sets with global coverage. However, it is notoriously difficult to compare measured radiances to modelled aerosol concentrations. An alternative to using radiances is to make use of satellite retrieval products. For instance, one of the products of the CALIPSO lidar instrument (Cloud-Aerosol Lidar and Infrared Pathfinder Satellite Observations) is a rough classification of the aerosol types (i.e. dust, smoke, clean/polluted continental, and clean/polluted marine). This retrieval product is based on lidar depolarisation measurements (Omar et al., 2009). For the evaluation of aerosol transport models this provides us with a qualitative check for the chemical composition of aerosols. However, this is of limited practical use, since what we really need is quantitative information on the particles' chemical composition (which can be size-dependent). The most popular approach in evaluating and constraining aerosol transport models is the use of retrieved optical properties, such as aerosol optical depth, or extinction and backscattering coefficients. Yet another idea is to provide the particles' refractive index as a retrieval product (e.g. Müller et al., 1999; Veselovskii et al., 2002). However, the use of such retrieval products still leaves us with the challenge of solving an ill-posed inverse problem, namely, of determining the particles chemical composition from their retrieved optical or dielectric properties.

A systematic class of statistical methods for solving this inverse problem is known as data assimilation. Recent studies have applied data assimilation to aerosol models with varying degrees of sophistication, ranging from simple dust models (Khade et al., 2013) and mass transport models (Zhang et al., 2014) to microphysical aerosol models based on modal (Rubin and Collins, 2014) or sectional descriptions (Sandu et al., 2005; Saide et al., 2013) of the aerosol size distribution. The assimilation techniques that have been used comprise variational methods, such as 2D (Zhang et al., 2014), 3D (Kahnert, 2008; Liu et al., 2011), and 4D variational methods (Benedetti et al., 2009), as well as ensemble approaches (Sekiyama et al., 2010). Assimilation of satellite products for trace gases is relatively straightforward, since observed and modelled trace gas concentrations are almost directly comparable. However, aerosol optical properties observed from satellites are not directly comparable to the modelled size distribution and chemical composition of the aerosols. Solving this problem amounts to regularising a severely under-constrained inverse problem. Previous aerosol assimilation attempts have been mainly based on educated guesses about the information content of the observations. For instance, there have been studies on the assimilation of aerosol optical depth (AOD) in which all chemical aerosol components in all size classes and at all model layers were used as independent control variables (Liu et al., 2011). This approach largely disregards the problems involved in inverse modelling. By contrast, it has been proposed to only allow for the total aerosol mass concentration to be corrected by data assimilation of AOD (Benedetti et al., 2009; Wang et al., 2014). This is a more prudent approach based on the plausible assumption that a single optical variable only contains enough information to control a single model variable. There have also been intermediate approaches in which the total aerosol mass per size bin have been used as control variables (Saide et al., 2013).

In all such approaches the choice of control variables is based on ad hoc assumptions. Numerical assimilation experiments by Kahnert (2009) suggest that observations of several aerosol optical properties at multiple wavelengths may allow us to constrain more than just the total mass concentration, but certainly not *all* aerosol parameters. However, it is still an unsolved mystery how

much information a given set of observations actually contains about the size distribution and chemical composition of aerosols, and exactly which model variables are related to the observed signals, and which ones are related to noise. Thus a prerequisite for assimilating remote sensing observations into aerosol transport models is to thoroughly understand the information content of the observations as well as the relation between the model variables and the signal degrees of freedom.

In numerical weather prediction (NWP) modelling, several studies have discussed the information content of satellite observations for meteorological variables. For instance, Joiner and da Silva (1998) applied a singular-value decomposition (SVD) approach in order to reduce the effect of prior information in the analysis, so that the retrieval and forecast errors can be assumed to be uncorrelated. Rabier et al. (2002) considered assimilation of IR sounders, which typically provide a large number of different channels. They applied methods of information and retrieval theory in order to decide which channels contain most information about the vertical variation of temperature and humidity. Cardinali et al. (2004) employed the influence matrix to compute diagnostics of the impact of observations in a global NWP data assimilation system. Johnson et al. (2005a, b) investigated filtering and interpolation aspects in a 4DVAR assimilation system by use of an SVD approach. They also used Tikhonov regularisation theory to optimise the signal-to-noise regularisation parameter in order to maximise the information that can be extracted from observations. Xu (2006) compared different metrics, namely, the relative entropy and the Shannon-entropy difference, to measure information contents of radar observations assimilated into a coupled atmosphere-ocean model. Bocquet (2009) used methods of information theory to address the question how to determine an optimum spatial resolution of the discretised space of control variables in geophysical data assimilation.

Burton et al. (2016) have recently investigated the information content of "$3\beta + 2\alpha$"lidar measurements, i.e., observations of backscattering at three wavelengths and extinction at two wavelengths, where the information content was analysed with regard to the refractive index and number distribution of the aerosol particles. Veselovskii et al. (2004, 2005) have performed similar analyses of the information content of multiwavelength Raman lidar measurements with regard to the complex refractive index and the effective radius of the aerosol particles. As mentioned earlier, the refractive index is a very useful retrieval product of remote sensing observations. However, from the point of view of chemical transport modelling, the main quantities of interest are the concentrations of the different chemical species of which the aerosol particles are composed. Although the chemical composition determines the refractive index, the inversion of this relationship is still under-determined, hence an ill-posed problem. In the present paper, we want to investigate the inverse problem that goes all the way from optical properties to the chemical composition of particles.

The two main goals of this paper are (i) to apply a systematic method for analysing the information content of aerosol optical properties with regard to the particles' chemical composition, and (ii) to test an algorithm for making an automatic choice of control variables in chemical data assimilation such that all control variables are signal-related, while the noise-related variables remain unchanged by the assimilation procedure. The main hypothesis is that by constraining the data assimilation algorithm to acting on the signal-related variables only, the output will be less noisy than in an unconstrained assimilation. The focus of our study will be on spectral observations of extinction and backscattering coefficients, which can be retrieved

from lidar observations.[1] We will not restrict this analysis to any fixed choice of wavelengths, such as $3\beta + 2\alpha$. Instead, we will
investigate the information content for varying combinations of the three main wavelengths of the commonly used neodymium-
doped yttrium aluminium garnet (Nd:YAG) laser. However, it should be mentioned that extinction measurements at the lowest
harmonic of 1064 nm can be difficult and plagued by high errors; in practice, this will affect the observation error, resulting in
a low information content of this particular measurement.
The paper is organised as follows. Section 2 gives a rather concise introduction of the modelling tools and of the numerical
approach employed to studying the information content of extinction and backscattering observations. Section 3 presents the
main results of this study, and Sect. 4 offers concluding remarks. To make this paper self-contained, we included an appendix
that gives a brief introduction to some essential concepts of data assimilation, and a detailed explanation of the methods we
used for quantifying the information content of aerosol optical observables.

## 2  Methods

This study consists of two parts. In the first part we quantify the information content of extinction and backscattering coeffi-
cients at multiple wavelengths. In the second part we perform a numerical test to investigate to what extent the concentrations
of different chemical aerosol components can be constrained by observations of extinction and backscattering coefficients.
The modelling tools required for this study are (i) a chemical transport model; (ii) an aerosol optics model; and (iii) a data
assimilation system.

### 2.1  Multiple scale Atmospheric Transport and CHemistry modelling system (MATCH)

We employ the chemical transport model MATCH, which is an off-line Eulerian CTM with flexible model domain. It has been
previously used from regional to hemispheric scales. Here we use a model version that contains a photochemistry module with
64 chemical species, among them four secondary inorganic aerosol (SIA), namely, ammonium sulphate, ammonium nitrate,
other sulphates, and other nitrates. It also contains a module with 16 primary aerosol variables, namely, sea salt, elemental
carbon (EC), organic carbon (OC), and dust particles, each emitted in four different size bins. Thus, the model contains 20
different aerosol variables. The size ranges of the four bins are as follows.
Size bin 1: 10–50 nm
Size bin 2: 50–500 nm
Size bin 3: 500–1250 nm

---

[1]In addition to lidar measurements from ground-based and aircraft-carried instruments (e.g. Burton et al., 2015), there are currently two space-borne lidar instruments in orbit. The CALIOP instrument on-board the CALIPSO satellite has been launched in April 2006; it has three receiver channels, one at 1064 nm, and two channels at 532 nm to measure orthogonally polarised components. The CATS instrument on-board the International Space Station has been operational since January 2015; It measures backscattering at 355 nm, 532 nm, and 1064 nm, were the latter two have two orthogonal polarisation channels. It is also capable of performing high spectral resolution measurements at 532 nm. A third instrument is planned to be launched in 2018 (ATLID on-board EarthCARE).

Size bin 4: 1250–5000 nm.
The model reads in emission data, meteorological data, and land use data and computes transport processes, chemical
transformation, and dry and wet deposition of the various trace gases and aerosols. As output, it provides concentration fields
of gases and aerosols, the deposition of these chemical species to land and water-covered areas, as well as the temporal
evolution of these variables.
We mention that there exists another model version that includes aerosol microphysical processes, such as nucleation, con-
densational growth, and coagulation. In that model version the aerosol size distribution evolves dynamically. The model has
20 size bins and seven chemical species (EC, OC, dust, sea salt, particulate sulphate (PSOX), particulate nitrate (PNOX), and
particulate ammonium (PNHX)), although not all species are encountered in all size bins. The total number of model variables
currently in that version is 82.
More complete information about the mass transport model can be found in Andersson et al. (2007). The sea salt module is
discussed in Foltescu et al. (2005). The aerosol microphysics module is described in Andersson et al. (2015).
For the sake of simplicity we here use the mass transport model without aerosol microphysical processes (see next sec-
tion). The model is set up over Europe covering $33°$ in the longitudinal and $42°$ in the latitudinal direction in a rotated lat-
long grid with $0.4° \times 0.4°$ horizontal resolution. In the vertical direction the model domain extends up to 13 hPa, using 40
terrain-following coordinates. The meteorological input data are taken from the numerical weather prediction model HIRLAM
(Undén et al., 2002). For the emissions of all aerosol components we used EMEP data for the year 2007, where EC and OC
emissions were computed from total primary particle emissions based on the data in Kupiainen and Klimont (2004, 2007).

## 2.2  Aerosol optics model

We have two different optics models coupled to MATCH, one to the mass transport module, and another to the aerosol mi-
crophysics module. The former assumes that all aerosol species are homogeneous spheres, and that each chemical species
is contained in separate particles. Under these assumptions the optics model is linear, i.e., the optical properties are linear
functions of the concentrations of the chemical aerosol species. The latter model accounts for the fact that in reality different
chemical species can be internally mixed, i.e., they can be contained in one and the same particle. That model also accounts
for the inhomogeneous internal structure of black carbon mixed with other aerosol components, and for the irregular fractal
aggregate morphology of bare black carbon particles (Kahnert et al., 2012a, 2013). Under these assumptions the optics model
becomes non-linear, which introduces additional complications in the inverse-modelling problem. This is the main reason why
we chose to use the simpler mass transport optics model in this study. Much of the theory explained in the appendix relies on
the assumption that the optics model is either linear, or that it is only mildly non-linear, so that it can be linearised — see Eq.
(B6).
Table 1 lists the refractive indices in the mass-transport optics model at the three lidar wavelengths considered in this study.
More information about the aerosol optics models implemented in MATCH can be found in Andersson and Kahnert (2016).

**Table 1.** Refractive indices at the three harmonics of the Nd:YAG laser assumed in the MATCH mass-transport optics model.

| wavelength [$\mu m$] | 0.355 | 0.532 | 1.064 |
|---|---|---|---|
| SIA | 1.53+5.0e-3 i | 1.53+5.6e-3 i | 1.52+1.6e-2 i |
| Dust | 1.53+1.7e-2 i | 1.53+6.3e-3 i | 1.53+4.3e-3 i |
| NaCl | 1.51+2.9e-7 i | 1.50+1.0e-8 i | 1.47+2.0e-4 i |
| OC | 1.53+5.0e-3 i | 1.53+5.6e-3 i | 1.52+1.6e-2 i |
| EC | 1.66+7.2e-1 i | 1.73+6.0e-1 i | 1.82+5.9e-1 i |

## 2.3 Three-dimensional variational data assimilation (3DVAR)

Data assimilation is a class of statistical methods for combining model results and observations. The algorithm weighs these two pieces of information according to their respective error variances and covariances. As output the assimilation returns a result in model space of which the error variances are smaller than those of the original model estimate. In our case the model variables are the mass mixing ratios of aerosol components in a three-dimensional discretised model domain. These model variables are summarised in a vector $\boldsymbol{x}$. The model provides us with a background (or first guess) estimate $\boldsymbol{x}_b$ (with an error $\boldsymbol{\epsilon}_b$). The observations, summarised in a vector $\boldsymbol{y}$, are related to the model state $\boldsymbol{x}$ by

$$\boldsymbol{y} = \hat{H}(\boldsymbol{x}) + \boldsymbol{\epsilon}_o, \tag{1}$$

where $\hat{H}$ is known as the observation operator, and $\boldsymbol{\epsilon}_o$ denotes the vector of observation errors. The problem is to determine the most likely state vector $\boldsymbol{x}_a$ given $\boldsymbol{x}_b$ and $\boldsymbol{y}$, and given the background error covariance matrix $\mathbf{B} = \langle \boldsymbol{\epsilon}_b \cdot \boldsymbol{\epsilon}_b^T \rangle$, and the observation error covariance matrix $\mathbf{R} = \langle \boldsymbol{\epsilon}_o \cdot \boldsymbol{\epsilon}_o^T \rangle$. Here $\langle \cdots \rangle$ denotes the expectation value. In the three-dimensional variational method (3DVAR), the maximum-likelihood solution is found by numerically minimising the cost function

$$J = \frac{1}{2}(\boldsymbol{x} - \boldsymbol{x}_b)^T \cdot \mathbf{B}^{-1} \cdot (\boldsymbol{x} - \boldsymbol{x}_b) + \frac{1}{2}[\hat{H}(\boldsymbol{x}) - \boldsymbol{y}]^T \cdot \mathbf{R}^{-1} \cdot [\hat{H}(\boldsymbol{x}) - \boldsymbol{y}]. \tag{2}$$

Data assimilation is commonly employed for constraining model results by use of observations. However, one can also employ data assimilation as an inverse-modelling tool, i.e. for retrieving a model state from measurements. A summary of the theoretical basis of variational data assimilation is given in the appendix.[2]

The MATCH model contains a 3DVAR data assimilation module. This model uses a spectral method, i.e., the model state vector is Fourier-transformed in the two horizontal coordinates. All error correlations in the horizontal direction are assumed to be homogeneous and isotropic. The background error covariance matrix is modelled with a method that follows similar principles to the NMC method (Parrish and Derber, 1992). A more complete description of our 3DVAR program can be found in Kahnert (2008).

---

[2]Many authors distinguish between data *assimilation* and data *analysis*. In data analysis one merely post-processes a model results by incorporating the information provided by observations. In data assimilation, the data analysis process is part of the time-integration of the CTM. Thus, in each time step the result of the analysis becomes the new initial state for the next model forecast. Our 3DVAR code can be used in either analysis or assimilation mode. However, in this study we only perform numerical tests at a fixed point in time. Thus we use the 3DVAR code as a data analysis tool.

## 2.4 Analysis of the information content of aerosol optical parameters

The questions we ask are these.

1. Suppose we have an $n$ dimensional model space. Given $m$ observations (e.g., $m_1$ different parameters at $m_2$ different wavelengths, so that $m_1 \cdot m_2 = m$), how many independent model variables $N \leq n$ can we constrain with the observations? Obviously, the best we can achieve would be $N = \min\{m, n\}$; but often we will have $N < \min\{m, n\}$.

2. Which are the $N$ model variables (or linear combinations of model variables) that can be constrained by the measurements?

Here we only give a summary of the most essential theoretical tools for answering these questions. A more thorough explanation of these concepts is given in the appendix.

First we want to explain what we mean by *signal degrees of freedom* and *noise degrees of freedom*, closely following an example in Rodgers (2000) (p. 29f). Suppose we have a direct measurement $y$ of a scalar variable $x$ with error $\epsilon_o$, i.e.

$$y = x + \epsilon_o. \tag{3}$$

Suppose further that we have a background estimate $x_b$ with background error variance $\sigma_b^2$, and that the error $\epsilon_o$ has variance $\sigma_o^2$. The prior variance of $y$ is given by $\sigma_y^2 = \sigma_b^2 + \sigma_o^2$, assuming that background and observation errors are uncorrelated. One can show that the best estimate $x_a$ of $x$ will be

$$x_a = \frac{\sigma_b^2 y + \sigma_o^2 x_b}{\sigma_b^2 + \sigma_o^2}. \tag{4}$$

Hence, if $\sigma_b^2 \gg \sigma_o^2$, then the measurement $y$ will provide information for estimating $x_a$, i.e., the measurement provides a *degree of freedom for signal*. However, if $\sigma_b^2 \ll \sigma_o^2$, then $x_a$ will be close to $x_b$, and $y$ provides little information to estimating $x_a$. The measurement mostly contains information on $\epsilon_o$, i.e., it provides a *degree of freedom for noise*.

In a more general case we have to consider a state vector $\boldsymbol{x}$ and a set of measurements $\boldsymbol{y}$ with errors $\boldsymbol{\epsilon}_o$. The number $N_s$ of signal degrees of freedom is a measure for the information content of the set of measurements. It provides us with an estimate of the number $N$ of model variables that can be controlled by assimilating measurements.

The mapping from model space to observation space given in Eq. (1) can be Taylor-expanded to first order according to

$$\boldsymbol{y} = \hat{H}(\boldsymbol{x}_b) + \mathbf{H} \cdot \delta\boldsymbol{x} + \boldsymbol{\epsilon}_o, \tag{5}$$

where $\hat{H}$ is the observation operator, $\mathbf{H}$ denotes its Jacobian, and $\delta\boldsymbol{x} = \boldsymbol{x} - \boldsymbol{x}_b$. The background or prior estimate $\boldsymbol{x}_b$ is often obtained from a model run. The (in general non-square) matrix $\mathbf{H}$ is the main quantity we need to investigate in order to address the questions formulated at the beginning of this subsection. It is transformed to the so-called observability matrix $\tilde{\mathbf{H}} = \mathbf{R}^{-1/2} \cdot \mathbf{H} \cdot \mathbf{B}^{1/2}$, where $\mathbf{R}$ is the observation error covariance matrix, and $\mathbf{B}$ denotes the error covariance matrix of the background estimate. Subsequently, one performs a singular-value decomposition (SVD)

$$\mathbf{R}^{-1/2} \cdot \mathbf{H} \cdot \mathbf{B}^{1/2} = \mathbf{V}_L \cdot \mathbf{W} \cdot \mathbf{V}_R^T, \tag{6}$$

where the matrices $\mathbf{V}_L$ and $\mathbf{V}_R$ contain the left and right singular vectors, respectively, and $\mathbf{W}$ is a matrix that contains the singular values along the main diagonal, while all other matrix elements are zero. It turns out that the singular values $w_i$ can be employed to compute the number of signal degrees of freedom $N_s$ according to

$$N_s = \sum_{i=1}^{\min\{n,m\}} w_i^2/(1+w_i^2). \tag{7}$$

Another useful measure is obtained by expressing our incomplete knowledge of the atmospheric aerosol state by use of the Shannon entropy. The use of measurement information reduces the entropy, and this entropy reduction $H$ can be expressed in terms of the singular values:

$$H = \frac{1}{2} \sum_{i=1}^{\min\{n,m\}} \log_2(1+w_i^2). \tag{8}$$

Both $N_s$ or $H$ allow us to quantify the information content of a set of measurements. More detailed explanations of these concepts are given in the appendix. A comprehensive discussion of information aspects and inverse methods for atmospheric sounding can be found in Rodgers (2000).

By performing the transformation

$$\delta \boldsymbol{x}' = \mathbf{V}_R^T \cdot \mathbf{B}^{-1/2} \cdot \delta \boldsymbol{x} \tag{9}$$

we go from our physical model space to an abstract phase space — see Eq. (C16) in appendix C. In this phase space the components of $\delta \boldsymbol{x}'$ can be separated into signal-related and noise-related variables. The signal-related components can be controlled by the measurements, the noise-related components cannot. We therefore introduce constraints into our 3DVAR program such that only the $N_s$ signal-related components of $\delta \boldsymbol{x}'$ are allowed to be adjusted in the data-analysis procedure, while the noise-related components are not altered. This is accomplished by adding an extra term $J_G$ to the cost function in Eq. (2), where

$$J_G = \frac{1}{2} \delta \boldsymbol{x}^T \cdot \mathbf{B}^{-1/2} \cdot \mathbf{V}_R \cdot \mathbf{B}_G^{-1} \cdot \mathbf{V}_R^T \cdot \mathbf{B}^{-1/2} \cdot \delta \boldsymbol{x}, \tag{10}$$

and where $\mathbf{B}_G$ is a diagonal matrix which we assume to have the form

$$\mathbf{B}_G = \sigma_G \operatorname{diag}(w_1, w_2, \ldots, \ldots, w_K, c, \ldots, c). \tag{11}$$

Here $K = \min\{n,m\}$, and the number $c$ is assumed to be much smaller than the smallest singular value. We note that the formulation of the constraint term in Eq. (11) is by no means unique. Other possible choices of the matrix $\mathbf{B}_G$ are discussed in appendix D3. However, we performed preliminary tests which indicate that the constrained 3DVAR approach is not very sensitive to exactly how one chooses to formulate the matrix $\mathbf{B}_G$, as long as it behaves in such a way that the noise-related phase-space variables are tightly constrained, while the signal-related variables can be varied relatively freely by the analysis. The free parameters $\sigma_G$ and $c$ should be tuned in such a way that the constrains are neither too hard nor too soft. In the former case, the analysis will stay too close to the background estimate. In the latter case, it will not differ much from the unconstrained analysis.

## 2.5 Numerical test of the constrained assimilation algorithm

We study the performance of the 3DVAR system by performing a numerical test. To this end, we first perform a reference run by driving the MATCH model with analysed meteorological data. These reference results are taken as the "true" chemical state of the atmosphere. We apply the optics model to the model output to generate synthetic "observations", i.e., a vertical profile at a selected observation point of extinction and backscattering coefficients at three typical lidar wavelengths. Next we run the MATCH model again, this time driven with 48 hour-forecast meteorological data. The results are taken as a proxy for a background model-estimate that is impaired by uncertainties. Finally, we perform a 3DVAR-analysis of the "observations" and the background estimate in an attempt to restore the reference results. In this numerical test we have perfect knowledge of the true state, and we assume that our optics model is nearly perfect, thus providing nearly perfect observations (we assumed that the observation error standard deviation is 10 % of the measurement value).The only factor that may prevent us from fully restoring the reference state is a lack of information in the observed parameters. Thus, comparison of the retrieval and reference results gives us an indication of how strongly different model variables can be controlled by the information contained in the observations.

We perform this test (i) with the unconstrained 3DVAR algorithm; and (ii) with the constrained 3DVAR algorithm. We compare both runs in order to make a first assessment of the impact of the constraints. In particular, we are interested in the prospect of reducing the risk of assimilating noise in such a highly under-constrained inverse problem.

## 3 Results

### 3.1 Analysis of the information content of aerosol optical parameters

We consider the set of parameters $\{k_{\text{ext}}(\lambda_1), k_{\text{ext}}(\lambda_2), \beta_{\text{sca}}(\lambda_1), \beta_{\text{sca}}(\lambda_2), \beta_{\text{sca}}(\lambda_3)\}$, where $k_{\text{ext}}$ and $\beta_{\text{sca}}$ denote the extinction and backscattering coefficients, respectively, and the wavelengths $\lambda_1 = 1064$ nm, $\lambda_2 = 532$ nm, and $\lambda_3 = 355$ nm denote the first three Nd:YAG harmonics. Hereafter, we will abbreviate these parameters by $k_{\text{ext}}(\lambda_i) = k_i$, $\beta_{\text{sca}}(\lambda_j) = \beta_j$, $i = 1, 2$, $j = 1, 2, 3$. Out of this five-parameter set we pick different subsets and analyse the singular values of the corresponding observability matrices. From those we compute the number of signal degrees of freedom as well as the change in Shannon entropy for each subset of measurements. We will focus on those parameter subsets that are technically relevant in practical lidar applications.

Table 2 shows the number of signal degrees of freedom $N_s$ and the reduction in Shannon entropy $H$ for different values of the observation standard deviation $\sigma_o$. For low values of $\sigma_o$, the number of signal degrees of freedom is identical to the number of observational parameters. However, as we increase $\sigma_o$ we observe a decrease in $N_s$. For instance, for $\sigma_o = 100$ % the five parameters $\beta_1 + \beta_2 + \beta_3 + k_2 + k_3$ (last row) only provide roughly $N_s = 3$ signal degrees of freedom. The reduction in Shannon entropy $H$ displays an analogous behaviour. For instance, for $\sigma_o = 1$ % we see that $H$ consistently increases as one increases the number of observational parameters. This is much less pronounced for $\sigma_o = 100$ %. In that case, $H$ does increase as one goes from a single parameter to two parameters (compare the first to the second and fourth rows). However, as one adds more

**Table 2.** Number of signal degrees of freedom $N_s$ and reduction in entropy $H$ as a function of observation standard deviation, taken from the lowest model layer (closest to the surface). Results are shown for different subsets of $k_1$, $k_2$, $\beta_1$, $\beta_2$, $\beta_3$, where $k_i$ and $\beta_i$ represents the extinction and backscattering coefficient, respectively, at the wavelengths $\lambda_1 = 1064$ nm, $\lambda_2 = 532$ nm, and $\lambda_3 = 355$ nm.

| | Obs. Std. dev. [%] | 1 | | 5 | | 10 | | 50 | | 100 | |
|---|---|---|---|---|---|---|---|---|---|---|---|
| No. | Parameters | $N_s$ | $H$ | $N_s$ | $H$ | $N_s$ | $H$ | $N_s$ | $H$ | $N_s$ | $H$ |
| 1. | $\beta_3$ | 1.00 | 10.9 | 1.00 | 8.58 | 1.00 | 7.58 | 1.00 | 5.26 | 1.00 | 4.26 |
| 2. | $\beta_1+\beta_2$ | 2.00 | 20.6 | 2.00 | 15.99 | 2.00 | 13.98 | 1.97 | 9.36 | 1.90 | 7.42 |
| 3. | $\beta_1+\beta_2+\beta_3$ | 3.00 | 27.3 | 3.00 | 20.3 | 2.99 | 17.3 | 2.72 | 10.5 | 2.33 | 8.00 |
| 4. | $\beta_3+k_3$ | 2.00 | 19.4 | 2.00 | 14.8 | 2.00 | 12.8 | 1.92 | 8.21 | 1.74 | 6.37 |
| 5. | $\beta_1+\beta_2+k_2$ | 3.00 | 28.0 | 3.00 | 21.0 | 2.99 | 18.0 | 2.77 | 11.2 | 2.42 | 8.63 |
| 6. | $\beta_1+\beta_2+\beta_3+k_2+k_3$ | 5.00 | 40.0 | 4.97 | 28.4 | 4.91 | 23.5 | 3.89 | 12.9 | 2.97 | 9.49 |

parameters, the increase in $H$ slows down considerably. For five parameters (last row), $H$ is only about twice as high as for a single parameter (first row).

This illustrates the pivotal importance of the observation error for the amount of information that can be obtained from measurements. It is important to understand that the observation error $\epsilon_o$ is not the same as the measurement error $\epsilon_m$. Rather, in our case we have $\epsilon_o = \epsilon_m + \epsilon_f$, where $\epsilon_f$ denotes the forward-model error [see, e.g., Eq. (1) and accompanying text in Rabier et al. (2002)]. Any simplifying assumptions in the optics model or incomplete knowledge of the particle size distribution, morphology, chemical composition, or dielectric properties can contribute to $\epsilon_f$. Such assumptions enter into our relatively simple optics model. [3] Note also that in operational applications there may be other terms contributing to $\epsilon_o$. For instance, if a point measurement is taken at a location that does not provide a good representation of the grid-cell average, then one would have to add a representativity error $\epsilon_r$ to the observation error.

The strong impact of the observation errors on the information content of measurements suggests two conclusions.

1. In order to make the forward-model error $\epsilon_f$ as small as possible, it is essential to develop accurate and realistic aerosol optics models. The most accurate measurements may intrinsically contain a wealth of information on aerosol properties. But we can only make use of this information to the extent that our observation operator is able to accurately describe the relation between the physical and chemical particle characteristics and their optical properties.

2. It is equally essential to accurately estimate the contribution of the uncertainties in the aerosol optics model, i.e., to estimate the forward-model error $\epsilon_f$. If we underestimate this error, we will rely too much on the measurements than we should, thus assimilating noise. If we overestimate this error, we will waste information contained in the observations. In practice, one way to estimate $\epsilon_f$ is to compute optical properties while varying the particles' size, morphology, and

[3]A more realistic optics model, such as the one investigated in Andersson and Kahnert (2016) would help to reduce the observation standard deviation. For future studies, such a model should be linearised and investigated in a similar way.

dielectric properties within typical ranges. The resulting variation in the optical properties then allows us to estimate $\epsilon_f$. (For a review of aerosol optics modelling see Kahnert et al. (2014, 2016) and references therein).

In Table 2 we sorted the results for $N_s$ and $H$ by different values of the observation standard deviation. However, it is important to realise that the results also depend on the background error standard deviation, or, more precisely, on how large the background error standard deviations are compared to the observation error standard deviations. Johnson et al. (2005a) made this point very explicit. They discussed an idealised case with diagonal background error covariance matrix $\mathbf{B} = \sigma_b^2 \mathbf{1}$ and observation error covariance matrix $\mathbf{R} = \sigma_o^2 \mathbf{1}$. They considered the case of direct measurements, i.e., the model variables and the observed parameters are the same type of variables. Under such idealised conditions, they showed that one can maximise the amount of information that can be obtained from the observations by optimising the regularisation parameter $\sigma_b/\sigma_o$ (or, equivalently, the regularisation parameter $\sigma_o^2/\sigma_b^2$). In our more general case, instead of $\sigma_b$ we need to consider the full matrix $\mathbf{B}^{1/2}$, instead of $\sigma_o^{-1}$ we need to consider $\mathbf{R}^{-1/2}$, and in order to compare the two matrices we need to first transform $\mathbf{B}^{1/2}$ from model to observation space according to $\mathbf{H} \cdot \mathbf{B}^{1/2}$. So in place of $\sigma_b/\sigma_o$ we need to consider the more general quantity $\mathbf{R}^{-1/2} \cdot \mathbf{H} \cdot \mathbf{B}^{1/2}$, and we need to diagonalise it by a singular value decomposition according to Eq. (6). Thus the singular values $w_i$ generalise the parameter $\sigma_b/\sigma_o$. The latter applies to the case of direct observations and error covariance matrices that are proportional to unit matrices. The former apply to the general case of non-diagonal error covariance matrices and indirect observations.

From this we learn that the singular values $w_i$ provide us with a (however abstract) means to quantify how the background standard deviations compare to the observation standard deviations. We pick one of the columns in Tab. 2, namely, the one for $\sigma_o = 50 \%$, and expand it in Tab. 3. We show the singular values $w_i$, as well as their contributions $N_s^i = w_i^2/(1+w_i^2)$ and $H_i = 0.5 \log_2(1 + w_i^2)$ to the sums in Eqs. (7) and (8), respectively. The results reveal that the singular values $w_i$ can decrease quite rapidly from the largest to the smallest value (see, e.g., case No. 6 in the table). However, the corresponding contribution $N_s^i$ to the number of signal degrees of freedom changes rather smoothly. Even those singular values that are only slightly larger than 1 make contributions $N_s^i$ that lie close to 1 (see, e.g., $i = 4$ in case No. 6). However, once $w_i$ falls below 1, the corresponding contribution $N_s^i$ becomes much smaller than 1 (see $i = 5$ in case No. 6).

Let us now compare the different subsets of parameters in Tab. 2 and 3. In case No. 1 we observe a single parameter that provides a single degree of freedom. In cases No. 2 and 4 we observe two parameters, which nearly doubles $N_s$. Comparison of these two cases shows that it does not make a significant difference whether we observe backscattering coefficients at different wavelengths, or both extinction and backscattering coefficients each at a single wavelengths. In either case the measurements provide roughly the same amount of information (in terms of $N_s$ or $H$). The same is true when considering three observational parameters (compare cases No. 3 and 5). The $3\beta + 2\alpha$ case (No. 6) clearly provides the largest amount of information in comparison to the other cases. However, as we saw in Tab. 2, observation errors that are large in comparison to the background errors can significantly reduce the effective information that can assimilated into a model.

**Table 3.** Signal degrees of freedom $N_s$ and change in entropy $H$ for the lowest model layer (closest to the surface). Also shown are the singular values $w_i$ and their contributions $N_s^i$ and $H_i$ to $N_s$ and $H$, respectively. The results have been obtained by assuming an observation standard deviation of 50 %.

| No. | Parameters | $i$ | $w_i$ | $N_s^i$ | $H_i$ | $N_s$ | $H$ |
|-----|-----------|-----|-------|---------|-------|-------|-----|
| 1. | $\beta_3$ | 1 | 38.2 | 1.00 | 5.26 | 1.00 | 5.26 |
| 2. | $\beta_1, \beta_2$ | 1 | 108 | 1.00 | 6.76 | 1.97 | 9.36 |
|    |            | 2 | 6.00 | 0.97 | 2.61 |       |      |
| 3. | $\beta_1, \beta_2, \beta_3$ | 1 | 115 | 1.00 | 6.84 | 2.71 | 10.5 |
|    |            | 2 | 6.54 | 0.98 | 2.73 |       |      |
|    |            | 3 | 1.68 | 0.74 | 0.97 |       |      |
| 4. | $\beta_3, k_3$ | 1 | 83.3 | 1.00 | 6.38 | 1.92 | 8.22 |
|    |            | 2 | 3.43 | 0.92 | 1.84 |       |      |
| 5. | $\beta_1, \beta_2, k_2$ | 1 | 128 | 1.00 | 7.00 | 2.77 | 11.24 |
|    |            | 2 | 8.71 | 0.99 | 3.13 |       |      |
|    |            | 3 | 1.90 | 0.78 | 1.10 |       |      |
| 6. | $\beta_1, \beta_2, \beta_3, k_2, k_3$ | 1 | 153 | 1.00 | 7.26 | 3.89 | 12.9 |
|    |            | 2 | 9.52 | 0.99 | 3.26 |       |      |
|    |            | 3 | 1.94 | 0.79 | 1.13 |       |      |
|    |            | 4 | 1.63 | 0.73 | 0.93 |       |      |
|    |            | 5 | 0.79 | 0.38 | 0.35 |       |      |

## 3.2 Numerical inverse-modelling test

We integrated the findings of 3.1 into our 3DVAR program by constraining the algorithm to varying only the signal-related model variables. To illustrate the method we conduct a numerical test as described in Sect. 2.5. We perform a 3DVAR analysis by assimilating "$3\beta + 2\alpha$" profiles, i.e., synthetic lidar measurements of $\beta_{\mathrm{sca}}$ at the three wavelengths 1064, 532, and 355 nm together with $k_{\mathrm{ext}}$ at the two wavelengths 532 and 355 nm. Thus in our case the number of singular values in each vertical layer is $K = 5$. We assume an idealised situation in which the observation standard deviation is only 10 %. As we see in Table 2 (case No. 6), the number of signal degrees of freedom is $N_s = 4.9$ in this case. So we roughly have as many signal degrees of freedom as we have measurements.

Figure 1 shows vertical profiles of selected aerosol components, namely (from top to bottom): organic carbon (OC) in the 3rd size bin (OC-3), OC in the 4th size bin (OC-4), elemental carbon (EC) in the 3rd size bin (EC-3), and mineral dust in the 1st size bin (DUST-1). The reference and background mixing ratios are shown in black and green, respectively. The 3DVAR analysis was first performed without any constraints; the results are shown in the left column by the blue line. Then the 3DVAR analysis

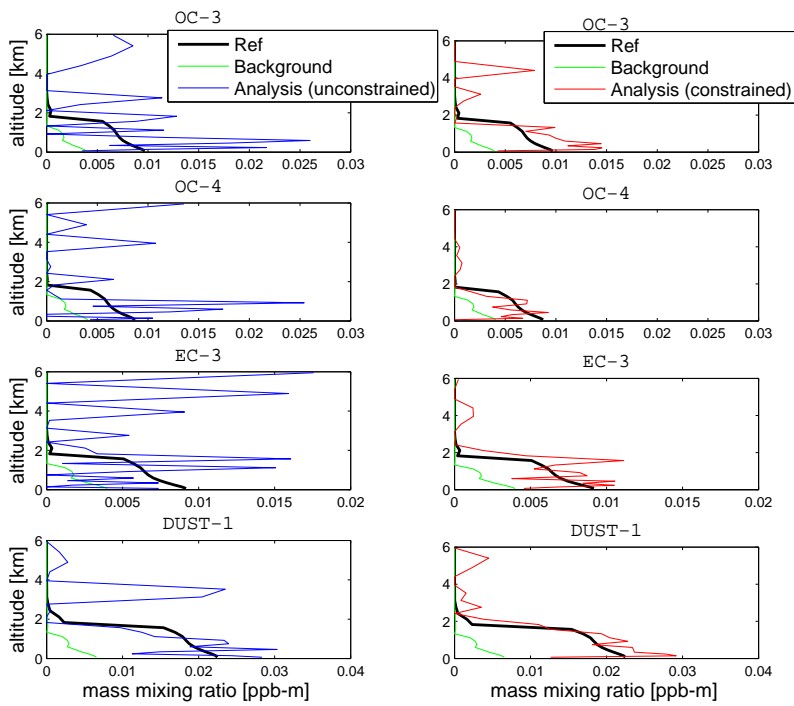

**Figure 1.** Vertical profiles of selected aerosol components in different size bins. From top to bottom: organic carbon in the 3rd size bin (OC-3), OC in the 4th size bin (OC-4), elemental carbon in the 3rd size bin (EC-3), and dust in the 1st size bin (DUST-1). The reference results are shown in black, and the background (first guess) estimate is shown in green. The unconstrained 3DVAR analysis results are presented in the left panels in blue, the constrained 3DVAR analysis results are shown in the right panels in red.

was repeated with the constraints in Eq. (10) and (11); the results are represented in the right column by the red line. Clearly,
the unconstrained analysis (blue lines in the left panels) yields results that oscillate quite erratically in the vertical direction.
Also, the unconstrained analysis can yield conspicuously high values at higher altitudes, even though both the reference and
background values are both close to zero. By contrast, the constrained analysis (red lines in the right panels) yields results that
better agree with the reference results. The noisiness in the vertical direction is significantly reduced, and the results at higher
altitudes are generally lower than those obtained with the unconstrained analysis.
Figure 2 shows analogous results for the mass mixing ratios of different aerosol components, each summed over all size bins.
The aerosol components are (from top to bottom): elemental carbon (EC), organic carbon (OC), mineral dust (DUST), sea salt
(NaCl), secondary inorganic aerosols (SIA, i.e., the sum over all sulphate, nitrate, and ammonium species), and PM10 (i.e., the
sum over all aerosol components). Clearly, the constrained analysis faithfully retrieves both PM10 and SIA. The unconstrained
analysis performs almost equally well for these two variables. Sea salt and mineral dust are not well retrieved from the mea-
surements in either the constrained or unconstrained approach. EC and OC are very well retrieved by the constrained analysis.

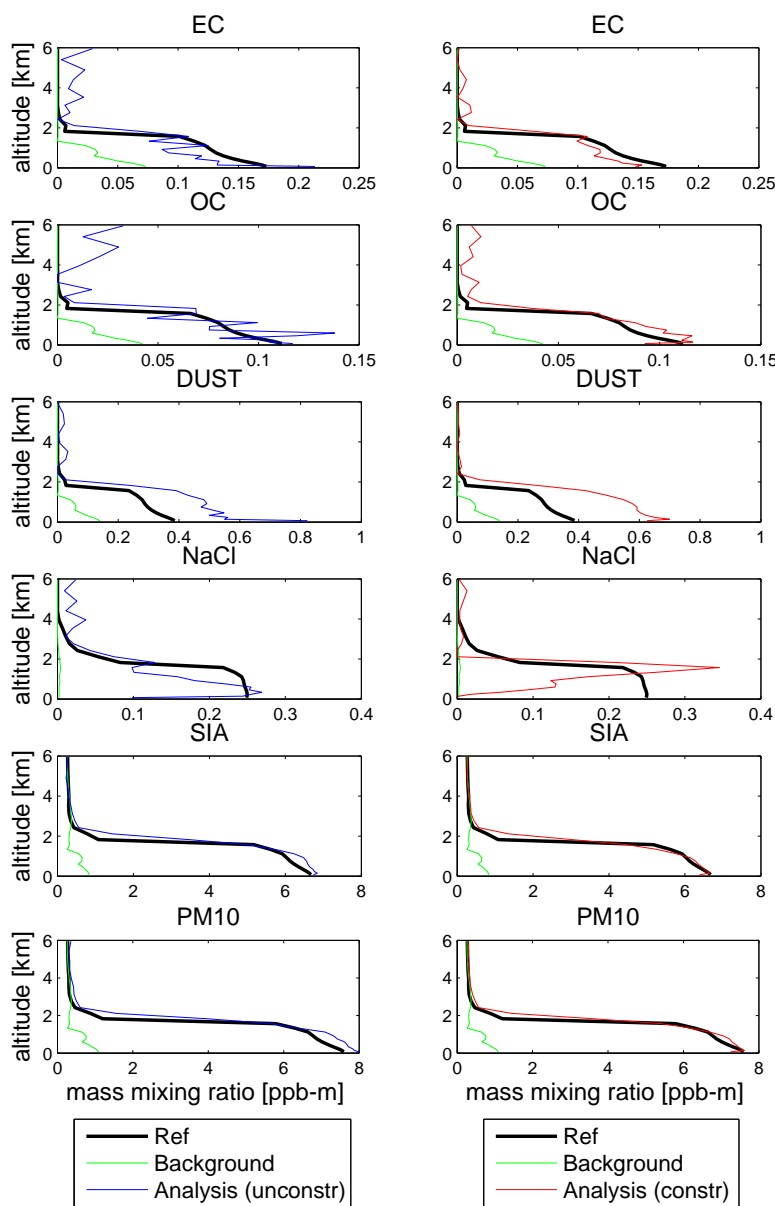

**Figure 2.** As Fig. 1, but for the total mass mixing ratio (summed over all size bins). The components are (from top to bottom): EC, OC, mineral dust, sea salt, secondary inorganic aerosols (sum of all sulphate, nitrate, and ammonium species), and PM10 (sum of all aerosol components).

For these components, the unconstrained analysis has a very small bias compared to the reference results, but it is considerably
more noisy (i.e., oscillating in the vertical direction) than the constrained analysis. We also see, again, that the mixing ratios

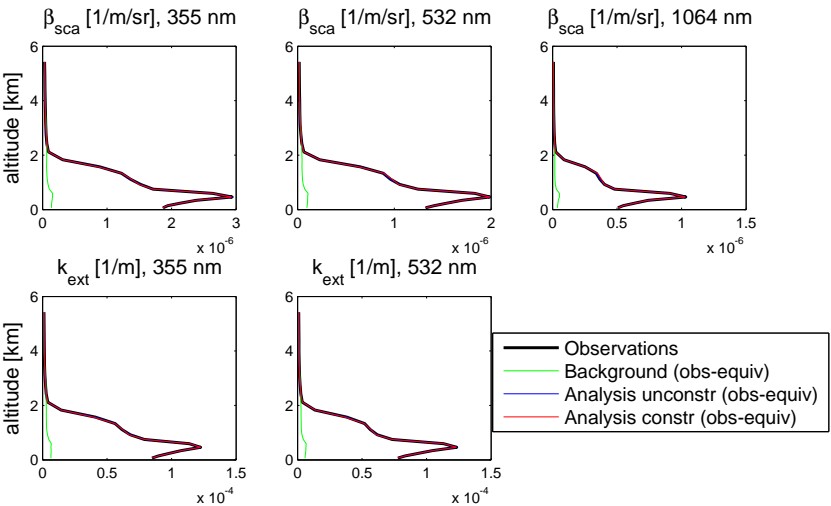

**Figure 3.** Observations (black solid line), and observation-equivalents of the background estimate (green), and of the unconstrained (blue) and constrained (red) 3DVAR analysis. The optical parameters and wavelengths are indicated above each panel.

at higher altitudes obtained with the unconstrained analysis can be unreasonably high. This is especially pronounced for OC.
In general, however, the problems we encounter in the unconstrained analysis are less pronounced in Fig. 2 than in Fig. 1. A
possible explanation is that SIA may be most strongly related to the measurement signal, and SIA is dominating the aerosol
mass in this case. We will return to this point shortly. Another possible factor is that the noise in the analysis can be damped
by summing up results over several size bins.
Figure 3 shows the observations (black) as well as the observation-equivalents of the background estimate (green) and the
unconstrained (blue) and constrained (red) 3DVAR analysis for all five observations. We learn from this figure that the analysis
follows the observations faithfully. The reason for this is that we assumed that the observations were highly accurate with an
error standard deviation of only 10 %. In fact, the difference between the observation-equivalent analysis and the observations
deviate by even less than 10 %. However, our tests confirmed that an increase in the observation error eventually results in
analysis results of which the observation-equivalent increasingly deviates from the observations (not shown).
We have seen that the analysis provides a reasonable, but, as expected, not a perfect answer to the inverse problem. We have
further seen that at the observation site it relies more on the observations than on the background estimate. Most importantly,
we have seen that the constraints introduced in the 3DVAR algorithm suppress noise in the analysis, especially in EC and OC.
However, the previous figures do not provide us with any direct insight of how exactly the constraints accomplish this. To learn
more about that we need to inspect the analysis in the abstract phase space of the transformed model variables $\delta \boldsymbol{x}'$. (Recall that
we defined this variable in Eq. (9) as $\delta \boldsymbol{x}' = \mathbf{V}_R^T \cdot \mathbf{B}^{-1/2} \cdot (\boldsymbol{x} - \boldsymbol{x}_b)$). Figure 4 shows vertical profiles of a selection of the, in total,
20 variables $\delta x_i'$. The background estimate corresponds to $\delta x_i' = 0$ and is represented by the green line. The unconstrained
3DVAR analysis increment is represented by the blue line, the constrained 3DVAR analysis increment is shown by the red

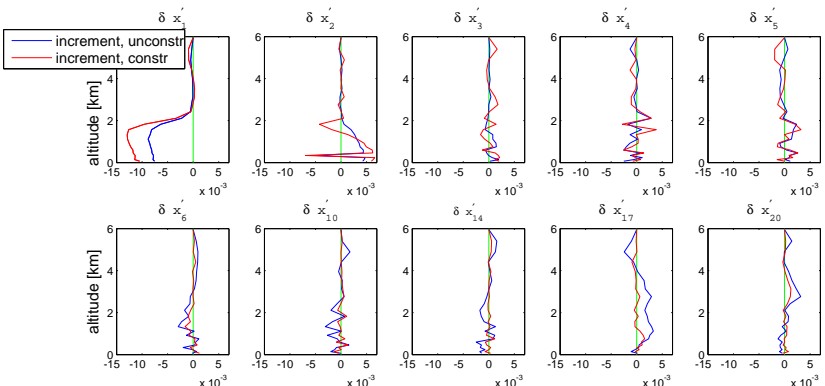

**Figure 4.** Vertical profiles of the transformed model variables $\delta\boldsymbol{x}'$, defined in Eq. (9). The figure shows results obtained with the constrained (red) and unconstrained (blue) 3DVAR analysis.

line. The first five phase space elements in the top row are the signal-related control variables. Generally, the magnitude of the constrained increments (red) is larger than that of the unconstrained increments (blue). The noise-related phase space elements, five of which are shown in the bottom row, display the opposite behaviour. The constrained increments are close to zero, as they should. The unconstrained elements consistently show higher magnitudes than the constrained elements. However, we also see that the unconstrained analysis does produce increments that are largest for the two elements $\delta x_1'$ and $\delta x_2'$, which most strongly relate to the measurement signal. Based on our single test case we cannot say if this is a lucky coincidence or a consistent property. If the latter, it may indicate that we are using rather reasonable background error statistics, so that the analysis increment in observation space is distributed to the different variables in model space in a sensible way. If the former, it could be the case that the success of the unconstrained analysis is largely dependent on whether or not those aerosol components dominate the total aerosol mass that most strongly relate to the signal degrees of freedom. (In our case the total mass is dominated by SIA, which is very well retrieved by the analysis).

Finally, we want to obtain a better understanding of how the aerosol components $\boldsymbol{x}$ in model space, or their increments $\delta\boldsymbol{x}$, are linked with the signal-related phase-space elements $\delta\boldsymbol{x}'$. To this end we inspect the first five row vectors of the transformation matrix $\mathbf{V}_R^T{\cdot}\mathbf{B}^{-1/2}$ in Eq. (9). The magnitude of these elements can be taken as a measure for how much each aerosol component of $\delta\boldsymbol{x}$ in model space contributes to the signal-related elements of $\delta\boldsymbol{x}'$, Figure 5 shows $|\,(\mathbf{V}_R^T{\cdot}\mathbf{B}^{-1/2})_{ij}\,|$ for $i = 1,\ldots,5$, and for $j = 1,\ldots,20$, where 5 is the number of signal-related phase-space elements, and 20 is the number of aerosol components in model space. Results are shown for model layers 2 (left column) and 22 (right column), which correspond to altitudes of about $100\,\mathrm{m}$ and $6\,\mathrm{km}$, respectively. The x-axis shows sea salt (NaCl), EC, OC, and dust, each in four size bins, as well as the four SIA components, i.e., sulphates (SOX) other than $(\mathrm{NH}_4)_2\mathrm{SO}_4$, ammonium sulphate (AS), ammonium nitrate (AN), and nitrates (NOX) other than $\mathrm{NH}_4\mathrm{NO}_3$.

Comparison of the two columns clearly demonstrates that the elements of the transformation matrix can vary considerably with vertical layer (or, more generally, with location). This is because the error covariance matrix $\mathbf{B}$ varies with location, and

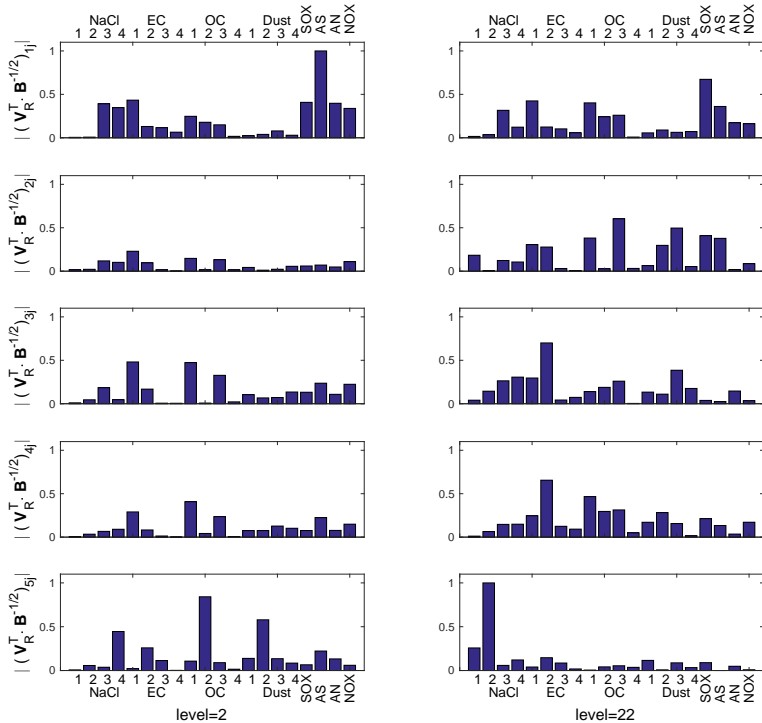

**Figure 5.** The first five rows (from top to bottom) of the matrix $\mathbf{V}_R^T \cdot \mathbf{B}^{-1/2}$ at the observation site, and for model layers 2 (left) and 22 (right). The y-values are normalised by dividing them by the maximum element. The x-axis indicates the aerosol components in model space to which the elements of the row vectors correspond, namely, sea salt (NaCl), EC, OC, and dust, each in four size bins, as well as the four SIA components: sulphates (SOX) other than $(NH_4)_2SO_4$, ammonium sulphate (AS), ammonium nitrate (AN), and nitrates (NOX) other than $NH_4NO_3$.

the matrix $\mathbf{R}$ varies from one observation site to another (in our case, from one altitude to another). Hence the matrix $\mathbf{V}_R$ is
also dependent on location — see Eq. (6). Consequently, it is very difficult to draw general conclusions about which aerosol
components make a dominant contriubution to the signal-related phase-space variables; this can vary with location, and it can
vary for different data sets.
However, in our case the SIA components consistently make a strong contriubution to the first signal-related element $\delta x_1'$.
Since SIA is dominating the aerosol mass mixing ratio in this test case, the analysis was able to retrieve PM10. We also see that
the dust components make only a weak contribution to most of the signal-related elements $\delta x_i'$, especially to the first one. This
is a likely explanation for the difficulties encountered in retrieving the dust mass mixing ratio. Sea salt is more complicated.
Size bins 3 and 4 do contribute considerably to $\delta x_1'$, and also to some of the other four increments, while size bins 1 and 2 do
not make a significant contribution to most of the five signal-related control variables. In our case the sea salt mass is strongly
dominated by the second size bin (not shown). This explains the difficulties we encountered in the retrieval of sea salt.

## 4  Summary and conclusions

We have quantified the information content of multiwavelength lidar measurements with regard to the chemical composition of aerosol particles. Different combinations of extinction and backscattering observations at several wavlengths have been investigated by determining the singular values of the scaled observation operator, by computing the number of signal degrees of freedom $N_s$, and by calculating the reduction in Shannon entropy $H$ caused by taking measurements. We first quantified $N_s$ and $H$ as a function of observation standard deviation $\sigma_o$. The information content of the observations, as expressed by $N_s$ and $H$, decreased as $\sigma_o$ was increased. This became the more pronounced the larger the number of simultaneously observed parameters was.

The observation error depends not only on the measurement error, but also on the forward-model error. The latter depends on the uncertainties in the aerosol-optics model. This highlights the importance of developing accurate aerosol optics models and of obtaining an accurate estimate of the observation error, especially of the uncertainty in the aerosol optics model. This is a prerequisite for extracting as much information as possible from the measurements, while avoiding to extract noise rather than signal. More often than not, computational limitations and lack of knowledge force us to introduce simplifying assumptions about the particles' morphologies. However, we know that aerosol optical properties can be highly sensitive to the shape (Mishchenko et al. (1997); Kahnert (2004)), small-scale surface roughness (Kahnert et al., 2012b), inhomogeneity (Mishchenko et al., 2014; Kahnert, 2015), aggregation (Fuller and Mackowski, 2000; Liu and Mishchenko, 2007; Kahnert and Devasthale, 2011), irregularity (Muinonen, 2000; Bi et al., 2010), porosity (Vilaplana et al., 2006; Lindqvist et al., 2011; Kylling et al., 2014), and combinations thereof (Lindqvist et al., 2009; Kahnert et al., 2013; Lindqvist et al., 2014). We need to know how much these sources of uncertainty contribute to the observation standard deviation. One way of estimating this is to compare aerosol optical properties computed with simple shape models to either measurements or to computations based on more realistic particle shape models — see Kahnert et al. (2014) for a recent review and a more detailed discussion.

The singular values of the scaled observation operator provide us with an abstract measure to compare the standard deviations of the background (prior) estimate to those of the observations. The reason why this is a rather abstract measure is because background and observation errors are, in general, in different spaces and cannot be directly compared. However, we constructed a mapping that transforms the state vector in physical (model) space to an abstract phase space in which the components of the state vector can be partitioned into signal-related and noise-related components. The singular values indicate to what extent the signal-related phase-space variables can be constrained by the measurements. We exploited this fact by constructing weak constraints in a 3DVAR data assimilation code, which limited the assimilation algorithm to acting on the signal-related phase-space variables only (hereafter referred to as the *constrained analysis*). The idea was to maximise the use of information, while avoiding the risk of assimilating noise by over-using the measurements. Thus, our main hypothesis was that the constrained analysis will yield less noisy results than the unconstrained analysis. Numerical tests confirmed this hypothesis. Notably in the case of elemental carbon (EC) and organic carbon (OC) the unconstrained analysis gave mixing ratios that oscillated considerably in the vertical direction. The constrained analysis results were considerably less noisy.

When mapped into observation space, the analysis result closely reproduced the measurements. When viewed in the abstract phase space, we found that the constrained analysis did, indeed, yield noise-related components that were close to zero, as they should. This was not so in the unconstrained analysis. Also, the magnitude of the signal-related phase-space components was generally larger in the constrained analysis than in the unconstrained analysis. This confirms that the constraints we introduced work as intended.

In our specific test case secondary inorganic aerosol components were most faithfully retrieved by the inverse modelling solution, followed by organic and black carbon. Dust and seasalt mass mixing ratios were more challenging to retrieve. We could explain this by inspecting the linear coefficients in the transformation from physical space to the abstract phase space. We found that those aerosol components that had the largest weight in the transformation were most faithfully retrieved by the analysis. However, these linear coefficients depend on the background error covariances (which can change with location), and on the observation error variances. Therefore, it is difficult to draw general conclusions about which aerosol components are most easily retrieved by a given set of measurements.

The results presented here suggest further questions for future studies. We have performed this investigation with a mass transport model, thus focusing on the information content of optical measurements with respect to the chemical composition of aerosols. When we include aerosol microphysical processes, then the model delivers the aerosols' size distribution, as well as their size-resolved chemical composition. This makes the problem quite different from that we investigated here. First, the dimension of the model space is considerably larger for an aerosol microphysics transport model. Constraining such a model with limited information from measurements becomes even more challenging than in the case of a mass transport model. On the other hand, an aerosol microphysics model delivers information on the particles size distribution and mixing state. Therefore, this would require us to make fewer assumptions in the aerosol optics model, which may reduce the observation error. The present study could be extended to investigate the information contained in extinction and backscattering measurements for simultaneously constraining the chemical composition and the size of aerosol particles.

Another important issue concerns the choice of the aerosol optics model. In the present study we employed a simple homogeneous-sphere model in which all chemical components were assumed to be externally mixed. There is little one can put forward in defence of this model other than pure convenience. [Regarding the applicability of simplified model particles in atmospheric optics see the review by Kahnert et al. (2014)]. As a result of the external-mixture assumption, the observation operator is linear, which is a prerequisite for much of the theoretical foundations of this study — see the appendix for details. However, it has been demonstrated that drastically simplifying assumptions, such as the external-mixture approximation, can give model results for aerosol optical properties that differ substantially from those obtained with more realistic nonlinear optics models (Andersson and Kahnert, 2016). It would therefore be important to extend the present study to include more accurate and realistic optics models. A first step could be to analyse the degree of nonlinearity of optics models that account for internal mixing of different aerosol species. If they turn out to be only mildly nonlinear, then one can linearise them and work with the Jacobian of the nonlinear observation operator. Otherwise the theoretical methods employed in this paper would have to be extended in order to accommodate nonlinear observation operators.

## Appendix A: Inverse problems

Suppose we have a system described by a set of variables $x_1, \ldots, x_n$, summarised in a vector $\boldsymbol{x}$. Suppose also that we have an operator $\hat{H} : \mathbb{R}^n \to \mathbb{R}^m$, $\boldsymbol{x} \mapsto \boldsymbol{y} = \hat{H}(\boldsymbol{x})$ that allows us to compute a set of variables $y_1, \ldots, y_m$, summarised in a vector $\boldsymbol{y}$. To take a specific example, we may think of $\boldsymbol{x}$ as a vector of mass mixing ratios of chemical aerosol species, $\boldsymbol{y}$ as a set of aerosol optical properties, and $\hat{H}$ as an aerosol optics model. The operator $\hat{H}$ maps from model space into observation space, which allows us to compare model output and observations. We consider the following two problems:

1. **Direct problem:** Given $\boldsymbol{x}$ and $\hat{H}$, calculate $\boldsymbol{y} = \hat{H}(\boldsymbol{x})$.

2. **Inverse problem:** Given $\boldsymbol{y}$ and $\hat{H}$, solve $\boldsymbol{y} = \hat{H}(\boldsymbol{x})$ for $\boldsymbol{x}$.

A pair of such problems is inverse *to each other*; it is, therefore, somewhat arbitrary which problem we choose to call the direct problem, and which one we call the inverse problem. However, one of the problems is usually *well-posed*, while the other one is *ill-posed*. Such is also the case in aerosol optics modelling. It is customary to call the well-posed problem the *direct problem*, and the ill-posed one the *inverse problem*.

An equation $\boldsymbol{y} = \hat{H}(\boldsymbol{x})$ is called *well-posed* if it has the following properties:

1. **Existence:** For every $\boldsymbol{y} \in \mathbb{R}^m$, there is at least one $\boldsymbol{x} \in \mathbb{R}^n$ for which $\boldsymbol{y} = \hat{H}(\boldsymbol{x})$.

2. **Uniqueness:** For every $\boldsymbol{y} \in \mathbb{R}^m$, there is at most one $\boldsymbol{x} \in \mathbb{R}^n$ for which $\boldsymbol{y} = \hat{H}(\boldsymbol{x})$.

3. **Stability:** The solution $\boldsymbol{x}$ depends continuously on $\boldsymbol{y}$.

If any of these properties is not fulfilled, then the problem is called *ill-posed*.

## Appendix B: Three-dimensional variational data assimilation

Data assimilation is usually employed for constraining models by use of measurements, but it can also be used to solve inverse problems. Here we focus on one specific data assimilation method known as three-dimensional variational data assimilation, or 3DVAR.

In a CTM we discretise the geographic domain of interest into a three-dimensional grid. In each grid cell, the aerosol particles are characterised by the mass mixing ratio of each chemical component in the aerosol phase, such as sulphate, nitrate, ammonium, mineral dust, black carbon, organic carbon, and sea salt. Suppose we summarise all these mass mixing ratios from all grid cells into one large vector $\boldsymbol{x} \in \mathbb{R}^n$. The model provides us with a first guess of the atmospheric aerosol state, known as a *background estimate* $\boldsymbol{x}_b$.[4] Suppose also that we have $m$ observations, which we summarise in a vector $\boldsymbol{y} \in \mathbb{R}^m$. We further have an observation operator $\hat{H} : \mathbb{R}^n \to \mathbb{R}^m$, $\boldsymbol{x} \mapsto \hat{H}(\boldsymbol{x})$ that maps the state vector $\boldsymbol{x}$ from model space to observation

---

[4]In the remote sensing and inverse modelling community, the background estimate is more commonly referred to as the *a priori* estimate.

space[5]. We further denote by $\boldsymbol{x}_t$ the true state of the atmosphere, by $\boldsymbol{\epsilon}_b = \boldsymbol{x}_t - \boldsymbol{x}_b$ the error of the background estimate, and by
$\boldsymbol{\epsilon}_o = \hat{H}(\boldsymbol{x}_t) - \boldsymbol{y}$ the observation error.[6] The background and observation errors are assumed to be unbiased and uncorrelated
with each other. Then their joint probability distribution becomes separable, i.e.
$$P(\boldsymbol{\epsilon}_b, \boldsymbol{\epsilon}_o) = P_b(\boldsymbol{\epsilon}_b) P_o(\boldsymbol{\epsilon}_o). \tag{B1}$$
The true state of the atmosphere is, of course, unknown. Therefore, our definition of the errors and their probability distri-
bution is only of conceptual use, but not of any practical value. However, we can reinterpret the probability distributions by
replacing $\boldsymbol{\epsilon}_b$ in the argument of $P_b$ with $\boldsymbol{x} - \boldsymbol{x}_b$, and by replacing $\boldsymbol{\epsilon}_o$ in the argument of $P_o$ with $\hat{H}(\boldsymbol{x}) - \boldsymbol{y}$. We further assume
that both the background and the observation errors are normally distributed. Thus we may write
$$P_b(\boldsymbol{x}) \;=\; (2\pi \,|\, \mathbf{B} \,|)^{-1/2} \exp\left(-\frac{1}{2}(\boldsymbol{x} - \boldsymbol{x}_b)^T \cdot \mathbf{B}^{-1} \cdot (\boldsymbol{x} - \boldsymbol{x}_b)\right) \tag{B2}$$
$$P_o(\boldsymbol{x}) \;=\; (2\pi \,|\, \mathbf{R} \,|)^{-1/2} \exp\left(-\frac{1}{2}(\hat{H}(\boldsymbol{x}) - \boldsymbol{y})^T \cdot \mathbf{R}^{-1} \cdot (\hat{H}(\boldsymbol{x}) - \boldsymbol{y})\right). \tag{B3}$$
Here $\mathbf{B}$ and $\mathbf{R}$ denote the covariance matrices of the background and observation errors, respectively, and $|\cdot|$ denotes the
matrix determinant. In this form, $P_b(\boldsymbol{x})$ represents the probability that the atmospheric aerosol particles are found in state $\boldsymbol{x}$,
given a background estimate $\boldsymbol{x}_b$ with error covariance matrix $\mathbf{B}$. Similarly, $P_o(\boldsymbol{x})$ is the probability that the system is found in
state $\boldsymbol{x}$, given measurements $\boldsymbol{y}$ with error covariances $\mathbf{R}$.[7]
Equations (B1)–(B3) can be summarised in the form
$$P(\boldsymbol{x}) \;=\; \frac{1}{2\pi(|\, \mathbf{B} \,|\cdot|\, \mathbf{R} \,|)^{1/2}} \exp(-J(\boldsymbol{x})) \tag{B4}$$
$$J(\boldsymbol{x}) \;=\; \frac{1}{2}\left[(\boldsymbol{x} - \boldsymbol{x}_b)^T \cdot \mathbf{B}^{-1} \cdot (\boldsymbol{x} - \boldsymbol{x}_b) + (\hat{H}(\boldsymbol{x}) - \boldsymbol{y})^T \cdot \mathbf{R}^{-1} \cdot (\hat{H}(\boldsymbol{x}) - \boldsymbol{y})\right],$$
$$\tag{B5}$$
where $J$ is suggestively called the cost function, since it can be interpreted as a measure for how "costly" it is for a state $\boldsymbol{x}$ to
simultaneously deviate from the background estimate and the measurements within the permitted error bounds. The deviations
are weighted with the inverse error covariance matrices. For instance, this means that for measurements with a small error
variance, a deviation $\hat{H}(\boldsymbol{x}) - \boldsymbol{y}$ becomes "more costly".

---

[5]The optics model $\hat{H}$ usually has to invoke assumptions about physical aerosol properties that are relevant for the optical properties, but not provided by the CTM output, e.g. assumptions about the morphology of the particles. If the CTM is a simple mass-transport model without aerosol microphysics, then it is also necessary to invoke assumptions about the size distribution of the aerosols.

[6]We stress, once more, that the observation error must not be confused with the measurement error $\boldsymbol{\epsilon}_m$. The latter contributes to the former, but the observation error contains also other sources of error. For instance, if we deal with morphologically complex particles, but our lack of knowledge forces us to make assumptions and invoke approximations about the particle shapes, then this forward-model error $\boldsymbol{\epsilon}_f$ contributes to the observation error. The same is the case if we lack information about the particles' size distribution. In operational applications the representativity error $\boldsymbol{\epsilon}_r$ can also make a substantial contribution to $\boldsymbol{\epsilon}_o$.

[7]The observation errors are often assumed to be uncorrelated (this is not always true). In such case the matrix $\mathbf{R}$ is diagonal, where the diagonal elements are the observation error variances.

We are interested in the most probable aerosol state of the atmosphere, i.e., in that state $\boldsymbol{x}_a$ for which the probability distribution attains its maximum. This is obviously the case when the argument of the exponential in Eq. (B4) assumes a minimum. Thus we seek to minimise the cost function $J$. The variational method is based on computing the gradient of the cost function, $\nabla_{\boldsymbol{x}} J$, and to use this in a descent algorithm to iteratively search for the minimum of $J$.

In practice it is common to introduce the variable $\delta\boldsymbol{x} = \boldsymbol{x} - \boldsymbol{x}_b$, and use the first-order Taylor expansion of the observation operator,

$$\hat{H}(\boldsymbol{x}) = \hat{H}(\boldsymbol{x}_b) + \mathbf{H} \cdot \delta\boldsymbol{x}, \tag{B6}$$

where the $(m \times n)$-matrix $\mathbf{H}$ denotes the Jacobian of $\hat{H}$ at $\boldsymbol{x} = \boldsymbol{x}_b$. If $\hat{H}$ is only mildly non-linear, and if the components of $\delta\boldsymbol{x}$ are sufficiently small, then we can substitute this first-order approximation into Eq. (B5), which yields

$$J \;=\; J_b + J_o \tag{B7}$$

$$J_b(\delta\boldsymbol{x}) \;=\; \frac{1}{2}\delta\boldsymbol{x}^T \cdot \mathbf{B}^{-1} \cdot \delta\boldsymbol{x} \tag{B8}$$

$$J_o(\delta\boldsymbol{x}) \;=\; \frac{1}{2}\left(\hat{H}(\boldsymbol{x}_b) + \mathbf{H} \cdot \delta\boldsymbol{x} - \boldsymbol{y}\right)^T \cdot \mathbf{R}^{-1} \cdot \left(\hat{H}(\boldsymbol{x}_b) + \mathbf{H} \cdot \delta\boldsymbol{x} - \boldsymbol{y}\right) \tag{B9}$$

The components of the vector $\delta\boldsymbol{x}$ are the *control variables* that are iteratively varied by the algorithm until the minimum of the cost function is found.

The solution to the equation $\nabla_{\boldsymbol{x}} J = \boldsymbol{0}_n$ is a solution to the inverse problem (where $\boldsymbol{0}_n$ denotes the null vector in $n$-dimensional model space); we input the observations $\boldsymbol{y}$ into the algorithm, and as output we obtain a result in model space that is consistent with the measurements (within the given error bounds).[8] What if the measurements contain insufficient information about the state $\boldsymbol{x}$? The algorithm will still provide an answer to the inverse problem, but the missing information will be supplemented by the background estimate $\boldsymbol{x}_b$. The weighting of the two pieces of information, $\boldsymbol{x}_b$ and $\boldsymbol{y}$, is controlled by the respective error covariance matrices. Thus data assimilation is a statistical approach, which can be expected to give good results *on average*, but not in every single time-step of the model run. This can become highly problematic if we only have very few observations, i.e., $m \ll n$, where $n$ is the dimension of the model space. If we allow all model variables to be freely adjusted by the assimilation algorithm in such a severely under-constrained case, then the algorithm may just assimilate noise from the measurements rather than signal, resulting in unreasonable solutions to the inverse problem (e.g. Kahnert, 2009). To avoid such problems, one needs to systematically analyse the information content of the observations and constrain the assimilation algorithm to only operate on the signal degrees of freedom.

---

[8]By solving the equation $\nabla J|_{\boldsymbol{x}=\boldsymbol{x}_a} = \boldsymbol{0}_n$ for the analysed state $\boldsymbol{x}_a$ it can be shown that the solution to the inverse problem is given by $\boldsymbol{x}_a = \boldsymbol{x}_b + \mathbf{K} \cdot (\boldsymbol{y} - \hat{H}(\boldsymbol{x}_b))$, where $\mathbf{K} = \mathbf{B} \cdot \mathbf{H}^T \cdot (\mathbf{H} \cdot \mathbf{B} \cdot \mathbf{H}^T + \mathbf{R})^{-1}$ is known as the gain matrix. This illustrates that the analysis updates the background estimate $\boldsymbol{x}_b$ by mapping the increment $(\boldsymbol{y} - \hat{H}(\boldsymbol{x}_b))$ from observation space to model space by use of the gain matrix. The correlations among the model variables enter into the gain matrix through the matrix $\mathbf{B}$. In our case the vertical correlations are rather weak in comparison to correlations among different aerosol species.

## Appendix C: Information content of measurements

Our ultimate goal is to formulate the data assimilation problem in such a way that the information contained in the measurements is fully exploited, but not over-used. To this end, we first need to know how many independent quantities can be determined from a specific set of measurements. We investigate this question by borrowing ideas from retrieval and information theory — see Rodgers (2000) for more detailed explanations.

The main idea is to compare the variances of the model variables to those of the observations. Only those model variables whose variance is larger than those of the observations can be constrained by measurements. However, to actually make such a comparison poses two problems. The first problem is that one cannot readily compare error covariance *matrices*. The second problem is that model variables and measurements are in different spaces. We first address the second problem.

When we account for observation errors $\boldsymbol{\epsilon}_o$, then the basic relation between model variables and observations is, to first order

$$\boldsymbol{y} = \hat{H}(\boldsymbol{x}_b) + \mathbf{H} \cdot \delta\boldsymbol{x} + \boldsymbol{\epsilon}_o. \tag{C1}$$

The error covariance matrices are given by the expectation values $\mathbf{B} = \langle \delta\boldsymbol{x} \cdot \delta\boldsymbol{x}^T \rangle$, and $\mathbf{R} = \langle \boldsymbol{\epsilon}_o \cdot \boldsymbol{\epsilon}_o^T \rangle$, where the dot denotes a dyadic product.[9] From Eq. (C1) we see that the covariance matrix of $\delta\boldsymbol{y} = \boldsymbol{y} - \hat{H}(\boldsymbol{x}_b)$ is given by $\langle \delta\boldsymbol{y} \cdot \delta\boldsymbol{y}^T \rangle = \mathbf{H} \cdot \mathbf{B} \cdot \mathbf{H}^T + \mathbf{R}$, where we assumed that background and observation errors are uncorrelated. This last equation suggests that we can compare model and observation errors in the same space by transforming the background error covariance matrix from the space of $(n \times n)$ matrices to the space of $(m \times m)$ matrices viz. $\mathbf{H} \cdot \mathbf{B} \cdot \mathbf{H}^T$.

To address the first problem, we diagonalise the covariance matrices by making the following change of variables

$$\delta\tilde{\boldsymbol{x}} = \mathbf{B}^{-1/2} \cdot \delta\boldsymbol{x} \tag{C2}$$

$$\delta\tilde{\boldsymbol{y}} = \mathbf{R}^{-1/2} \cdot (\boldsymbol{y} - \hat{H}(\boldsymbol{x}_b)) \tag{C3}$$

$$\tilde{\mathbf{H}} = \mathbf{R}^{-1/2} \cdot \mathbf{H} \cdot \mathbf{B}^{1/2}. \tag{C4}$$

Here $\mathbf{B}^{1/2}$ denotes the positive square root[10] of the matrix $\mathbf{B}$, and $\mathbf{B}^{-1/2}$ denotes its inverse. The scaled observation operator $\tilde{\mathbf{H}}$ is sometimes referred to as the observability matrix. In the new basis, the cost function in (B7)–(B9) becomes

$$J = \frac{1}{2}\delta\tilde{\boldsymbol{x}}^T \cdot \delta\tilde{\boldsymbol{x}} + \frac{1}{2}\left(\tilde{\mathbf{H}} \cdot \delta\tilde{\boldsymbol{x}} - \delta\tilde{\boldsymbol{y}}\right)^T \cdot \left(\tilde{\mathbf{H}} \cdot \delta\tilde{\boldsymbol{x}} - \delta\tilde{\boldsymbol{y}}\right). \tag{C5}$$

The covariance matrices are now unit matrices. This can also be seen by considering the transformed errors, e.g. $\tilde{\boldsymbol{\epsilon}}_o = \mathbf{R}^{-1/2} \cdot \boldsymbol{\epsilon}_o$ and computing $\langle \tilde{\boldsymbol{\epsilon}}_o \cdot \tilde{\boldsymbol{\epsilon}}_o^T \rangle = \mathbf{R}^{-1/2} \cdot \langle \boldsymbol{\epsilon}_o \cdot \boldsymbol{\epsilon}_o^T \rangle \cdot \mathbf{R}^{-1/2} = \mathbf{1}_{m \times m}$, since $\langle \boldsymbol{\epsilon}_o \cdot \boldsymbol{\epsilon}_o^T \rangle = \mathbf{R}$. (Here, $\mathbf{1}_{m \times m}$ denotes the unit matrix in $m$-dimensional observation space.) Similarly, we find $\langle \delta\tilde{\boldsymbol{x}} \cdot \delta\tilde{\boldsymbol{x}}^T \rangle = \mathbf{1}_{n \times n}$. The covariance matrix of the transformed measurement vector $\delta\tilde{\boldsymbol{y}}$ is given by $\langle \delta\tilde{\boldsymbol{y}} \cdot \delta\tilde{\boldsymbol{y}}^T \rangle = \tilde{\mathbf{H}} \cdot \tilde{\mathbf{H}}^T + \mathbf{1}_{m \times m}$. The first term is the model error covariance term transformed into observation space, while the second term (the unit matrix) is the diagonalised observation error covariance matrix.

---

[9] The expectation value of a discrete variable $a$ that assumes values $a_1, a_2, \ldots, a_n$ with corresponding probabilities $p_1, p_2, \ldots, p_n$ is given by $\langle a \rangle = \sum_{i=1}^{n} p_i a_i$.

[10] A matrix $\mathbf{A}$ is called a square root of a matrix $\mathbf{B}$ if $\mathbf{A}^T \cdot \mathbf{A} = \mathbf{B}$. The *positive* square root of $\mathbf{B}$, which is denoted by $\mathbf{B}^{1/2}$, has the property $\boldsymbol{x}^T \cdot \mathbf{B}^{1/2} \cdot \boldsymbol{x} \geq 0$ for all $\boldsymbol{x}$. If $\mathbf{B}$ is itself positive and symmetric, as is the case for covariance matrices, then the positive square root exists and is unique.

We are still not in a position to make a meaningful comparison of model and observation errors, since the first term, $\tilde{\mathbf{H}} \cdot \tilde{\mathbf{H}}^T$,
is still not diagonal. To make it so we need to perform one more transformation. To this end, we consider the singular value
decomposition of the matrix $\tilde{\mathbf{H}}$,

$$\tilde{\mathbf{H}} = \mathbf{R}^{-1/2} \cdot \mathbf{H} \cdot \mathbf{B}^{1/2} = \mathbf{V}_L \cdot \mathbf{W} \cdot \mathbf{V}_R^T. \tag{C6}$$

Here $\tilde{\mathbf{H}}$ is a $(m \times n)$-matrix, the matrix of the left-singular vectors $\mathbf{V}_L$ is a $(m \times m)$-matrix, the matrix $\mathbf{V}_R$ containing the right-
singular vectors is a $(n \times n)$-matrix, and the $(m \times n)$-matrix $\mathbf{W}$ consists of two blocks. If $m < n$, then the left block of $\mathbf{W}$ is a
$(m \times m)$-diagonal matrix containing the $m$ singular values $w_1, \ldots, w_m$ on the diagonal; the right block is a $(m \times (n-m))$-null
matrix. Similarly, if $m > n$, then the upper block of $\mathbf{W}$ is a $(n \times n)$-diagonal matrix containing the $n$ singular values on the
diagonal, while the lower block is a $((m-n) \times n)$-null matrix.
We now make another change of variables:

$$\delta \boldsymbol{x}' = \mathbf{V}_R^T \cdot \delta \tilde{\boldsymbol{x}} \tag{C7}$$

$$\delta \boldsymbol{y}' = \mathbf{V}_L^T \cdot \delta \tilde{\boldsymbol{y}} \tag{C8}$$

$$\mathbf{H}' = \mathbf{V}_L^T \cdot \tilde{\mathbf{H}} \cdot \mathbf{V}_R. \tag{C9}$$

The matrices $\mathbf{V}_L$ and $\mathbf{V}_R$ are orthogonal, i.e., $\mathbf{V}_L^T \cdot \mathbf{V}_L = \mathbf{1}_{m \times m}$, and similarly for $\mathbf{V}_R$. Thus, substitution of (C7)–(C9) into
(C5) yields

$$J = \frac{1}{2} \delta \boldsymbol{x}'^T \cdot \delta \boldsymbol{x}' + \frac{1}{2} \left( \mathbf{H}' \cdot \delta \boldsymbol{x}' - \delta \boldsymbol{y}' \right)^T \cdot \left( \mathbf{H}' \cdot \delta \boldsymbol{x}' - \delta \boldsymbol{y}' \right). \tag{C10}$$

Evidently, the transformation given in (C7)–(C9) preserves the diagonality of the background and observation error covariance
matrices. What about the covariance matrix $\langle \delta \boldsymbol{y}' \cdot \delta \boldsymbol{y}'^T \rangle$ in the new basis? Using $\boldsymbol{\epsilon}_o' = \mathbf{V}_L^T \cdot \tilde{\boldsymbol{\epsilon}}_o = \mathbf{V}_L^T \cdot \mathbf{R}^{-1/2} \cdot \boldsymbol{\epsilon}_o$, as well as
Eqs. (C1), (C2)–(C4), and (C6)–(C9), we obtain $\langle \delta \boldsymbol{y}' \cdot \delta \boldsymbol{y}'^T \rangle = \mathbf{H}' \cdot \mathbf{H}'^T + \mathbf{1}_{m \times m}$. The contribution of the background error
covariances in this coordinate system is $\mathbf{H}' \cdot \mathbf{H}'^T$, which is a diagonal matrix. This becomes clear from Eqs. (C6) and (C9),
which yields

$$\mathbf{H}' \cdot \mathbf{H}'^T = \mathbf{W} \cdot \mathbf{W}^T, \tag{C11}$$

which is a $(m \times m)$ diagonal matrix. Thus in this coordinate system we can readily compare the diagonal elements of the
transformed background error covariance matrix $\mathbf{H}' \cdot \mathbf{H}'^T$ to the diagonal (unit) elements of the observation error covariance
matrix $\mathbf{1}_{m \times m}$. Roughly, those singular values $w_i$ on the diagonal of $\mathbf{W}$ that are larger than unity correspond to model variables
$\delta x_i'$ that can be controlled by the measurements. Those singular values smaller than unity correspond to model variables that
are only related to noise.
In the above discussion we relied on plausibility arguments. We mention that there are more systematic ways of approaching
the problem. Here we merely state some key results without going into details. The interested reader is referred to chapter 2 in
Rodgers (2000). However, in all approaches the main quantities of interest are always the singular values of the observability
matrix $\mathbf{R}^{-1/2} \cdot \mathbf{H} \cdot \mathbf{B}^{1/2}$.
One can compute the number of signal degrees of freedom $N_s$ from the expectation value of $J_b$ in Eq. (B8). The result can
be expressed in terms of the singular values $w_i$ of the observability matrix:
$$N_s = \sum_{i=1}^{\min\{m,n\}} w_i^2/(1+w_i^2),$$           (C12)
where $n$ is the dimension of model space, and $m$ is the dimension of observation space.
Another approach is based on information theory. Given a system described by a probability distribution function $P(x)$, one
defines the Shannon entropy
$$S(P) = -\int P(x)\log_2\left(\frac{P(x)}{P_0(x)}\right)\mathrm{d}x,$$           (C13)
where $P_0$ is a normalisation factor needed to make the argument of the logarithm dimensionless. A decrease in entropy ex-
presses an increase in our knowledge of the system. For instance, if we initially describe the system by $P_i(x)$, and, after taking
measurements, by $P_f(x)$, then the measurement process has changed the entropy by an amount
$$H = S(P_i) - S(P_f).$$           (C14)
In our case, we assume that all errors are normally distributed. In that case, one can show that
$$H = \frac{1}{2}\sum_{i=1}^{\min\{m,n\}} \log_2(1+w_i^2).$$           (C15)
$H$ can be interpreted as a measure for the information content of a set of measurements.
Our findings so far suggest a general strategy for how to optimise the amount of information that can be extracted from
measurements. First, we need to compute the singular value decomposition in Eq. (C6), as well as the transformation given in
(C2) and (C7), which we can summarise as
$$\delta \boldsymbol{x}' = \mathbf{V}_R^T \cdot \mathbf{B}^{-1/2} \cdot \delta \boldsymbol{x}.$$           (C16)
Then we want to formulate the minimisation of the cost function in such a way that only those components of $\delta \boldsymbol{x}'$ are adjusted
by the assimilation algorithm that correspond to the largest singular values of the matrix $\mathbf{W}$ in (C6). All other elements of
$\delta \boldsymbol{x}'$ should be left alone. In other words, we want to constrain the minimisation of the cost function to the subspace of the
signal degrees of freedom of the state vector. Thus, in order to implement this idea, we first need to discuss how to incorporate
constraints into the theory.
**Appendix D: Minimisation of the cost function with constraints**
In the minimisation of the cost function all elements of the control vector $\delta \boldsymbol{x}$ are independently adjusted until the minimum
of $J$ is found. This may not be a prudent approach if the information contained in the observations is insufficient to constrain
all model variables. In such case one should introduce constraints that reduce the number of independent control variables.
However, this needs to be done in a clever way; the goal is to neither under-use the measurements (thus wasting available
information), nor to over-use them (thus assimilating noise).
For reasons we will explain later we formulate the constraints as weak conditions. However, for didactic reasons as well as
for the sake of completeness, we will also mention how to formulate constraints as strong conditions.

## D1 Minimisation of the cost function with strong constraints

Given $k$ constraints in the form $g_i(\delta \boldsymbol{x})=0$, $i=1,\ldots,k$, the most general way of finding the minimum of $J(\delta \boldsymbol{x})$ under the
constraints $g_i$ is the method of Lagrange multipliers. More specifically, one introduces $k$ Lagrange multipliers $\lambda_1,\ldots,\lambda_k$ and
defines the function

$$L(\delta x_1,\ldots,\delta x_n,\lambda_1,\ldots,\lambda_k) = J(\delta x_1,\ldots,\delta x_n) + \sum_{i=1}^{k} \lambda_i g_i(\delta x_1,\ldots,\delta x_n); \tag{D1}$$

then one solves the minimisation problem

$$\nabla L(\delta x_1,\ldots,\delta x_n,\lambda_1,\ldots,\lambda_k) = \mathbf{0}_{n+k}, \tag{D2}$$

where $\nabla = \nabla_{\delta x_1,\ldots,\delta x_n,\lambda_1,\ldots,\lambda_k}$ is now a $(n+k)$-dimensional gradient operator, and where $\mathbf{0}_{n+k}$ denotes the null vector in an
$(n+k)$-dimensional space. Note that in this general formulation of the problem the constraints can even be nonlinear. We are
specifically interested in linear constraints, which can be expressed in the form $\mathbf{G} \cdot \delta \boldsymbol{x} = \mathbf{0}_k$. Then the constrained minimisation
problem becomes

$$L(\delta \boldsymbol{x}, \boldsymbol{\lambda}) = J(\delta \boldsymbol{x}) + \boldsymbol{\lambda}^T \cdot \mathbf{G} \cdot \delta \boldsymbol{x} \tag{D3}$$

$$\nabla_{\delta \boldsymbol{x}, \boldsymbol{\lambda}} L(\delta \boldsymbol{x}, \boldsymbol{\lambda}) = \begin{pmatrix} \nabla_{\delta \boldsymbol{x}} J(\delta \boldsymbol{x}) + \boldsymbol{\lambda}^T \cdot \mathbf{G} \\ \mathbf{G} \cdot \delta \boldsymbol{x} \end{pmatrix} = \mathbf{0}_{n+k}. \tag{D4}$$

Compared to the unconstrained minimisation problem, the introduction of $k$ constraints has increased the dimension of the
problem from $n$ to $n+k$. Naively, one may have expected that the dimension would, on the contrary, be reduced to $n-k$. This
is indeed the case if the constraints are linear, and if the function $J$ is quadratic, as is the case in Eqs. (B7)–(B9). To see this,
let us first write those equations more concisely in the form

$$J = \frac{1}{2}\left(\delta \boldsymbol{x}^T \cdot \mathbf{Q}_1 \cdot \delta \boldsymbol{x} + \mathbf{Q}_2^T \cdot \delta \boldsymbol{x} + \delta \boldsymbol{x}^T \cdot \mathbf{Q}_2 + \mathbf{Q}_3\right) \tag{D5}$$

$$\mathbf{Q}_1 = \mathbf{B}^{-1} + \mathbf{H}^T \cdot \mathbf{R}^{-1} \cdot \mathbf{H} \tag{D6}$$

$$\mathbf{Q}_2 = \mathbf{H}^T \cdot \mathbf{R}^{-1} \cdot (\hat{H}(\boldsymbol{x}_b) - \boldsymbol{y}) \tag{D7}$$

$$\mathbf{Q}_3 = (\hat{H}(\boldsymbol{x}_b) - \boldsymbol{y})^T \cdot \mathbf{R}^{-1} \cdot (\hat{H}(\boldsymbol{x}_b) - \boldsymbol{y}). \tag{D8}$$

643 Note that the covariance matrices and their inverses are symmetric (i.e. $\mathbf{R}^T = \mathbf{R}$, etc.) The unconstrained minimisation problem

644 requires us to solve the equation $\nabla_{\delta \boldsymbol{x}} J = \mathbf{Q}_1 \cdot \delta \boldsymbol{x} + \mathbf{Q}_2 = \mathbf{0}_n$. Now we want to minimise the cost function subject to the the

645 linear constraints

$$\mathbf{G} \cdot \delta \boldsymbol{x} = \mathbf{0}_k, \tag{D9}$$

where $\mathbf{G}$ is a $(k \times n)$-matrix, $\delta\boldsymbol{x}$ is an $n$-vector, and $\mathbf{0}_k$ is the null-vector in $\mathbb{R}^k$. Let us denote the kernel[11] of $\mathbf{G}$ by $\ker(\mathbf{G})$. Let further $\boldsymbol{z}_1, \ldots, \boldsymbol{z}_{n-k}$ denote a basis of $\ker(\mathbf{G})$. We define the $(n \times (n-k))$-matrix

$$\mathbf{Z} = \left( \begin{array}{ccc} \boldsymbol{z}_1 & \cdots & \boldsymbol{z}_{n-k} \end{array} \right) \tag{D10}$$

the column vectors of which are just the basis vectors of $\ker(\mathbf{G})$. Obviously, $\mathbf{G} \cdot \mathbf{Z} = \mathbf{0}_{k \times (n-k)}$, where $\mathbf{0}_{k \times (n-k)}$ denotes the $((k \times (n-k))$-null matrix. If $\delta\boldsymbol{x}$ is a vector in $\mathbb{R}^n$ for which there exists a vector $\boldsymbol{\xi} \in \mathbb{R}^{n-k}$ such that $\mathbf{Z} \cdot \boldsymbol{\xi} = \delta\boldsymbol{x}$, then we automatically have $\mathbf{G} \cdot \delta\boldsymbol{x} = \mathbf{0}_k$, i.e., $\delta\boldsymbol{x}$ satisfies the linear constraints. Thus we can formulate the constrained minimisation problem by substitution of $\delta\boldsymbol{x} = \mathbf{Z} \cdot \boldsymbol{\xi}$ into Eq. (D5), which yields

$$J = \frac{1}{2} \left( \boldsymbol{\xi}^T \cdot \mathbf{Z}^T \cdot \mathbf{Q}_1 \cdot \mathbf{Z} \cdot \boldsymbol{\xi} + \mathbf{Q}_2^T \cdot \mathbf{Z} \cdot \boldsymbol{\xi} + \boldsymbol{\xi}^T \cdot \mathbf{Z}^T \cdot \mathbf{Q}_2 + \mathbf{Q}_3 \right) \tag{D11}$$

$$\mathbf{0}_k = \nabla_{\boldsymbol{\xi}} J = \mathbf{Z}^T \cdot \mathbf{Q}_1 \cdot \mathbf{Z} \cdot \boldsymbol{\xi} + \mathbf{Z}^T \cdot \mathbf{Q}_2. \tag{D12}$$

Thus we have reduced the $(n+k)$-dimensional constrained minimisation problem given in Eq. (D4) to a problem consisting of the following two steps.

1. Determine a basis of the null space $\ker(\mathbf{G})$; this yields the matrix $\mathbf{Z}$.

2. Solve the unconstrained $(n-k)$-dimensional optimisation problem given in Eq. (D12). From the $(n-k)$-vector $\boldsymbol{\xi}$ that minimises the cost function in (D11), we then obtain the solution $\delta\boldsymbol{x} = \mathbf{Z} \cdot \boldsymbol{\xi}$ that minimises the cost function in (D5) subject to the constraint (D9).

## D2 Minimisation of the cost function with weak constraints

In the approach described in the previous section the solution satisfies the constraints exactly. Therefore, this approach is known as the minimisation of the cost function with *strong constraints*. In the *weak-constraint* approach the constraints only need to be satisfied within specified error bounds.

The formulation of the weak-constraint approach is conceptually quite simple. One incorporates the constraints by adding an extra term to the cost function (B7), i.e.

$$J = J_b + J_o + J_G \tag{D13}$$

$$J_G = \frac{1}{2} \delta\boldsymbol{x}^T \cdot \mathbf{G}^T \cdot \mathbf{B}_G^{-1} \cdot \mathbf{G} \cdot \delta\boldsymbol{x}, \tag{D14}$$

which also gives an extra term in the gradient of the cost function,

$$\nabla_{\delta\boldsymbol{x}} J_G = \mathbf{G}^T \cdot \mathbf{B}_G^{-1} \cdot \mathbf{G} \cdot \delta\boldsymbol{x}. \tag{D15}$$

We will assume that the matrix $\mathbf{B}_G = \mathrm{diag}(\sigma_1^G, \ldots, \sigma_k^G)$ is diagonal, where $k$ is the number of constraints. The "error variances" $\sigma_i^G$ along the diagonal of $\mathbf{B}_G$ allow us to fine-tune the influence of each constraint on the solution. If $\sigma_i^G$ is small, then the $i$th

---

[11]The *kernel* or *null space* of a matrix is the set of all vectors $\boldsymbol{z}$ such that $\mathbf{G} \cdot \boldsymbol{z} = \mathbf{0}$. The kernel is a subspace of the full vector space $\mathbb{R}^n$ with dim $\ker(\mathbf{G}) = n - k$.

constraint is relatively strong, and vice versa. Typically, if the $\sigma_i^G$ are made too large, then there is a risk that the minimisation
algorithm ignores the constraints all together. In that case the solution will be very similar to the unconstrained solution. On the
other hand, if the $\sigma_i^G$ are made too small, then $J_G$ can make the dominant contribution to $J$. In that case, there is a risk that the
minimisation routine largely ignores the observations and returns a solution that lies quite close to the background estimate.

## 678 D3 Constraints designed for making optimum use of the information contained in the observations

We now want to incorporate the results of Section C into the variational data assimilation method. More specifically, we want
to formulate weak constraints, Eq. (D14), based on the singular values of the observation operator in Eq. (C6). To this end,
we make the change of variables given in Eq. (C16). We assume, without loss of generality, that the first $\ell$ singular values are
greater than unity. Thus we only want to use the corresponding components $\delta x_1', \ldots, \delta x_\ell'$ as independent control variables in the
3DVAR algorithm, while the remaining components remain unchanged, at least approximately, within specified error bounds.
If we were to formulate this requirement as a strong constraint, as in Eq. (D9), then it would take the form

$$
\quad \delta \boldsymbol{x}' = \mathbf{V}_R^T \cdot \mathbf{B}^{-1/2} \cdot \delta \boldsymbol{x} = \begin{pmatrix} \delta x_1' \\ \vdots \\ \delta x_\ell' \\ 0 \\ \vdots \\ 0 \end{pmatrix}. \tag{D16}
$$

Thus the matrix expressing the constraints is given by $\mathbf{G} = \mathbf{V}_R^T \cdot \mathbf{B}^{-1/2}$, which is a $(n \times n)$ matrix.
The weak constraint approach is, arguably, more suitable in our case. We have, in the preceding text, frequently used the
terms *signal degrees of freedom* and *noise degrees of freedom*. Although it was conceptually useful to make this distinction, it
is important to stress that there is no sharp boundary between the two. Rather, there is a smooth transition from singular values
$w_1 > w_2 > \cdots > w_\ell \geq 1$ to singular values $1 > w_{\ell+1} > w_{\ell+2} > \cdots > w_K$ ($K = \min\{n, m\}$). For this reason we choose to
formulate the constraints as weak constraints. This allows us to make a smooth transition from free to constrained control
variables, where the transition from one regime to the other can be controlled by the singular values.
In order to apply the weak-constraint approach, we need to substitute the constraint-matrix $\mathbf{G} = \mathbf{V}_R^T \cdot \mathbf{B}^{-1/2}$ into Eq. (D14),
which yields

$$
\quad J_G = \frac{1}{2} \delta \boldsymbol{x}^T \cdot \mathbf{B}^{-1/2} \cdot \mathbf{V}_R \cdot \mathbf{B}_G^{-1} \cdot \mathbf{V}_R^T \cdot \mathbf{B}^{-1/2} \cdot \delta \boldsymbol{x}, \tag{D17}
$$

where $\mathbf{B}_G$ is a $(n \times n)$ matrix. We want to set up this matrix in such a way that we obtain a smooth transition from freely
adaptable control variables $\delta x_1', \ldots \delta x_\ell'$ to increasingly constrained variables $\delta x_{\ell+1}', \ldots \delta x_k', \ldots, \delta x_n'$. One possible choice of
the matrix $\mathbf{B}_G$ would be

$$
\quad \mathbf{B}_G = \sigma_G \operatorname{diag}(w_1, w_2, \ldots, w_\ell, \ldots, w_k, c, \ldots, c), \tag{D18}
$$

where $\sigma_G$ is a free scaling factor, and where the last $n - k$ diagonal elements are equal to a constant $c \ll w_k$ chosen to be much smaller than the smallest singular value $w_k$.

Clearly, how we set up the matrix $\mathbf{B}_G$ is not unique. For instance, a more general choice would be

$$\mathbf{B}_G = \sigma_G \,\mathrm{diag}(w_1^p, w_2^p, \ldots, w_\ell^p, \ldots, w_k^p, c, \ldots, c), \tag{D19}$$

where $c \ll w_k^p$, and where the exponent $p$ would be another parameter that can be employed to tune how steeply the transition from unconstrained to constrained control variables takes place. Yet another choice would be

$$\mathbf{B}_G = \sigma_G \cdot \mathrm{diag}(\mu_1, \mu_2, \ldots, \mu_\ell, \ldots, \mu_k, c, \ldots, c), \tag{D20}$$

$$\mu_i = w_i^2/(1 + w_i^2), \tag{D21}$$

where $c \ll \mu_k$. This ansatz is suggested by Eq. (C12), i.e., each of the elements $\delta x_1', \ldots \delta x_k'$ is weighted with its corresponding contribution to the number of signal degrees of freedom. We tested all three approaches (the one in Eq. (D19) for $p = 2$). These tests showed that the different approaches often yield analysis results that are quite similar. However, in each approach the free parameters $\sigma_G$ and $c$ are tuned to different values. If they are not well tuned, then the analysis tends either toward the background estimate or toward the unconstrained analysis, as explained earlier in the text following Eq. (D15).

## Appendix E:  Practical aspects of the implementation

We will here discuss some practical aspects that are mainly interesting for model developers.

One of the main practical problems is the dimension $n$ of the model space. The grid-size is typically on the order $N_x \times N_y \times N_z \sim 100 \times 100 \times 10$, and the number of aerosol components is on the order of $N_c \sim 10$–$100$. Hence the dimension of the model space is $n \sim 10^6$–$10^7$. In our case, the matrix $\tilde{\mathbf{H}}$ in (C6) is a ($m \times n$) matrix. To numerically perform a singular value decomposition of such a large matrix would be a formidable task.

In variational data assimilation we encounter a similar problem in the inversion of the matrix $\mathbf{B}$. In our 3DVAR code this problem is alleviated by using a so-called spectral formulation. The idea is to make a Fourier-transformation in the horizontal coordinates and to assume that all horizontal error correlations are homogeneous and isotropic. Under these assumptions one obtains one background error covariance matrix for each horizontal wavenumber; each of these matrices has dimension $N_z \times N_c \sim 10^3$–$10^4$. This can further be reduced to about $10^2$ by making a reduced eigenvalue diagonalisation. The details are explained in Kahnert (2008).

In our case we are primarily interested in constraining the aerosol components. Therefore, we formulate our weak constraints in a suitable subspace of the physical space. Suppose, for simplicity, that we have reduced all data to the vertical resolution of our model. Let $\nu_l = 1, \ldots, m_l$ label all measurements that lie within model layer $l$. Suppose further than $(i_\alpha, j_\alpha)$ is the horizontal grid point belonging to observation $\nu_l$ (so that the index $\alpha$ depends on the layer $l$ and the observation $\nu_l$). Consider the reduced background error covariance matrix with elements $B_{k,k'}^{(\alpha,l)} = B_{i_\alpha j_\alpha l k, i_\alpha j_\alpha l k'}$, $k, k' = 1, \ldots, N_c$, and $N_c$ is the number of aerosol components. Consider further the reduced observability matrix with elements $\tilde{H}_{\nu_l,k}^{(l)} =$

$\sum_{k'=1}^{N_c} R_{\nu_l,\nu_l}^{-1/2} H_{m,i_\alpha,j_\alpha l k'} \{(B^{(\alpha,l)})^{1/2}\}_{k',k}$, where $m = m(l,\nu_l)$ labels the $\nu_l$th observation in model layer $l$. Analogous to
Eq. (C6), we now perform a singular value decomposition in the reduced space

$$\tilde{H}_{\nu_l,k}^{(l)} = \sum_{s=1}^{\min\{m_l,N_c\}} (V_L^{(l)})_{\nu_l,s} w_s^{(l)} (V_R^{(l)})_{k,s}. \tag{E1}$$

The dimension of this SVD-problem is now considerably reduced. The number of singular values is equal to $K = \min\{N_c, m_l\}$.
The constraint matrix $\mathbf{G} = \mathbf{V}_R^T \cdot \mathbf{B}^{-1/2}$ reduces to

$$G_{s,k} = \sum_{k'=1}^{N_c} (V_R^{(l)})_{k',s} \{(B^{(\alpha,l)})^{-1/2}\}_{k',k}. \tag{E2}$$

We now invoke the assumption that the constraints computed at the observation site are also valid at neighbouring points, i.e.,
we apply the constraint matrix given in Eq. (E2) in Eq. (D17) according to

$$J_G = \frac{1}{2} \sum_{ijlkk's} \delta x_{ijlk'} G_{s,k'} (B_G^{-1})_s G_{s,k} \delta x_{ijlk}, \tag{E3}$$

where $(B_G)_s$ denotes the diagnonal elements of the matrix given in (D18).[12]
Another aspect concerns the positive square root of the background error covariance matrix, which appears in essential
parts of the theory, namely, in Eqs. (C6) and (D16). In theoretical developments it is, arguably, didactically expedient to work
with the matrix $\mathbf{B}^{1/2}$. But in practice there are numerically more efficient formulations. One such approach is discussed in
Kahnert (2008) in the context of a spectral formulation of the variational method. The spectral formulation is applied to the full
B-matrix in order to reduce the dimension of the problem of diagonalising this matrix. This method is our method of choice
in the formulation of the background and observation terms in the cost function given in Eqs. (B8) and (B9), respectively.
However, in the formulation of the constraint term given in Eq. (D17) we can substantially reduce the dimension of the matrix
$\mathbf{B}$ by working in the reduced space in which only the covariances $\mathbf{B}^{(\alpha,l)}$ among aerosol components are considered. One could
compute the matrix $(\mathbf{B}^{(\alpha,l)})^{-1/2}$ in Eq. (D17) by diagonalising the matrix $\mathbf{B}^{(\alpha,l)}$. However, a numerically much more efficient
approach is to not work with positive square root, but with the so-called Cholesky decomposition[13] of the B-matrix,

$$\mathbf{B}^{(\alpha,l)} = \mathbf{C}_u^T \cdot \mathbf{C}_u, \tag{E4}$$

where $\mathbf{C}_u$ is an upper triangular matrix. Thus the actual algorithm we used for formulating the constrained minimisation of the
cost function is obtained by replacing in the preceding formulas all incidences of the matrix $\mathbf{B}^{1/2}$ with the matrix $\mathbf{C}_u^T$ (and,
similarly, by replacing the inverse matrix $\mathbf{B}^{-1/2}$ by the inverse of the Cholesky factor, $\mathbf{C}_u^{-T}$).

---

[12]For those readers interested in spectral formulations of 3DVAR we refer to Eqs. (28)–(30) in Kahnert (2008). Expressed by the spectral control vector $\boldsymbol{\chi} = \mathbf{U} \cdot \delta\boldsymbol{x}$, the weak constraint in the cost function takes the spectral form $J_G = \frac{1}{2} \boldsymbol{\chi}^\dagger \cdot \mathbf{U}^{-\dagger} \cdot \mathbf{G}^T \cdot \mathbf{B}_G^{-1} \cdot \mathbf{G} \cdot \mathbf{U}^{-1} \cdot \boldsymbol{\chi}$, and its contribution to the gradient of the cost function becomes $\nabla_{\boldsymbol{\chi}} J_G = \mathbf{U}^{-\dagger} \cdot \mathbf{G}^T \cdot \mathbf{B}_G^{-1} \cdot \mathbf{G} \cdot \mathbf{U}^{-1} \cdot \boldsymbol{\chi}$. We see that these expressions involve the computation of the variable $\delta\boldsymbol{x} = \mathbf{U}^{-1} \cdot \boldsymbol{\chi}$ in physical space. Thus, even when using a spectral formulation of the 3DVAR method, one can still compute the constraints in physical space and add their contributions to $J$ and $\nabla J$. The advantage of this is, as explained above, that the SVD of the observability matrix can be computed in the reduced subspace, which substantially reduces the dimension of the numerical SVD problem.

[13]The Cholesky decomposition is, essentially, a special case of a LU-decomposition, which applies to symmetric real (or Hermitian complex), positive definite matrices.

*Author contributions.* MK worked with the theoretical developments and and numerical implementation, EA performed the testing of the
method.
*Acknowledgements.* This work was funded by the Swedish National Space Board through project nos. 100/16 (MK) and 101/13 (EA).

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
