# Peer review of "How much information do extinction and backscattering measurements contain about the chemical composition of atmospheric aerosol?"

_Atmospheric Chemistry and Physics, 2016_

## Referee Comment (RC1) · Anonymous Referee #4 · 11 Nov 2016

**Summary**

The authors consider the case of assimilation of remote-sensing data (specifically aerosol extinction and backscattering coefficients) applied to aerosols fields within a chemical transport model. They describe how an additional term can be added to the 3D-var cost-function so that the assimilation adjusts only those components (in a transformed space) for which the observations provide information. The additional term relies on the singular value decomposition of the scaled observation operator. In this way, the assimilation automates the choice of control variables in an otherwise highly

under-constrained inverse problem.

**Verdict**

The paper is very well written and is surprisingly clear, given the subject matter. The manuscript introduces a potentially very powerful concept for variable selection into the field of aerosol data assimilation. The authors have probed the idea in a minimal test case, which assists in understanding the effects. I found that the shortcomings of the paper were relatively minor. I felt there was insufficient discussion of the literature of related treatment. I was unsure about whether the organisation of the material was optimal (see the "Main comments"). Finally, a counter-experiment without the addition of the new constraint in the 4D-var cost function was, in my opinion, lacking. All in all, I believe that the paper should be published, pending the minor revisions suggested below.

**Main comments**

- There was little or no discussion of literature on related treatments. I have not the time to read all of these myself, however I have included a list at the end of articles that may be relevant, for example those that deal with information content of observations in data assimilation or those that refer to the singular value decomposition of the observability matrix.

- I believe that a small counter-experiment was lacking. In the results presented in section 3.2, I would suggest also presenting results for the assimilation experiment which did *not* include the additional constraint in the 3D-var cost-function.

- I was unsure whether the organisation of the material was optimal - I highlight this as an issue that the editor may wish to take up. The introduction concludes by urging the reader to read the Appendix before proceeding onto the rest of the methods and results section. Much of the interesting methodology is contained within the Appendix, and we agree that it would be difficult to make sense of the main part of the paper without a good understanding of the contents of the Appendix. As such, I would suggest incorporating the Appendix into the main body of the text. At one level, this is really a matter of taste, and thus I leave it to the editor.

**Minor comments**

- When describing observation errors, there was no reference to the component from "representativity errors" (i.e. measurements are made at a point, or over a small area in the case of remote sensing, while model grid-boxes are typically in the order of kilometres across in the horizontal dimensions). All of the discussion about observation errors was in terms of the measurement error and errors in the observation operator, both of which are relavent. However the representativity component is not insignificant in many contexts.

- Observation standard deviation was reported in percentage, but it was unclear what this was a percentage of. Please clarify.

- I would suggest replacing all instances of the term "costfunction" with "cost function" (or "cost-function"). The latter is about 15 times more common (on the web, at least). Similarly, I believe that the compound word "nullmatrix" is used in German (capitalised, that is) whereas it is "null matrix" (or "zero matrix") in English.

- I could not find a definition to the term "signal degrees of freedom". Please include this somewhere (preferably at first usage, or in the Appendix).

- Line 48: Please replace "This is a rather bold approach that largely disregards ..." with "This is approach largely disregards ..." – please use argument rather than rhetoric to explain what is wrong with the work of others.

- Line 54: The reference to Kahnert (2009) is used to show that several optical properties at multiple wavelengths may allow constraining more than just the total mass concentration. Surely other authors have looked into this. If so, please summarise other work done. If not, please say so.

- Line 98: "... using 40 eta-layers with variable thickness depending on the underlying topography" – do you just mean that this is a terrain-following coordinate? Or is there something more sophisticated about this?

- Line 125: "The background error covariance matrix of the model a priori is modelled with the NMC method ...". I checked the reference (Kahnert, 2008), in which I believe this is described. If the implementation is the same here as in the 2008 article, then I believe that it is best to say that it "follows similar principles to the NMC method" or "is inspired by the NMC method". If it is indeed the NMC method, the authors should clarify the difference to methodology laid out in Kahnert (2008).

- Line 129: I would suggest replacing "Given m observations of, e.g., $m_1$ different parameters at $m_2$ different wavelengths, so that $m_1 \cdot m_2 = m$, how many..." with "Given m observations (e.g., $m_1$ different parameters at $m_2$ different wavelengths, so that $m_1 \cdot m_2 = m$), how many..."

- Line 130: "... we can constrain to better than observation error" – do you mean "model error"? If not, please explain that the transformation makes the (rescaled) observation errors and (rescaled) model variables comparable.

[Figure]

- Line 134: Please replace "... a singular value decomposition of the Jacobian of the observation operator ..." with "... a singular value decomposition of the Jacobian of the scaled observation operator ..." or something similar. By the way, this scaled observation operator appears to have a name: "the observability matrix"

- Footnote 2, page 5: I found this distinction a bit cryptic. Please consider rephrasing.

- Line 150: I realise that this is something that is clarified later on, but I would suggest saying a few words at this point about the synthetic observations; namely, what kind of observations they were and how many observation points there were.

- Line 154: I would suggest the following change "thus providing nearly perfect observations. (We assumed an observation error standard deviation of 10 %) The only ..." becomes "thus providing nearly perfect observations (we assumed an observation error standard deviation of 10 %). The only...". See also my comment about about describing the units for the observation error standard deviation.

- Line 162: What is "Nd:YAG"? Please clarify. I suspect that this is some error with the bibliography manager.

- Line 168: I would suggest the following change: "... two wavelengths. (Compare, e.g., cases 1., 2., and 3. to cases 4., 5., and 6.) Hence ..." becomes "... two wavelengths (compare, e.g., cases 1., 2., and 3. to cases 4., 5., and 6.). Hence ..."

- Line 171: a missing full stop after the right parenthesis.

- Table 1, caption: the "Nd:YAG" term appears again.

- Line 181: I believe that "weak constrains" should be "weak constraints"

- Line 189: See my comment above about the representativity component to the observation error

- Figure 1: I think it would be interesting to see the increment as an additional panel in this figure

- Figure 1: The text on the scale is a bit too small. I would suggest having one scale, rather than three, and enlarging the scale so that the labels can be read.

- Figure 2: The units appear to be "mixing ratio [ppb-m]". Do you mean mass mixing ratio? Please clarify.

- Figure 4: Do we need all panels? Why not just show the first three or four, and then a selection of the remaining terms.

- Line 262: I would suggest the following change " ... dramatic decrease in both the entropy and signal degrees of freedom ..." becomes " ... dramatic decrease in both the entropy-change and signal degrees of freedom ..."

- Line 282: "It also appeared that among the original model variables, secondary inorganic aerosol components were most faithfully retrieved by the inverse modelling solution" – why is this? why SIA? Do they have specific optical properties to make them more observable by such LIDAR pseudo-observations?

- Line 293: I would suggest the following change: "The present study should be extended..." becomes "The present study could be extended..."

- Line 295: I believe that the expression "highly underrated" is somewhat dramatic and relatively colloquial, and does not fit with the tone in the rest of the paper. The authors are encouraged to use argument rather than rhetoric to make their point.

- Line 297: Regarding the statement "There is little one can put forward in defence of this model other than pure convenience". Some justification is required (e.g. some references) to demonstrate why this model is untenable. There's a saying (attributed to George Box) "All models are wrong, some models are useful". Does this model give significantly worse results than representations, or is it just inaccurate in its assumptions?

- Paragraph beginning at line 307: It may be worth making it clear that $y$ is not observed, but a model equivalent of the observations

- Lines 324 and 326: I would suggest replacing all instances of "3-dimensional" with "three-dimensional"

- Paragraph beginning 336: I would suggest mentioning that the assumption of unbiased background and observation errors

- Footnote 6, page 15: See my comments above about the representativity component of the observation error.

- Footnote 7, page 16: I would suggest the following change: "The observation errors are often uncorrelated" becomes "The observation errors are often assumed to be uncorrelated (this is not always true)"

- Paragraph beginning at line 368: Please comment on the role of spatial and inter-species correlations, particularly in light of the comment "if we allow all model variables to be freely adjusted" (line 374).

- Line 369: It might be worth noting that $\delta x$ is not constrained to ensure that all components of $x$ remain positive in the analysis.

- Line 386: The phrase "rather tricky" strikes me as somewhat colloquial. I would suggest the following change: "However, to actually make such a comparison is

rather tricky" becomes "However, to actually make such a comparison poses two problems."

- Paragraph beginning at line 390: Please introduce the meaning of the angle-bracket notation. I believe that this is common in physics, but other disciplines (e.g. statistics) often use different notation for the expectation.

- Footnote 8, page 17: Should $A \cdot A = B$ be $A^T \cdot A = B$?

- Line 425: Should "(C7)-(C9)" not be "(C6)-(C9)"? As far as I can see, Eq. (C6) is required here.

- Line 434-435: Please state which particular sections/chapters of Rodger (2000) the reader is referred to.

- Equations C12, C15: I would suggest showing the range of the summation to indicate that it is a summation over observations (i.e. $i$ ranges from 1 to m)

- Line 479: "Naively, one may have expected that the dimension would, on the contrary, be reduced to $n - k$" – why? is this because the number of unknowns remains the same but the number of equations to be solved has increased by $k$?

- Line 486: I would suggest the following change: "(Note that the covariance matrices and their inverses are symmetric, i.e., $R^T = R$, etc.)" becomes "Note that the covariance matrices and their inverses are symmetric (i.e. $R^T = R$, etc.)."

- Appendix: For all unit and zero matrices (and vectors), I would suggest indicating the dimension as a sub-script

- Line 498: I would suggest adding a subscript to clarify with respect to what the differentiation refers (i.e. replace $\nabla$ with $\nabla_\xi$)

- Paragraph beginning line 515: how was this tuning done in practice?

- Line 549: "It turns out that Eq. (D18) gives a relatively sharp transition from unconstrained to constrained model variables, while Eq. (D19) gives a very gentle transition" – this can be seen from the equations. I would suggest replacing the sentence with "It can be seen that Eq. (D18) gives a relatively sharp transition from unconstrained to constrained model variables, while Eq. (D19) gives a very gentle transition"

- Paragraph beginning line 567: I found that this went too fast and skipped a bit too much detail, after what was otherwise a very well-written paper that included a fair bit of theory. In particular, can you please explain in further detail the reduced matrices. The phrase "we are primarily interested in constraining the chemical components" was surprising, since I thought the authors were mainly interested in the aerosol components. What does it mean to "restrict ourselves to the chemical subspace"?

- Line 570: Full stop missing after $N_c$

- Paragraph beginning 574: similar to the above comment, I found that this skipped over too much detail. Please add further explanation. The authors state that in their present study, they use a Cholesky decomposition of the B-matrix. Is this what was used in Kahnert (2008), or is this described as the "spectral formulation"? If it is different, it may be relevant to understand why the Cholesky decomposition was preferable to the author's previously presented methodology. This is mainly to understand the requirements and limitations of the proposed methodology.

**Minor formatting issues**

- References with parentheses inside parentheses: lines 33, 268, 269, 376

- Some of the in-line equations appeared to be missing spaces on one or both sides of the equals sign – this only appeared in the appendix. See lines: 391, 403, 404, 425, 515. I might just be imagining it. The paper was otherwise very well laid out.

**References the authors may wish to consider**

- Qin, X. Measuring information content from observations for data assimilation: relative entropy versus shannon entropy difference. Tellus: Series A. 59, 2, 198-209, 2007.

- J. Joiner, A. M. da Silva. Efficient methods to assimilate remotely sensed data based on information content. Q. J. R. Meteorol, SOC. (1998), 124, pp. 1669-1694

- C Cardinali, S Pezzulli, E Andersson. Influence-matrix diagnostic of a data assimilation system. Q. J. R. Meteorol. Soc. (2004), 130, pp. 2767-2786. doi: 10.1256/qj.03.205

- C. Johnson, N. K. Nichols; B. J. Hoskins. Very large inverse problems in atmosphere and ocean modelling. Int. J. Numer. Meth. Fluids 2005; 47:759-771.

- M Bocquet, 2009: Toward Optimal Choices of Control Space Representation for Geophysical Data Assimilation. Mon. Wea. Rev., 137, 2331-2348, doi: 10.1175/2009MWR2789.1.

- F Rabier, N Fourrie, D Chafai, P Prunet. Channel selection methods for Infrared Atmospheric Sounding Interferometer radiances. Q. J. R. Meteorol. Soc. (2002), 128, pp. 1011-1027

- C Johnson, B. J. Hoskins, N. K. Nichols. A singular vector perspective of 4D-Var: Filtering and interpolation. Q. J. R. Meteorol. Soc. (2005), 131, pp. 1-19 doi: 10.1256/qj.03.231

---

## Referee Comment (RC2) · Anonymous Referee #1 · 17 Nov 2016

The ACPD paper by Kahnert and Andersson deals with the assimilation of lidar observations into a chemical transport model. They investigate how much information about the chemical composition can be extracted from backscatter and extinction measurements and how this information is best assimilated into a chemical transport model.

Overall the paper is very well written and should be published as it is an interesting and important contribution to aerosol research. I only have a few minor comments which the authors may consider for their final paper. I have to say that my experience lies more on the lidar and aerosol optics side than on the information theory / mathematical

side, thus I was not really able to review all theory details described in the appendix.

Comments:

It may be beneficial to say a few words about the refractive index and the size bins of the individual species of MATCH. I suggest to add a table with the refractive index of these species at the lidar wavelengths.

Line 105-109: The description of the MATCH aerosol microphysics module could be shortend as it is not used in this paper.

Line 118: What about the emissions of the other species? Are they also from EMEP?

Line 134: "an" -> "and"

Line 147 "we constrain to better than observation error": It is not clear to me what this means.

Line 151: Remove "the".

Line 177: "To be specific" could be removed.

Line 177: Do the results (N_s and H) presented in this section depend on the order of the parameters? If yes, are the changes significant?

Line 185: "around 7.4 for a single wavelength to around 10-12 for two wavelengths" would be more precise.

Line 203: I was not aware about the difference between "observation error" and "measurement error". Is this generally accepted terminology? Maybe you can add a reference here so that the reader not familiar with this terminology can see that is used also elsewhere or was introduced by someone (maybe Rodgers?).

Fig. 1: The difference between the middle and the right subplot is hardly visible. Perhaps you find a better way to visualize it.

Line 229 (and at other places): You use beta_sca and beta_bak for the backscatter

coefficient. Please use only a single symbol throughout the paper.

Line 241 "the secondary inorganic aerosol (SIA) species are almost completely restored by the 3DVAR": Is it understandable why exactly SIA is restored? Because of the refractive index? Or does it have something to do with the order (index number) of the species in the model?

Line 274 "there appeared ...": This was not really shown in the paper, so you might remove this sentence or write it in a different way.

Fig. 3: In this figure the difference between "observations" and "analysis" is much smaller than 10 % (the assumed "observation error"). As this is somewhat unexpected (but understandable as an optimization is applied) you may add a brief discussion about the effect a "measurement error" (noise) would have. Because of the assumed linearity this probably is not very difficult to explain.

Fig. 4: Could it be of interest to see which aerosol species (size bins) the individual variables represent? What would be the effect of changing the order of the species?

Line 277: "to be sure" could be removed.

Line 314: I think some aerosol species exist for which assuming externally mixed spheres is not that wrong.

---

## Referee Comment (RC3) · Anonymous Referee #2 · 2 Dec 2016

General comments

This paper details an interesting way to assess the information content in lidar measurements of aerosol backscatter and extinction with respect to model assimilation. It also demonstrates how this knowledge may be used to optimize the incorporation of lidar measurements in the model. This is a very interesting and relevant topic. Assimilation of lidar data into models is a field that is still developing rapidly, with a few different groups using very different techniques; therefore, well designed research into how best to use lidar data is very valuable. It is also potentially informative to the lidar commu-

nity, since work must begin soon to design the next satellite lidar instruments if the lidar record is to continue. The choice of which measurements and which wavelengths to include has a large bearing on cost and technological difficulty, so having quantitative information about which measurements are most useful for improving models is critical. To that end, I would like to suggest some additional cases for Table 1, please see the specific comments below.

The paper is well written with very nice clarity. However, the overall organization is somewhat difficult. The current organization consists of a very streamlined and easy-to-read main text with five very technically dense appendices. While the main text is pleasantly easy to read on the first pass, there is too much information missing. While it's appropriate to include extra, more detailed information in appendices, the main text still needs to be able to stand on its own, and in my opinion, it doesn't quite. I would suggest that the main equations and brief explanations should also be included in the main text, including all the equations that a reader would need to apply to calculate the kinds of results presented in this work. The appendices also include a lot of pedagogical development; this is the kind of information that I think rightly belongs in the appendix for readers who want more details. Since the appendices are 5 different topics, I also suggest that each appendix should be exist as a separate entity, with all variables defined, so that a reader can read Appendix D to learn about the application of constraints or Appendix E for the "practical aspects" without a close reading of Appendix A,B, and C, to find the definitions of the variables.

The results and conclusions are also a little too abbreviated. Some key aspects are missing, like how was the specific weighting chosen and how do we know this is the best weighting? Also, as pointed out by another reviewer, the assessment (section 3.2) is really more of a demonstration. That is, although the theoretical development is compelling, the application/assessment section isn't sufficient to convince readers that this is a better way to assimilate lidar data than another way. This paper clearly reflects a lot of research on the part of the authors and I think the missing information probably exists but was left out in the effort to streamline the manuscript. I think adding this additional information should be fairly straightforward and would improve the usefulness of this research for the modeling and lidar communities without adding too much complication to the nice flow of the paper.

Specific comments

Lines 151-158: Here is an example where I think some important things are missing from the main text which only appear in the appendices. These eight short lines are the methodology section for the key calculations that are the novel part of your research and are critically important for a reader to understand. I suggest that a way to decide what should also be included here would be to target the subset of equations that a reader would need to apply to calculate results like yours, but without their derivations. Also include enough supporting explanation to describe what the equations say and how to use them.

L152-153: Specifically here, Eq C6 and C?16 should be included in the text, since they are required to understand the meaning of the sentence. Later, at L159-160 where readers are directed to the appendix for more background information, I think that's fine.

L155-157: The equations for signal degrees of freedom and Shannon information content should also be included in the text.

L165: "a numerical experiment". In fact, it's more of a demonstration than an experiment. It's useful as a demonstration of the results of the technique, but there's nothing in the demonstration that addresses a hypotheses. Sharing more of the background work would make the paper more compelling. For example, as another reviewer suggested, comparing to a control experiment would be necessary for convincing readers that this technique is useful. For another example, a pair of runs with different weightings in the assimilation would help answer the question of why the weighting that was ultimately chosen was the best one.

L177: Depolarization is not included in the studied parameters, yet lidar studies have shown that depolarization measurements contain some information about aerosol composition (for example, Omar et al. 2009 as referenced in the introduction, but there are many others). Do the authors have any comment on depolarization and why it isn't included in this study?

Table 1 and related discussion: From a lidar standpoint, some combinations of channels are more technologically affordable than others, so the discussion of which channels add significant information content is very interesting. However, the utility for the lidar community would be maximized if the combinations were ordered such that they roughly increase in technological difficulty. Also, some combinations don't really make sense from a technological standpoint. There is no lidar that measures extinction but not backscatter at the same channel (although modelers may use only the extinction). On the other hand, backscatter (actually attenuated backscatter) without a direct measurement of extinction is common. Also, since CALIPSO, CATS, EarthCARE and the $3\beta+2\alpha$ combination of airborne HSRL2 are mentioned in the introduction and motivation sections, it would be useful if the combinations relevant to those instruments were included. CALIPSO = CATS = $\beta(\lambda1) + \beta(\lambda2)$. EarthCARE = $\beta(\lambda3) + k(\lambda3)$. HSRL2 = $\beta(\lambda1) + \beta(\lambda2) + \beta(\lambda3) + k(\lambda2) + k(\lambda3)$. I would suggest these combinations of backscatter and extinction would be most interesting and useful to the lidar community:

$\beta(\lambda3)$

$\beta(\lambda1) + \beta(\lambda2)$

$\beta(\lambda1) + \beta(\lambda2) + \beta(\lambda3)$

$\beta(\lambda3) + k(\lambda3)$

$\beta(\lambda1) + \beta(\lambda2) + k(\lambda2)$

$\beta(\lambda1) + \beta(\lambda2) + \beta(\lambda3) + k(\lambda2) + k(\lambda3)$

For these experiments, it appears that the observation error was always assumed to

be the same in every channel. I think it's a reasonable assumption, to first approxima-tion, that the _measurement error_ would be similar in every channel, but as pointed out at L78-79, some lidar retrievals include additional non-random errors that can be much larger. This could and should affect the choice of channels to assimilate. For example, the Raman, HSRL, and transmittance techniques are fairly direct measures of extinction, but techniques that require an inferred lidar ratio to convert backscat-ter to extinction have relatively little additional measurement information content in the extinction.

L 197-201. Here also the discussion of incorporating soft constraints and the specifics of the three weighting schemes should be in the main text of the paper and not just the appendix, since it is discussed here in the results section. This section is not understandable without the equations from the appendix and most of section D3.

L 203-204. Discussion of observation error vs. measurement error. This is interesting and useful, but could be clarified as to whether the forward model error (due to poor as-sumptions) is considered part of the observation error or is another separate source of error. If it is part of the observation error, how are the forward model errors represented and how are they transformed into the space of the measurement vector?

L 207 While there may be retrieval errors in the lidar backscatter and extinction due to assumptions, assumptions on particle shape and size distribution are not among the assumptions used in lidar retrievals. These examples belong only to the optics model (forward model). So, perhaps delete "also".

Poor assumptions in the optics model or in lidar retrievals would presumably lead to bias errors, whereas measurement errors would more typically be random. Does this make a difference in the analysis?

L 219. I strongly agree that estimating the uncertainties in the optics model is very important. Some discussion here seems warranted about how that can be done. Later I see that this is discussed in the summary (L281 – 292) but I think it would be better if

it comes up first here in the discussion section.

L 256 and caption to Fig 4. In both places, it would be kind to remind readers that the delta notation in $\delta x'$ means this is the difference between the value and the background value.

L 259-263. The choice of D21 with its sharp dropoff in weighting appears to mean that only one transformed variable is allowed to change in a meaningful way, although the measurement scenario chosen has nearly the maximum amount of information content available, close to DOF=4. Why was D21 chosen instead of D18, which would allow the measurements to play a bigger role? The only discussion of this choice is the rather vague comment in the Appendix "it is a matter of experience to test different approaches and select the one that proves to be most suited". How and why was this approach determined to be the most suited?

Comparison of Figure 3 and Figure 2, if I understand right, underscores the fact that there is a significant null space, not controlled by the measurements, since essentially the same measurements in Fig 3 correspond to both the black and red lines in Fig 2. What is not clear to me is what happens in a standard assimilation to the variables that are not well controlled by the measurements? Do they remain close to the background values, or do they vary wildly and arbitrarily? If the former, then the exercise of determining the singular values wouldn't help the assimilation very much (but would still be useful in terms of building knowledge about what we can and can't actually measure). On the other hand, if a standard assimilation arbitrarily varies state variables in the null space, then this is a very important motivation for this technique (and maybe that motivation could be emphasized a little bit more in the introduction and conclusions). Not being very familiar with the field of model assimilation, I guess but don't actually know that there must be other "regularization" techniques in use to prevent an assimilation from arbitrarily varying parameters that are mostly in the null space of the observations, although I imagine existing techniques may be more ad hoc than the method presented here. Can you comment on other methods and demonstrate how this method performs

better than other methods?

L 298-299. "It also appeared". This result is disappointingly empirical for such a well-founded theoretical study. This observation that SIC was most faithfully retrieved was made in a single case– would you expect this result to be general for all cases, and why? Answering the question is complicated since the singular variables are defined only in the transformed space and therefore the information about what variables are or are not constrained by the measurements is only in this transformed space, not the state space. Yet this statement highlights that it's desirable to have information about which chemical species and size bins are constrained by the measurements. Is there any way to provide information about this quantitatively? For example, since each state variable is a linear combination of the transformed variables, would showing the linear coefficients in a table make it more obvious which state variables are most closely related to the most significant transformed variables? Perhaps there is a way to use the coefficients to calculate a "fractional significance" that would indicate that x% of the variability in a given state parameter is orthogonal with significant transformed variables while (1-x)% is orthogonal with insignificant variables?

Minor comments

L37: Muller et al. 1999 and Veselovskii et al. 2002 and related papers (there are many) would be more relevant references here since they detail retrievals of refractive index, etc., from lidar. (Mishchenko et al. 2007 is an introduction to the Glory satellite and was about retrievals from a polarimeter.)

Müller, D., U. Wandinger, and A. Ansmann (1999), Microphysical particle parameters from extinction and backscatter lidar data by inversion with regularization: theory, Appl Optics, 38(12), 2346-2357, doi: 10.1364/AO.38.002346.

Veselovskii, I., A. Kolgotin, V. Griaznov, D. Müller, U. Wandinger, and D. N. Whiteman (2002), Inversion with regularization for the retrieval of tropospheric aerosol parameters from multiwavelength lidar sounding, Appl Optics, 41(18), 3685-3699, doi:

10.1364/AO.41.003685.

L99: I infer that the ratios in the different size bins are fixed, or else there would be much more than 20 total variables. Is there a way to concisely clarify this in the sentence?

L109: maybe replace "in the present setup" with "currently in that version". "The present setup" seems to refer to "the setup used in the present study" but that is misleading, since the present study uses the 20-variable version of the model.

L134: "an" should be "and"

L142: "Error correlations . . . are not assumed to be separable". I'm not sure what this means. What is (or is not) separable from what?

L153: "see Eq. D16". Should this be C16?

L162-164: Should this sentence perhaps be part of section 2.4, as part of the description of the new technique? The rest of this paragraph (L164-174) is more about the demonstration of the new technique and so seems like a somewhat distinct topic.

Figure 1: The caption says "note the nonlinear colour scale" Actually, the scale is hardly visible. Please expand the axis labels so they are a similar text size to the caption text.

Figure 2: The axis labels' and inset box labels' font size should also be increased here.

L 391. The variable n is not defined. Possibly this is the only case, but I would also request that variables be re-defined frequently when used in key equations. If a reader is directed from another part of the paper to Equation D18 or C12, for example, then it would be nice if all the information relevant to understanding that equation is given immediately after that equation, rather than having to scroll through 8 or 10 pages to relocate the definitions of key variables.

L563. The symbol lambda is used for wavelength elsewhere in the text. You might consider using a different symbol here.

---

## Referee Comment (RC4) · Anonymous Referee #3 · 3 Dec 2016

Review to manuscript of Michael Kahnert and Emma Andersson

It is well known, that the problem of inversion of standard 3β+2α lidar measurements to the particle microphysics is undetermined and to constrain it, numerous techniques were considered. The authors suggest an interesting approach to assimilation of lidar measurements into chemical transport model. It looks like a promising concept to extract the information about particle parameters from lidar measurements.

Paper is very well written and should be published. The structure may be questionable, because a half of material is put in appendices. These appendices are clearly written and are definitely useful for unprepared reader. I personally, had no problems with material structure. Additional references to the previous studies of lidar data inversion would be desirable, and other Referees have already suggested several.

Stability of retrieval strongly depends on aerosol type. It is more challenging for aerosols with dominant coarse mode and for particles with strong absorption. The authors consider only one example (not the most challenging) in their simulation, so it is not very clear how the approach will work for other aerosol types. But this may be a subject of separate study.

The Referees have already suggested numerous improvements and corrections so I have not much to add.

---

## Short Comment (SC1) · 3 Dec 2016

May be you can check the papers of Wang at al. in ACP (2013, 2014a and b), where lidar assimilation is tested.

---

## Referee Comment (RC5) · Anonymous Referee #5 · 8 Dec 2016

Summary

The inversion of aerosol optical properties into the aerosol chemical composition is a ill posed problem. The authors use information theory techniques to estimate the amount of information contained in LIDAR observations. They present different methods to make use of it as contains in a 3DVAR algorithm. This is meant to avoid assimilating noise inherent to the observations. To evaluate their constrain methods, they create synthetic observations from CTM simulations and assimilate them back into the CTM.

Recommendation

[Figure]

The paper is well written and should be published. The methodology proposed is novel and can be applied to different observations within the variational assimilation framework.

Main comments

The authors choose to place all equations and their derivations into different appendixes. This hindered slightly the reading of sections 2.4, 3.1 and 3.2. However, the overall readability of the manuscript is improved by the focus on the description and evaluation of the method in the main text.

Minor comments

Figure 1 is hard to read, specially the colour bar. Otherwise, the previous Referees have a number of valid suggestions for improvement, and I have nothing to add.

---

## Author Comment (AC1) · 2 Feb 2017

Below the reviewer comments are marked in blue, our response is marked in black.

The ACPD paper by Kahnert and Andersson deals with the assimilation of lidar observations into a chemical transport model. They investigate how much information about the chemical composition can be extracted from backscatter and extinction measurements and how this information is best assimilated into a chemical transport model.

Overall the paper is very well written and should be published as it is an interesting and important contribution to aerosol research. I only have a few minor comments which

the authors may consider for their final paper. I have to say that my experience lies more on the lidar and aerosol optics side than on the information theory / mathematical side, thus I was not really able to review all theory details described in the appendix.

We very much appreciate receiving comments from the lidar and aerosol optics community. The parts that deal with theoretical developments and chemical data assimilation have been very well covered by reviewers 2 and 4. We thank the reviewer for his supportive review and helpful comments!

Comments:

1. It may be beneficial to say a few words about the refractive index and the size bins of the individual species of MATCH. I suggest to add a table with the refractive index of these species at the lidar wavelengths.
   We will add a table providing the refractive indices, and an itemised list of the size bins and the corresponding size ranges.

2. Line 105-109: The description of the MATCH aerosol microphysics module could be shortened as it is not used in this paper.
   It is difficult to shorten these 5 lines. We could only remove them. Then again, we would like the reader to understand that there do exist more realistic optics models, but they are not so straight forward to test, owing to their nonlinearity. Thus the present study is meant as a first step in a larger project, which we will, hopefully, be able to follow up with an investigation of information content based on a more sophisticated description of aerosol optics.

3. Line 118: What about the emissions of the other species? Are they also from EMEP?
   Yes. But EMEP does not deliver gridded emission for black carbon and elemental carbon, only for total primary particulate matter. The sentence in question was

meant to explain how we converted these into gridded emission data for black carbon and elemental carbon. We will reformulate this to make it less ambiguous.

4. Line 134: "an" -> "and"
   Yes.

5. Line 147 "we constrain to better than observation error": It is not clear to me what this means.
   This formulation was also criticised by another reviewer. We will reformulate this part as follows: "Suppose we have an $n$ dimensional model space. Given $m$ observations (e.g., $m_1$ different parameters at $m_2$ different wavelengths, so that $m_1 \cdot m_2 = m$), how many independent model variables $N \leq n$ can we constrain with the observations?"

6. Line 151: Remove "the".
   OK. However, we will reformulate this entire section to accommodate the comments by reviewer 2.

7. Line 177: "To be specific" could be removed.
   Agreed.

8. Line 177: Do the results ($N_s$ and $H$) presented in this section depend on the order of the parameters? If yes, are the changes significant?
   We do not quite understand this question, especially not what the reviewer means by "order". Is the reviewer inquiring about the *ordering* and *grouping*, or about the *magnitude*? In the latter case, the answer is no, because $N_s$ and $H$ are computed from the scaled Jacobian of the observation operator, which does not depend on the magnitude of the parameters. In the former case, the results do depend on which parameters are being measured, but, of course, not on the ordering.

9. Line 185: "around 7.4 for a single wavelength to around 10-12 for two wavelengths" would be more precise.

OK. This text is likely to change significantly in the revised version, owing to comments by reviewer 2, who asked us to consider different and technically more realistic combinations of observables in table 1.

10. Line 203: I was not aware about the difference between "observation error" and "measurement error". Is this generally accepted terminology? Maybe you can add a reference here so that the reader not familiar with this terminology can see that is used also elsewhere or was introduced by someone (maybe Rodgers?).
    We will add the formal definition for the observation error as $\epsilon_o = \epsilon_f + \epsilon_m$ and a reference to Rabier et al. (2002). They use the same terminology as we do, and they also denote the forward model error by $\epsilon_f$. However, they use the symbol $\epsilon_o$ for the measurement error, which is potentially confusing. We find it less confusing to denote the measurement error by $\epsilon_m$, and to reserve the symbol $\epsilon_o$ for the observation error. We will also mention that there can be other contributions to the observation error, such as representativity error. These concepts are well understood both in the data assimilation and in the satellite remote sensing/retrieval community, but not necessarily among instrument developers, who tend to identify $\epsilon_o$ with $\epsilon_m$, while forgetting about $\epsilon_f$. This can be a serious mistake in cases where $\epsilon_f \gg \epsilon_m$, as is the case, e.g., in lidar depolarisation measurements. We find this point sufficiently significant to repeat it, in rephrased form, in appendix B.

11. Fig. 1: The difference between the middle and the right sub-plot is hardly visible. Perhaps you find a better way to visualize it.
    We will remove this figure. The regional model is merely used to generate a test case, but we do not address questions of regional modelling or horizontal information spreading in 3DVAR. Therefore this figure conveys no useful information for this study.

12. Line 229 (and at other places): You use $\beta_{\text{sca}}$ and $\beta_{\text{bak}}$ for the backscatter coeffi-

cient. Please use only a single symbol throughout the paper.
Yes, we will correct this and consistently use $\beta_{\text{sca}}$.

13. Line 241 "the secondary inorganic aerosol (SIA) species are almost completely restored by the 3DVAR": Is it understandable why exactly SIA is restored? Because of the refractive index? Or does it have something to do with the order (index number) of the species in the model?
This question has also been brought up by other reviewers. We will add a figure in which we show the linear coefficients in the transformation of the control variables in Eq. (C16). Based on this extra figure we will discuss which aerosol components in model space make the dominant contribution to the signal-related variables in the transformed space. This will facilitate the interpretation of the analysis results.

14. Line 274 "there appeared ...": This was not really shown in the paper, so you might remove this sentence or write it in a different way.
OK, we will strike this sentence.

15. Fig. 3: In this figure the difference between "observations" and "analysis" is much smaller than 10 % (the assumed "observation error"). As this is somewhat unexpected (but understandable as an optimization is applied) you may add a brief discussion about the effect a "measurement error" (noise) would have. Because of the assumed linearity this probably is not very difficult to explain.
OK, we will add the following text. "In fact, the difference between the observation-equivalent analysis and the observations deviate by even less than 10 %. However, our tests confirmed that an increase in the observation error eventually results in analysis results of which the observation-equivalent increasingly deviates from the observations (not shown)."

16. Fig. 4: Could it be of interest to see which aerosol species (size bins) the individual variables represent? What would be the effect of changing the order of the

species?

We will add an extra figure that will show at least a selection of aerosol species in specific size bins. In response to reviewer 2 and 4, we will even show a comparison with an unconstrained analysis. This will likely make it clearer that the constrained analysis reduces the noisiness of the analysis, since it is being constrained to assimilating signal rather than measurement noise.

We do not understand the last question about changing the "order of the species".

17. Line 277: "to be sure" could be removed.
    Agreed.

18. Line 314: I think some aerosol species exist for which assuming externally mixed spheres is not that wrong.
    It is unclear what kind of species the reviewer refers to. Certainly not dust or black carbon (BC). Sea salt is either mixed with water, or else it is nonspherical. Organic carbon (OC) and secondary inorganic aerosols (SIA) are rarely found in pure form. They are often mixed with each other, with water, NaCl, and even BC and dust. Even nucleation-mode particles are often the result of at least binary nucleation involving more than one species. In our more realistic aerosol microphysics model there is not a single size bin in which liquid-phase (i.e., spherical) aerosols consist of a single compound. We therefore prefer to keep the text in its present form.

**References**

Rabier, F., Fourrié, N., Chafaï, D., and Prunet, P.: Channel selection methods for infrared atmospheric sounding interferometer radiances, Q. J. R. Meteorol. Soc., 128, 1011–1027, 2002.

---

## Author Comment (AC2) · 2 Feb 2017

Below the reviewer comments are marked in blue, our response is marked in black.

**1   General comments**

1. This paper details an interesting way to assess the information content in lidar measurements of aerosol backscatter and extinction with respect to model assimulation. It also demonstrates how this knowledge may be used to optimize the incorporation of lidar measurements in the model. This is a very interesting and relevant topic. Assimilation of lidar data into models is a field that is still developing rapidly, with a few different groups using very different techniques; therefore, well designed research into how best to use lidar data is very valuable. It is also potentially informative to the lidar community, since work must begin soon to design the next satellite lidar instruments if the lidar record is to continue. The choice of which measurements and which wavelengths to include has a large bearing on cost and technological difficulty, so having quantitative information about which measurements are most useful for improving models is critical. To that end, I would like to suggest some additional cases for Table 1, please see the specific comments below.

We thank the reviewer for the considerably thorough and supportive review, which will help us to improve various aspects of the manuscript. Our detailed response to the review comments follows.

2. The paper is well written with very nice clarity. However, the overall organization is somewhat difficult. The current organization consists of a very streamlined and easy-to-read main text with five very technically dense appendices. While the main text is pleasantly easy to read on the first pass, there is too much information missing. While it's appropriate to include extra, more detailed information in appendices, the main text still needs to be able to stand on its own, and in my opinion, it doesn't quite. I would suggest that the main equations and brief explanations should also be included in the main text, including all the equations that a reader would need to apply to calculate the kinds of results presented in this work. The appendices also include a lot of pedagogical development; this is the kind of information that I think rightly belongs in the appendix for readers who want more details. Since the appendices are 5 different topics, I also suggest that each appendix should be exist as a separate entity, with all variables defined, so

that a reader can read Appendix D to learn about the application of constraints or Appendix E for the "practical aspects" without a close reading of Appendix A,B, and C, to find the definitions of the variables.

The organisation of the paper is indeed a delicate issue that was also brought up by other reviewers. Our main goal is to make this paper accessible to a broad community, including lidar instrument developers, remote sensing groups, and data assimilation researchers. For this reason, we prefer to include most of the theoretical developments in the appendix. However, we agree that this creates a significant problem by removing essential information from the main body of the paper. In the revised paper we will follow the reviewer's suggestion and re-state the most essential theoretical results from the appendix in the main text. This will make the paper more readable and self-contained, while avoiding the risk of making it too technical, which could narrow down the readership of this work.

3. The results and conclusions are also a little too abbreviated. Some key aspects are missing, like how was the specific weighting chosen and how do we know this is the best weighting? Also, as pointed out by another reviewer, the assessment (section 3.2) is really more of a demonstration. That is, although the theoretical development is compelling, the application/assessment section isn't sufficient to convince readers that this is a better way to assimilate lidar data than another way. This paper clearly reflects a lot of research on the part of the authors and I think the missing information probably exists but was left out in the effort to streamline the manuscript. I think adding this additional information should be fairly straightforward and would improve the usefulness of this research for the modeling and lidar communities without adding too much complication to the nice flow of the paper.

This is also an important point, which was brought up by several reviewers. We will perform additional computations using the unconstrained assimilation algorithm and compare the constrained to the unconstrained analysis. The hypothesis

is that the constrained analysis should be less noisy, because the unconstrained analysis is at risk of assimilating noise. Also, we will eliminate all instances of "numerical experiment" and replace it by a more appropriate term, e.g., "numerical test", "demonstration", or "illustration". Finally, we will add more explanations to Sect. 2.4 about the construction of the covariance matrix in the constraint term.

**2 Specific comments**

1. Lines 151-158: Here is an example where I think some important things are missing from the main text which only appear in the appendices. These eight short lines are the methodology section for the key calculations that are the novel part of your research and are critically important for a reader to understand. I suggest that a way to decide what should also be included here would be to target the subset of equations that a reader would need to apply to calculate results like yours, but without their derivations. Also include enough supporting explanation to describe what the equations say and how to use them.

   We agree, and we will make changes following the more detailed suggestions given in the following comments.

2. L152-153: Specifically here, Eq C6 and C?16 should be included in the text, since they are required to understand the meaning of the sentence. Later, at L159-160 where readers are directed to the appendix for more background information, I think that's fine.

   OK, we will revise the text and include the equations for the observation operator, the observability matrix, and the singular-value decomposition thereof.

3. L155-157: The equations for signal degrees of freedom and Shannon information content should also be included in the text.

   OK, this will be added with accompanying text.

4. L165: "a numerical experiment". In fact, it's more of a demonstration than an experiment. It's useful as a demonstration of the results of the technique, but there's nothing in the demonstration that addresses a hypotheses. Sharing more of the background work would make the paper more compelling. For example, as another reviewer suggested, comparing to a control experiment would be necessary for convincing readers that this technique is useful. For another example, a pair of runs with different weightings in the assimilation would help answer the question of why the weighting that was ultimately chosen was the best one.

We will replace "numerical experiment" everywhere in the paper, as mentioned previously. Next, we will show a control run with the unconstrained assimilation system. The hypothesis is that the constrained analysis should be less noisy than the unconstrained analysis. We will revise the Figures and show both the unconstrained and the constrained analysis in the same plot. Also, we will add an extra figure to show both analysis results for different aerosol species in different size bins, as these are even more sensitive than size-integrated total mass mixing ratios. Finally, the case we picked in the original manuscript was not particularly challenging, since the background state was fairly close to the reference state. In the revised paper, we will pick a more challenging case in order to make the differences between both analysis runs as clear as possible. As for the different weightings, our tests, so far, indicate that the different approaches result in rather similar analysis results. So, the constrained analysis is not as strongly dependent on the weighting as one may expect. We will clarify this point by adding a discussion to Sect. 2.4.

5. L177: Depolarization is not included in the studied parameters, yet lidar studies have shown that depolarization measurements contain some information about aerosol composition (for example, Omar et al. 2009 as referenced in the introduction, but there are many others). Do the authors have any comment on depolarization and why it isn't included in this study?

There are two major problems. The obvious practical problem is that the forward model would need to be based on nonspherical particles (as spherical particles do not depolarise). However, our simpler optics model is entirely based on spherical particles, while our newer optics model only accounts for the nonsphericity of bare black carbon, but not for that of mineral dust or dry sea salt. Thus our capabilities of modelling depolarisation are presently limited. The second problem is that the observation error for depolarisation may be very high, even though the measurement error is very low. This is because the forward-model error is likely to be quite high, since even slight variations in particle geometry (e.g. Kahnert et al. (2012)) or inhomogeneity (e.g. Kahnert (2015)) can result in large variations in the depolarisation ratio. If the forward-model error is, indeed, high, then the prospects of using depolarisation for constraining CTM model results are likely to be low. However, this question is open and will be investigated in future studies. But in order to do so, one would first need to obtain estimates of the forward-model error (e.g, by computing depolarisation ratios while varying particle morphology).

6. Table 1 and related discussion: From a lidar standpoint, some combinations of channels are more technologically affordable than others, so the discussion of which channels add significant information content is very interesting. However, the utility for the lidar community would be maximized if the combinations were ordered such that they roughly increase in technological difficulty. Also, some combinations don't really make sense from a technological standpoint. There is no lidar that measures extinction but not backscatter at the same channel (although modelers may use only the extinction). On the other hand, backscatter (actually attenuated backscatter) without a direct measurement of extinction is common. Also, since CALIPSO, CATS, EarthCARE and the $3\beta + 2\alpha$ combination of airborne HSRL2 are mentioned in the introduction and motivation sections, it would be useful if the combinations relevant to those instruments were

included. CALIPSO = CATS = $\beta(\lambda 1) + \beta(\lambda 2)$. EarthCARE = $\beta(\lambda 3) + k(\lambda 3)$. HSRL2 = $\beta(\lambda 1) + \beta(\lambda 2) + \beta(\lambda 3) + k(\lambda 2) + k(\lambda 3)$. I would suggest these combinations of backscatter and extinction would be most interesting and useful to the lidar community: $\beta(\lambda 3)$

$\beta(\lambda 1) + \beta(\lambda 2)$

$\beta(\lambda 1) + \beta(\lambda 2) + \beta(\lambda 3)$

$\beta(\lambda 3) + k(\lambda 3)$

$\beta(\lambda 1) + \beta(\lambda 2) + k(\lambda 2)$

$\beta(\lambda 1) + \beta(\lambda 2) + \beta(\lambda 3) + k(\lambda 2) + k(\lambda 3)$

For these experiments, it appears that the observation error was always assumed to be the same in every channel. I think it's a reasonable assumption, to first approximation, that the measurement error would be similar in every channel, but as pointed out at L78-79, some lidar retrievals include additional non-random errors that can be much larger. This could and should affect the choice of channels to assimilate. For example, the Raman, HSRL, and transmittance techniques are fairly direct measures of extinction, but techniques that require an inferred lidar ratio to convert backscatter to extinction have relatively little additional measurement information content in the extinction.

We welcome the reviewer's suggestion to take technical realisations of lidar systems into account, and we will revise Tables 1 and 2 according to the reviewer's specific suggestions. We will also add a comment on the observation errors of lidar measurements, specifically on the fact that the observation errors may be different for different channels/parameters.

7. L 197-201. Here also the discussion of incorporating soft constraints and the specifics of the three weighting schemes should be in the main text of the paper and not just the appendix, since it is discussed here in the results section. This section is not understandable without the equations from the appendix and most of section D3.

We will remove this discussion here. Instead, we will briefly discuss the construction of the constraint covariance matrix in Sect. 2.4.

8. L 203-204. Discussion of observation error vs. measurement error. This is interesting and useful, but could be clarified as to whether the forward model error (due to poor assumptions) is considered part of the observation error or is another separate source of error. If it is part of the observation error, how are the forward model errors represented and how are they transformed into the space of the measurement vector?
   We will extend the text to clarify that the observation error is given by $\epsilon_o = \epsilon_m + \epsilon_f$, where $\epsilon_f$ denotes the forward-model error. We will also add a citation to the paper by Rabier et al. (2002) with a hint to their Eq. (1), which explains this terminology. A way to determine the forward-model errors theoretically is to perform light-scattering calculations while varying various parameters, such as particle morphology, refractive index, and size distribution within typical uncertainty ranges. This can provide us with an estimate of $\epsilon_f$. To the best of our knowledge, it would be very difficult to determine $\epsilon_f$ with experimental methods.

   We are not sure if we understand the last question. $\epsilon_f$ enters into the definition of the observation error covariance matrix, i.e. $\mathbf{R} = \langle \epsilon_o \cdot \epsilon_o^T \rangle$, which is a matrix in the space of the measurement vector. No further transformation is necessary.

9. L 207 While there may be retrieval errors in the lidar backscatter and extinction due to assumptions, assumptions on particle shape and size distribution are not among the assumptions used in lidar retrievals. These examples belong only to the optics model (forward model). So, perhaps delete "also". Poor assumptions in the optics model or in lidar retrievals would presumably lead to bias errors, whereas measurement errors would more typically be random. Does this make a difference in the analysis?
   OK, we will delete "also". We would generally not be sure if assumptions in the optics model necessarily (mainly) lead to biases. For instance, model errors may

be dependent on size and morphology of the actual particles. The errors would, correspondingly, fluctuate over time. The amplitude of this fluctuation may well be larger than any possible biases. However, in case that the forward-model does introduce a large bias, than this would, indeed, be a problem, since analysis algorithms are typically based on the assumption that the errors are unbiased.

10. L 219. I strongly agree that estimating the uncertainties in the optics model is very important. Some discussion here seems warranted about how that can be done. Later I see that this is discussed in the summary (L281 – 292) but I think it would be better if it comes up first here in the discussion section.
Agreed. We will add an explanation, but we will also mention it again in the conclusion section.

11. L 256 and caption to Fig 4. In both places, it would be kind to remind readers that the delta notation in $\delta x'$ means this is the difference between the value and the background value.
It is not so simple. $\delta x$ in physical space is the difference between the value and the background, while $\delta x'$ is obtained from $\delta x$ by applying the transformation $\delta \vec{x}' = \mathbf{V}_R^T \cdot \mathbf{B}^{-1/2} \cdot \delta \vec{x}$. We will repeat this definition in the text with a reference to the definition (which is now found both in the main text and the appendix), and we will add a reference to the defining equation both at this point in the text and in the caption to the figure. But we think it would be a bit overdone to repeat the equation in the figure caption.

12. L 259-263. The choice of D21 with its sharp drop-off in weighting appears to mean that only one transformed variable is allowed to change in a meaning-ful way, although the measurement scenario chosen has nearly the maximum amount of information content available, close to DOF=4. Why was D21 chosen instead of D18, which would allow the measurements to play a bigger role? The only discussion of this choice is the rather vague comment in the Appendix "it is a

matter of experience to test different approaches and select the one that proves to be most suited". How and why was this approach determined to be the most suited?

We have done some additional tests and found, in fact, that the analysis is less sensitive to the choice of weighting than we expected. We will explain this in the revised paper in Sect. 2.4. Also, we will do the following changes to Fig. 4. First, we will show $\delta x'$ for both the constrained and the unconstrained analysis. Thus the whole discussion of the figure will shift from a mere description of the behaviour of the constrained analysis to a comparative discussion. This will make it much clearer what kind of effects the weak constraints have on the analysis increments. Second, following a suggestion by reviewer 4, we will not show all 20 panels, but only a subset of panels sufficient to illustrate the different behaviour of signal- and noise-related (phase-space) model variables. Third, as mentioned earlier, we will pick a more challenging case in which the reference and background results differ more strongly than in the case we originally picked. So this figure will be changed considerably, and the accompanying discussion will become a lot more informative.

13. Comparison of Figure 3 and Figure 2, if I understand right, underscores the fact that there is a significant null space, not controlled by the measurements, since essentially the same measurements in Fig 3 correspond to both the black and red lines in Fig 2. What is not clear to me is what happens in a standard assimilation to the variables that are not well controlled by the measurements? Do they remain close to the background values, or do they vary wildly and arbitrarily? If the former, then the exercise of determining the singular values wouldn't help the assimilation very much (but would still be useful in terms of building knowledge about what we can and can't actually measure). On the other hand, if a standard assimilation arbitrarily varies state variables in the null space, then this is a very important motivation for this technique (and maybe that motivation could be

emphasized a little bit more in the introduction and conclusions). Not being very familiar with the field of model assimilation, I guess but don't actually know that there must be other "regularization" techniques in use to prevent an assimilation from arbitrarily varying parameters that are mostly in the null space of the observations, although I imagine existing techniques may be more ad hoc than the method presented here. Can you comment on other methods and demonstrate how this method performs better than other methods?

The reviewer's comment about the null space and the behaviour of the unconstrained (standard) assimilation raises an important issue. As mentioned earlier, we have now run an additional unconstrained assimilation, and we will show a comparison of both methods. Figure 2 will be replaced by two figures. The first figure will, similarly to the old figure 2, show the total mass concentration of different aerosol species, but now for both the constrained and the unconstrained analysis. The second figure will show a similar comparison of a selection of aerosol species in specific size bins. We anticipate that this comparison will illustrate that the unconstrained analysis yields more erratically varying vertical profiles (i.e., results that vary more wildly in the null-space).

As for ad hoc methods, we did review previously reported approaches in the introduction, such as the one by Benedetti et al. (2009) (L 53-54) based on constraining the total aerosol mass mixing ratio, and the one by Saide et al. (2013) (L 55-56) based on constraining the mass mixing ratio per size bin. One obvious disadvantage is that these approaches are quite inflexible. The number of constraints is fixed in these methods, so one cannot easily adapt the number of constraints to the number of independent measurements to be assimilated, as we can in our approach. (In fact, our method automatises this process.) Also, the available information may not be optimally exploited by these methods (L 57-59). We have not tested such methods, so we cannot comment on their performance. However, we also believe that the burden of proof for such a demonstration does not lie with us. We are employing a mathematically well-founded approach based

on information theory. If other groups choose to not follow us, but continue to use ad hoc methods (which, admittedly, may be quite attractive owing to their simplicity), then it is up to them to demonstrate that such ad hoc methods yield sufficiently accurate results while exploiting the available measurement information. Owing to the ad hoc nature of these methods, such a demonstration would have to be repeated for any new set of measurements to be assimilated. Our method can serve as a reference for such tests.

14. L 298-299. "It also appeared". This result is disappointingly empirical for such a well-founded theoretical study. This observation that SIC was most faithfully retrieved was made in a single case– would you expect this result to be general for all cases, and why? Answering the question is complicated since the singular variables are defined only in the transformed space and therefore the information about what variables are or are not constrained by the measurements is only in this transformed space, not the state space. Yet this statement highlights that it's desirable to have information about which chemical species and size bins are constrained by the measurements. Is there any way to provide information about this quantitatively? For example, since each state variable is a linear combination of the transformed variables, would showing the linear coefficients in a table make it more obvious which state variables are most closely related to the most significant transformed variables? Perhaps there is a way to use the coefficients to calculate a "fractional significance" that would indicate that x% of the variability in a given state parameter is orthogonal with significant transformed variables while (1-x)% is orthogonal with insignificant variables?
This is a very good suggestion. We will add an extra figure with accompaning discussion and show the magnitude of the linear coefficients for the signal-related control variables. However, the coefficients will depend on the B- and R-matrices, which vary spatially. So, we do not anticipate that we can draw very general conclusions from a single test case. But we do think that such a discussion can help

us understand why the analysis behaves the way it does in our specific case.

**3   Minor comments**

1. L37: Muller et al. 1999 and Veselovskii et al. 2002 and related papers (there are many) would be more relevant references here since they detail retrievals of refractive index, etc., from lidar. (Mishchenko et al. 2007 is an introduction to the Glory satellite and was about retrievals from a polarimeter.)
Müller, D., U. Wandinger, and A. Ansmann (1999), Microphysical particle parameters from extinction and backscatter lidar data by inversion with regularization: theory, Appl Optics, 38(12), 2346-2357, doi: 10.1364/AO.38.002346.
Veselovskii, I., A. Kolgotin, V. Griaznov, D. Müller, U. Wandinger, and D. N. Whiteman (2002), Inversion with regularization for the retrieval of tropospheric aerosol parameters from multiwavelength lidar sounding, Appl Optics, 41(18), 3685-3699, doi: 10.1364/AO.41.003685.
Agreed. The references will be replaced.

2. L99: I infer that the ratios in the different size bins are fixed, or else there would be much more than 20 total variables. Is there a way to concisely clarify this in the sentence?
There is no way to say this in a simple sentence, because it is not quite as simple as the reviewer suspects. We have gridded emission data, which means that the ratios among size bins can vary from one grid cell to the next. Although the mass-transport model does not account for microphysical processes (such as condensation, which would result in a dynamic evolution of the size distribution), this ratio can still dynamically evolve in each grid cell owing to transport processes and mixing of air masses originating from different emission sources.

3. L109: maybe replace "in the present setup" with "currently in that version". "The present setup" seems to refer to "the setup used in the present study" but that is misleading, since the present study uses the 20-variable version of the model.
Agreed.

4. L134: "an" should be "and"
Yes.

5. L142: "Error correlations ::: are not assumed to be separable". I'm not sure what this means. What is (or is not) separable from what?
Vertical and horizontal correlations are often assumed to be separable. We do not make such assumptions, because vertical correlations are often stronger on larger horizontal length scales. In our spectral model (where the horizontal correlations are Fourier-transformed) this means that vertical correlations are larger for smaller horizontal wavenumbers. Since this is not so essential in the context of this study (and potentially confusing), we will remove this text in L 142.

6. L153: "see Eq. D16". Should this be C16?
Yes. However, following earlier suggestions by the reviewer, this text will now be revised and supplied with the main equations from the appendix. So the text in its present form will be replaced.

7. L162-164: Should this sentence perhaps be part of section 2.4, as part of the description of the new technique? The rest of this paragraph (L164-174) is more about the demonstration of the new technique and so seems like a somewhat distinct topic.
Agreed, we will move this text.

8. Figure 1: The caption says "note the nonlinear colour scale" Actually, the scale is hardly visible. Please expand the axis labels so they are a similar text size to the caption text.

Actually, we think that this figure is not particularly relevant in the context of our study, since we do not consider aspects of regional modelling or horizontal information spreading in the analysis. It merely shows one out of many model variables in a single model layer, which does not convey much useful information. Also, since we consider a single profile, the analysis impacts the mass mixing ratio only at and around the observation site, which is difficult to see in a regional plot. We therefore suggest to remove this figure in the revised manuscript.

9. Figure 2: The axis labels' and inset box labels' font size should also be increased here.
   OK, we will increase the font sizes in all figures wherever necessary.

10. L 391. The variable n is not defined. Possibly this is the only case, but I would also request that variables be re-defined frequently when used in key equations. If a reader is directed from another part of the paper to Equation D18 or C12, for example, then it would be nice if all the information relevant to understanding that equation is given immediately after that equation, rather than having to scroll through 8 or 10 pages to relocate the definitions of key variables.
    Agreed, we will add the definition of n. Also, the problem with directing the reader to equations in the appendix will be significantly alleviated in the revised versions, since we will re-state the key equations in the main body of the paper (see our response to an earlier comment).

11. L563. The symbol lambda is used for wavelength elsewhere in the text. You might consider using a different symbol here.
    OK, we will replace it by mu.
**References**

Benedetti, A., Morcrette, M. J.-J., Boucher, O., Dethof, A., Engelen, R. J., Huneeus, M. F. H. F. N., Jones, L., andS. Kinne, J. W. K., Mangold, A., Razinger, M., Simmons, A. J., and Suttie, M.: Aerosol analysis and forecast in the European Centre for Medium-Range Weather Forecasts Integrated Forecast System: 2. Data assimilation, J. Geophys. Res., 114, D13 205, 2009.

Kahnert, M.: Modelling radiometric properties of inhomogeneous mineral dust particles: Applicability and limitations of effective medium theories, J. Quant. Spectrosc. Radiat. Transfer, 152, 16–27, 2015.

Kahnert, M., Nousiainen, T., Lindqvist, H., and Ebert, M.: Optical properties of light absorbing carbon aggregates mixed with sulfate: assessment of different model geometries for climate forcing calculations, Opt. Express, 20, 10 042–10 058, 2012.

Rabier, F., Fourrié, N., Chafaï, D., and Prunet, P.: Channel selection methods for infrared atmospheric sounding interferometer radiances, Q. J. R. Meteorol. Soc., 128, 1011–1027, 2002.

Saide, P. E., Charmichael, G. R., Liu, Z., Schwartz, C. S., Lin, H. C., da Silva, A. M., and Hyer, E.: Aerosol optical depth assimilation for a size-resolved sectional model: impacts of observationally constrained, multi-wavelength and fine mode retrievals on regional scale analysis and forecasts, Atmos. Chem. Phys., 13, 10 425–10 444, 2013.

---

## Author Comment (AC3) · 2 Feb 2017

Below the reviewer comments are marked in blue, our response is marked in black.

1. It is well known, that the problem of inversion of standard $3\beta + 2\alpha$ lidar measurements to the particle microphysics is undetermined and to constrain it, numerous techniques were considered. The authors suggest an interesting approach to assimilation of lidar measurements into chemical transport model. It looks like a promising concept to extract the information about particle parameters from lidar measurements. Paper is very well written and should be published.

[Figure]

We thank the reviewer for his positive evaluation of our manuscript and for his helpful comments.

2. The structure may be questionable, because a half of material is put in appendices. These appendices are clearly written and are definitely useful for unprepared reader. I personally, had no problems with material structure.
We agree that the structure was not optimal for all types of readers. We found that the compromise suggested by reviewer 2 would adequately address these concerns. We refer to our detailed response to reviewer 2, which explains the changes we intend to implement in the revisions.

3. Additional references to the previous studies of lidar data inversion would be desirable , and other Referees have already suggested several.
Agreed. We will add a paragraph in the introduction with a brief discussion of other studies, also from numerical weather prediction data assimilation.

4. Stability of retrieval strongly depends on aerosol type. It is more challenging for aerosols with dominant coarse mode and for particles with strong absorption. The authors consider only one example (not the most challenging) in their simulation, so it is not very clear how the approach will work for other aerosol types. But this may be a subject of separate study.
Yes. Although it is not the subject of this paper to comprehensively test all sorts of mixed aerosol populations, we do agree that the case we picked was a little bit too easy. This is mostly because the background and reference cases were very close to each other. In such a case one does not see very clear differences between a constrained and unconstrained analysis. In the revised paper we will pick a more challenging case, and we will show our test results for both the constrained and the unconstrained 3DVAR algorithm. This will help to better illustrate what practical significance the constraints can have.

---

## Author Comment (AC4) · 2 Feb 2017

Below the reviewer comments are marked in blue, our response is marked in black.

**Summary**

The authors consider the case of assimilation of remote-sensing data (specifically aerosol extinction and backscattering coefficients) applied to aerosols fields within a chemical transport model. They describe how an additional term can be added to the 3D-var cost-function so that the assimilation adjusts only those components (in a transformed space) for which the observations provide information. The additional term

relies on the singular value decomposition of the scaled observation operator. In this way, the assimilation automates the choice of control variables in an otherwise highly under-constrained inverse problem.

**Verdict**

The paper is very well written and is surprisingly clear, given the subject matter. The manuscript introduces a potentially very powerful concept for variable selection into the field of aerosol data assimilation. The authors have probed the idea in a minimal test case, which assists in understanding the effects. I found that the shortcomings of the paper were relatively minor. I felt there was insufficient discussion of the literature of related treatment. I was unsure about whether the organisation of the material was optimal (see the "Main comments"). Finally, a counter-experiment without the addition of the new constraint in the 4D-var cost function was, in my opinion, lacking. All in all, I believe that the paper should be published, pending the minor revisions suggested below.

We are grateful for this encouraging assessment of our work, as well as for the insightful comments and suggestions. It is obvious that the reviewer has devoted considerable time into studying the manuscript and providing constructive criticism on various aspects of the content and organisation of the paper. Our detailed response to these comments follows.

**1  Main comments**

1. There was little or no discussion of literature on related treatments. I have not the time to read all of these myself, however I have included a list at the end of articles that may be relevant, for example those that deal with information content of observations in data assimilation or those that refer to the singular value decomposition of the observability matrix.

   We have, indeed, only cited studies on aerosol data assimilation. Most of the

studies cited by the reviewer are concerned with numerical weather prediction (NWP). We will add a paragraph to discuss related NWP studies and include the citations suggested by the reviewer.

2. I believe that a small counter-experiment was lacking. In the results presented in section 3.2, I would suggest also presenting results for the assimilation experiment which did not include the additional constraint in the 3D-var cost-function. This is a very valid point that was also brought up by other reviewers. We will include these results and revise the figures and discussion accordingly. In particular, we intend to replace Fig. 2 by two new figures. The first figure will, similarly to the old Fig. 2, show the total mass concentration of different aerosol species, but now for both the constrained and the unconstrained analysis. The second figure will show a similar comparison of a selection of aerosol species in specific size bins. We anticipate that this comparison will illustrate that the unconstrained analysis yields more erratically varying vertical profiles (i.e., results that vary more wildly in the null-space).

3. I was unsure whether the organisation of the material was optimal - I highlight this as an issue that the editor may wish to take up. The introduction concludes by urging the reader to read the Appendix before proceeding onto the rest of the methods and results section. Much of the interesting methodology is contained within the Appendix, and we agree that it would be difficult to make sense of the main part of the paper without a good understanding of the contents of the Appendix. As such, I would suggest incorporating the Appendix into the main body of the text. At one level, this is really a matter of taste, and thus I leave it to the editor.
This is a tricky point. We put some thought into this before writing the paper, and we concluded that the appendix is, indeed, most interesting for readers who are mainly interested in data assimilation methodology, and for those who are very eager to learn something about it. But other readers, e.g. lidar instrument

developers, will most likely be deterred from reading the paper if we merge the entire appendix with the main body of the paper. However, the reviewer's criticism is very valid, and it has been brought up by several reviewers. We believe that reviewer 2 has suggested a very good compromise, namely, to state and explain the main results (equations) from the appendix in the methodology section of the paper, while retaining the derivations and more detailed explanations in the appendix. This alleviates the problem that parts of the main text are hard to understand without the information given in the appendix. At the same time, we avoid the risk of making the paper inaccessible (or just too boring) for those who do not mainly work with data assimilation methodology.

We therefore propose to follow the suggestions of reviewer 2 in this point. It seems to us that this will also adequately address the main point of criticism brought up by reviewer 4.

**2   Minor comments**

1. When describing observation errors, there was no reference to the component from "representativity errors" (i.e. measurements are made at a point, or over a small area in the case of remote sensing, while model grid-boxes are typically in the order of kilometres across in the horizontal dimensions). All of the discussion about observation errors was in terms of the measurement error and errors in the observation operator, both of which are relevant. However the representativity component is not insignificant in many contexts.
Agreed. We will add a discussion of the representativity error in the text accompanying table 2, where we will make it clear that in this numerical test we have neglected this source of error.

2. Observation standard deviation was reported in percentage, but it was unclear

what this was a percentage of. Please clarify.

It is a percentage of the observed backscattering coefficient or extinction coefficient. We will change the text in Sect. 2.5 from "We assumed an observation error standard deviation of 10 %" to "We assumed that the observation error standard deviation is 10 % of the measurement value."

3. I would suggest replacing all instances of the term "costfunction" with "cost function" (or "cost-function"). The latter is about 15 times more common (on the web, at least). Similarly, I believe that the compound word "nullmatrix" is used in German (capitalised, that is) whereas it is "null matrix" (or "zero matrix") in English.

Agreed.

4. I could not find a definition to the term "signal degrees of freedom". Please include this somewhere (preferably at first usage, or in the Appendix).

We will add an explanation of the terminology to Sect. 2.4. Following the suggestions of reviewer 2, we will provide key equations of the appendix with explanations in the main text. This will also apply to Eq. C12, which will be provided in the methodology section. Thus the explanation and definition of the term "signal degrees of freedom" will appear much earlier in the revised paper. (Note that in the remote sensing community the number of signal degrees of freedom is also known as the "effective rank" of the problem.)

5. Line 48: Please replace "This is a rather bold approach that largely disregards ..." with "This is approach largely disregards ..." – please use argument rather than rhetoric to explain what is wrong with the work of others.

Agreed.

6. Line 54: The reference to Kahnert (2009) is used to show that several optical properties at multiple wavelengths may allow constraining more than just the total mass concentration. Surely other authors have looked into this. If so, please summarise other work done. If not, please say so.

We do cite the study by Burton et al. (2016) (L64), although we do so in the introduction. We will add two more references that analyse the information content of lidar observations, namely, the papers by Veselovskii et al. (2004) and Veselovskii et al. (2005). However, these papers analyse the information content with respect to particle size and refractive index, not with respect to chemical composition. Therefore, the citation of these papers fits better into the introduction, together with the citation of Burton et al. (2016).

7. Line 98: "... using 40 eta-layers with variable thickness depending on the underlying topography" – do you just mean that this is a terrain-following coordinate? Or is there something more sophisticated about this?
OK, we will replace this with "using 40 terrain-following coordinates".

8. Line 125: "The background error covariance matrix of the model a priori is modelled with the NMC method ..." . I checked the reference (Kahnert, 2008), in which I believe this is described. If the implementation is the same here as in the 2008 article, then I believe that it is best to say that it "follows similar principles to the NMC method" or "is inspired by the NMC method". If it is indeed the NMC method, the authors should clarify the difference to methodology laid out in Kahnert (2008).
OK, we will replace this with "follows similar principles to the NMC method".

9. Line 129: I would suggest replacing "Given m observations of, e.g., m1 different parameters at m2 different wavelengths, so that m1 m2=m, how many..." with "Given m observations (e.g., m1 different parameters at m2 different wavelengths, so that m1 m2=m), how many..."
Agreed.

10. Line 130: "... we can constrain to better than observation error" – do you mean "model error"? If not, please explain that the transformation makes the (rescaled) observation errors and (rescaled) model variables comparable.

This was a bit confusing. We will reformulate this sentence, and we will add a more detailed explanation of the terminology *signal degrees of freedom*.

11. Line 134: Please replace "... a singular value decomposition of the Jacobian of the observation operator ..." with "... a singular value decomposition of the Jacobian of the scaled observation operator ..." or something similar. By the way, this scaled observation operator appears to have a name: "the observability matrix"
Yes. Actually, this text will be extended with a lot more explanations, and it will provide the main equations from the appendix. We will follow the reviewer's suggestion and introduce the term *observability matrix* for the scaled Jacobian.

12. Footnote 2, page 5: I found this distinction a bit cryptic. Please consider rephrasing.
There seem to be two fractions in the community. One that uses *data analysis* and *data assimilation* almost interchangeably, and another that insist on keeping these two concepts apart. We are mostly guilty of belonging to the first one, but we do not want to make a big deal out of mere questions of terminology (which is why we put this into a footnote rather than into the main text). However, we will do our best to clarify the text in the revised manuscript.

13. Line 150: I realise that this is something that is clarified later on, but I would suggest saying a few words at this point about the synthetic observations; namely, what kind of observations they were and how many observation points there were.
Agreed; we will add this information in the revised manuscript.

14. Line 154: I would suggest the following change "thus providing nearly perfect observations. (We assumed an observation error standard deviation of 10 %) The only ..." becomes "thus providing nearly perfect observations (we assumed an observation error standard deviation of 10 %). The only...". See also my comment about about describing the units for the observation error standard deviation.

Agreed (replacing "observations. (We" by "observations (we". In addition, in response to an earlier request to be more specific what me mean by "10 %" (percent of what?), we will replace the text in parenthesis with "(we assumed that the observation error standard deviation is 10 % of the measurement value)".

15. Line 162: What is "Nd:YAG"? Please clarify. I suspect that this is some error with the bibliography manager.

    It is no error. "Nd:YAG" is the standard abbreviation for "neodymium-doped yttrium aluminium garnet" laser, one of the most commonly used solid-state lasers in remote sensing. We will add this information.

16. Line 168: I would suggest the following change: "... two wavelengths. (Compare, e.g., cases 1., 2., and 3. to cases 4., 5., and 6.) Hence ..." becomes "... two wavelengths (compare, e.g., cases 1., 2., and 3. to cases 4., 5., and 6.). Hence ..."

    Agreed. However, reviewer 2 has suggested to replace the cases considered in Table 1 with different cases that are more closely associated to combinations of wavelengths and parameters that are technologically feasible and common. Thus the text accompanying Table 1 is likely to change considerably.

17. Line 171: a missing full stop after the right parenthesis.
    Agreed.

18. Table 1, caption: the "Nd:YAG" term appears again.
    See our earlier response.

19. Line 181: I believe that "weak constrains" should be "weak constraints".
    Yes.

20. Line 189: See my comment above about the representativity component to the observation error.
    Agreed, see our earlier response.

21. Figure 1: I think it would be interesting to see the increment as an additional panel in this figure.
    Figure 1 has been criticised by several reviewers. In fact, this figure is not particularly useful in the context of our paper. We are not discussing any aspects of regional modelling or horizontal information spreading in the assimilation algorithm. The model merely serves us to provide us with a test case. So, we will remove this figure in the revised manuscript (see also our response to reviewer 2).

22. Figure 1: The text on the scale is a bit too small. I would suggest having one scale, rather than three, and enlarging the scale so that the labels can be read.
    See the previous item.

23. Figure 2: The units appear to be "mixing ratio [ppb-m]". Do you mean mass mixing ratio? Please clarify.
    Yes. This will be corrected in the revisions.

24. Figure 4: Do we need all panels? Why not just show the first three or four, and then a selection of the remaining terms.
    Agreed, we will show 10 instead of 20 panels. Following the comment by reviewer 2, we will run a $3\alpha + 2\beta$ test case, in which case we will have 5 signal degrees of freedom. Thus we will show the first 5 signal-related transformed increments, and 5 out of the 15 noise-related increments.

25. Line 262: I would suggest the following change " ... dramatic decrease in both the entropy and signal degrees of freedom ..." becomes " ... dramatic decrease in both the entropy-change and signal degrees of freedom ..."
    Agreed.

26. Line 282: "It also appeared that among the original model variables, secondary inorganic aerosol components were most faithfully retrieved by the inverse modelling solution" – why is this? why SIA? Do they have specific optical properties to make them more observable by such LIDAR pseudo-observations?

This question has been brought up by several reviewers. We follow the suggestion of reviewer 2 and add an analysis of the linear coefficients that transform the elements in model space to the signal-related control variables. We will add a figure and a discussion — see our detailed response to reviewer 2.

27. Line 293: I would suggest the following change: "The present study should be extended..." becomes "The present study could be extended..."
Agreed.

28. Line 295: I believe that the expression "highly underrated" is somewhat dramatic and relatively colloquial, and does not fit with the tone in the rest of the paper. The authors are encouraged to use argument rather than rhetoric to make their point.
OK, we will replace the text with "Another important issue concerns the choice of ...".

29. Line 297: Regarding the statement "There is little one can put forward in defence of this model other than pure convenience". Some justification is required (e.g. some references) to demonstrate why this model is untenable. There's a saying (attributed to George Box) "All models are wrong, some models are useful". Does this model give significantly worse results than representations, or is it just inaccurate in its assumptions?
Worst of all, this model is rather unpredictable, since its accuracy depends on the size, refractive index, and shape of the aerosols. Also, it may, in some cases, give reasonable results at one wavelength and for one specific parameter, and fail at other wavelength or for other optical parameters.

There is a large body of work concerned with aerosol optics and the shortcomings of simplified model particles. Some of these studies focus on specific types of

aerosols, others on specific morphological properties, such as non-sphericity, inhomogeneity, surface roughness, or chemical heterogeneity. It is difficult to pick just a few of such studies as representative citations. So, perhaps the best we can do is to cite a recent review paper on aerosol optics modelling that discusses the strengths and shortcomings of various morphological models (Kahnert et al., 2014).

30. Paragraph beginning at line 307: It may be worth making it clear that y is not observed, but a model equivalent of the observations
We will insert the following sentence: "The operator $\hat{H}$ maps from model space into observation space, which allows us to compare model output and observations."

31. Lines 324 and 326: I would suggest replacing all instances of "3-dimensional" with "three-dimensional"
Agreed.

32. Paragraph beginning 336: I would suggest mentioning that the assumption of unbiased background and observation errors
Agreed.

33. Footnote 6, page 15: See my comments above about the representativity component of the observation error.
OK; see our earlier response.

34. Footnote 7, page 16: I would suggest the following change: "The observation errors are often uncorrelated" becomes "The observation errors are often assumed to be uncorrelated (this is not always true)"
Agreed.

35. Paragraph beginning at line 368: Please comment on the role of spatial and interspecies correlations, particularly in light of the comment "if we allow all model

variables to be freely adjusted" (line 374).

OK. We will add the following footnote (after "(within the given error bounds).": By solving the equation $\nabla J|_{\vec{x}=\vec{x}_a} = \vec{0}$ for the analysed state $\vec{x}_a$ it can be shown that the solution to the inverse problem is given by $\vec{x}_a = \vec{x}_b + \mathbf{K} \cdot (\vec{y} - \hat{H}(\vec{x}_b))$, where $\mathbf{K} = \mathbf{B} \cdot \mathbf{H}^T \cdot (\mathbf{H} \cdot \mathbf{B} \cdot \mathbf{H}^T + \mathbf{R})^{-1}$ is known as the gain matrix. This illustrates that the analysis updates the background estimate $\vec{x}_b$ by mapping the increment $(\vec{y} - \hat{H}(\vec{x}_b))$ from observation space to model space by use of the gain matrix. The correlations among the model variables enter into the gain matrix through the matrix $\mathbf{B}$ In our case the vertical correlations are rather weak in comparison to correlations among different aerosol species.

36. Line 369: It might be worth noting that $\delta x$ is not constrained to ensure that all components of x remain positive in the analysis.

There is no such constraint in the minimisation process itself, but we do post-process the results for $\delta x$ such that negative concentrations would be set to zero. In practice, this rarely ever happens.

37. Line 386: The phrase "rather tricky" strikes me as somewhat colloquial. I would suggest the following change: "However, to actually make such a comparison israther tricky" becomes "However, to actually make such a comparison poses two problems."

Agreed.

38. Paragraph beginning at line 390: Please introduce the meaning of the angle-bracket notation. I believe that this is common in physics, but other disciplines (e.g. statistics) often use different notation for the expectation.

OK, we will add a formal definition of the expectation value for discrete variables in a footnote.

39. Footnote 8, page 17: Should $A \cdot A = B$ be $A^T \cdot A = B$?

Yes!
40. Line 425: Should "(C7)-(C9)" not be "(C6)-(C9)"? As far as I can see, Eq. (C6) is required here.
Yes.

41. Line 434-435: Please state which particular sections/chapters of Rodger (2000) the reader is referred to.
Agreed.

42. Equations C12, C15: I would suggest showing the range of the summation to indicate that it is a summation over observations (i.e. i ranges from 1 to m)
This is not generally true. The summation goes from 1 to $\min\{m, n\}$, where $n$ is the dimension of model space, and $m$ is the dimension of observation space. We will add these summation limits to the sums.

43. Line 479: "Naively, one may have expected that the dimension would, on the contrary, be reduced to $n - k$" – why? is this because the number of unknowns remains the same but the number of equations to be solved has increased by k?

In physics one usually learns about holomorphic constraints in theoretical mechanics, often by considering a point mass moving on a hypersurface. So, this is often the mental picture one invokes when dealing with constrained problems. For instance, a point mass in three-dimensional Euclidean space with a single holomorphic (i.e. strong) constraint can be pictured as moving on a two-dimensional surface. Thus this constraint reduces the dimension of the manifold on which the the point mass can move from three to two. One would therefore *naively* expect that one is now dealing with a two dimensional problem. The reason why this is naive is because a nonlinear constraint will correspond to a *curved* manifold. To characterise this manifold requires additional equations. Only if we have *linear* constraints, then the hypersurface is simply a tilted plane, which, by a suitable rotation-translation, can be brought into coincidence with, e.g., the xy plane. In

such cases, and only in such cases, can the dimension of the problem actually be reduced, as one would naively have expected.

44. Line 486: I would suggest the following change: "(Note that the covariance matrices and their inverses are symmetric, i.e., $R^T = R$, etc.)" becomes "Note that the covariance matrices and their inverses are symmetric (i.e. $R^T = R$, etc.)."
Agreed.

45. Appendix: For all unit and zero matrices (and vectors), I would suggest indicating the dimension as a sub-script.
Agreed. We will change this throughout the manuscript.

46. Line 498: I would suggest adding a subscript to clarify with respect to what the differentiation refers (i.e. replace $\nabla$ with $\nabla_\xi$).
Agreed.

47. Paragraph beginning line 515: how was this tuning done in practice?
As it is explained in the text. When the error variance is too large, one can see that the analysis is close to the unconstrained one. When it is too small, the analysis lies very close to the background estimate. One varies the variance until one obtains an analysis that departs from the background without drifting over to the (often noisy) unconstrained analysis.

48. Line 549: "It turns out that Eq. (D18) gives a relatively sharp transition from unconstrained to constrained model variables, while Eq. (D19) gives a very gentle transition" – this can be seen from the equations. I would suggest replacing the sentence with "It can be seen that Eq. (D18) gives a relatively sharp transition from unconstrained to constrained model variables, while Eq. (D19) gives a very gentle transition"
We will replace the text with "Equation (D18) can be expected to give a sharper transition from unconstrained to constrained model variables than Eq. (D19)."

Our tests, so far, showed that the differences between these approaches are not quite as dramatic as we expected.

We will replace the text after Eq. (D21) with " The test we performed, so far, showed that these different approaches often yield analysis results that are quite similar. However, in each approach the free parameters $\sigma_G$ and $c$ may assume different values. If they are not well tuned, then the analysis tends either toward the background estimate or the toward the unconstrained analysis."

49. Paragraph beginning line 567: I found that this went too fast and skipped a bit too much detail, after what was otherwise a very well-written paper that included a fair bit of theory. In particular, can you please explain in further detail the reduced matrices. The phrase "we are primarily interested in constraining the chemical components" was surprising, since I thought the authors were mainly interested in the aerosol components. What does it mean to "restrict ourselves to the chemical subspace"?

This seems to be a misunderstanding. What we mean by "chemical components" is "chemical components in the aerosol phase". Since our paper is exclusively concerned with aerosols, we thought that there was no risk of misunderstanding. Thus, by "chemical subspace" we mean "subspace of aerosol components". We will revise the text accordingly and replace all instances of "chemical components" by "aerosol components", and similarly for "chemical subspace". Also, we will revise the text in response to point 51 (see below).

50. Line 570: Full stop missing after $N_c$.
OK.

51. Paragraph beginning 574: similar to the above comment, I found that this skipped over too much detail. Please add further explanation. The authors state that in their present study, they use a Cholesky decomposition of the B-matrix. Is this what was used in Kahnert (2008), or is this described as the "spectral formula-

tion"? If it is different, it may be relevant to understand why the Cholesky decomposition was preferable to the author's previously presented methodology. This is mainly to understand the requirements and limitations of the proposed methodology.

We *are* using the spectral formulation for the minimisation of the cost function. However, we formulate the weak constraints in a subspace of physical space, as explained above. The Cholesky decomposition is only applied to the reduced B-matrix in the formulation of the weak constraints. We do not go into the details of spectral data assimilation, since these questions are rather specific to our particular implementation, while the paper is not restricted to spectral methods. However, we will rewrite this entire subsection and explain the reduced subspace approach in much more detail. We will also add a short footnote on how to incorporate this into the spectral formulation.

**3 Minor formatting issues**

1. References with parentheses inside parentheses: lines 33, 268, 269, 376
   This will be corrected.

2. Some of the in-line equations appeared to be missing spaces on one or both sides of the equals sign – this only appeared in the appendix. See lines: 391, 403, 404, 425, 515. I might just be imagining it. The paper was otherwise very well laid out.
   Our latex program seems to insert spaces when using the eqnarray environment, but not when using the equation environment. We trust that the copy editor will take care of this problem.

**4 References the authors may wish to consider**

- Qin, X. Measuring information content from observations for data assimilation: relative entropy versus shannon entropy difference. Tellus: Series A. 59, 2, 198-209, 2007.

- J. Joiner, A. M. da Silva. Efficient methods to assimilate remotely sensed data based on information content. Q. J. R. Meteorol, SOC. (1998), 124, pp. 1669-1694

- C Cardinali, S Pezzulli, E Andersson. Influence-matrix diagnostic of a data assimilation system. Q. J. R. Meteorol. Soc. (2004), 130, pp. 2767-2786. doi: 10.1256/qj.03.205

- C. Johnson, N. K. Nichols; B. J. Hoskins. Very large inverse problems in atmosphere and ocean modelling. Int. J. Numer. Meth. Fluids 2005; 47:759-771.

- M Bocquet, 2009: Toward Optimal Choices of Control Space Representation for Geophysical Data Assimilation. Mon. Wea. Rev., 137, 2331-2348, doi: 10.1175/2009MWR2789.1.

- F Rabier, N Fourrie, D Chafai, P Prunet. Channel selection methods for Infrared Atmospheric Sounding Interferometer radiances. Q. J. R. Meteorol. Soc. (2002), 128, pp. 1011-1027

- C Johnson, B. J. Hoskins, N. K. Nichols. A singular vector perspective of 4D-Var: Filtering and interpolation. Q. J. R. Meteorol. Soc. (2005), 131, pp. 1-19 doi: 10.1256/qj.03.231

Agreed; these will be added to and discussed in the introduction.

**References**

Burton, S. P., Chemyakin, E., Liu, X., Knobelspiesse, K., Stamnes, S., Sawamura, P., Moore, R. H., Hostetler, C. A., and Ferrare, R. A.: Information content and sensitivity of the $3\beta + 2\alpha$ lidar measurement system for aerosol microphysical retrievals, Atmos. Meas. Techniques, 9, 5555–5574, 2016.

Kahnert, M., Nousiainen, T., and Lindqvist, H.: Review: Model particles in atmospheric optics, J. Quant. Spectrosc. Radiat. Transfer, 146, 41–58, 2014.

Veselovskii, I., Kolgotin, A., Griaznov, V., Müller, D., Franke, K., and Whiteman, D. N.: Inversion of multiwavelength Raman lidar data for retrieval of bimodal aerosol size distribution, Appl. Opt., 43, 1180–1195, 2004.

Veselovskii, I., Kolgotin, A., Müller, D., and Whiteman, D. N.: Information content of multiwavelength lidar data with respect to microphysical particle properties derived from eigenvalue analysis, Appl. Opt., 44, 5292–5303, 2005.

---

## Author Comment (AC5) · 2 Feb 2017

May be you can check the papers of Wang at al. in ACP (2013, 2014a and b), where lidar assimilation is tested.
We thank Patrick Chazette for bringing these three papers, which he co-authored, to our attention. The results reported in these articles are very interesting. The paper by Wang, Sartelet, Bocquet, and Chazette (2013) is particularly impressive. It investigated assimilation of lidar and ground observations of PM10 and performed an observing system simulation experiment. The results demonstrate that a relatively small lidar

network can give analyses and forecasts of similar, and in some cases even higher accuracy than corresponding results obtained with an extensive network of ground stations, such as AirBase. This clearly demonstrates the potential of lidar observations. However, this study is only marginally relevant in the context of our paper, because it considers assimilation of lidar measurements for determining PM10, not for determining the concentrations of each aerosol component. It does not discuss the question of how to constrain the assimilation algorithm in order not to assimilate noise. For this reason, we do not feel compelled to add a citation to this article.

The paper by Wang, Sartelet, Bocquet, and Chazette (2014) presents a comparison of modelled and measured backscattering profiles, where the measurements were taken by a mobile lidar in the vicinity of Paris. The results of this comparison are highly encouraging. They also describe their assimilation methodology. If we understand it correctly, they set up the assimilation to correct PM10, and they distribute the analysis increment back to the various aerosol components in each size class according to the a priori distribution. In the context of our study, this is the most relevant fact in this paper, since it describes an ad hoc method for specifying constraints. Essentially, this approach seems to be based on the same idea as that described in Benedetti et al. (2009). However, we found that the explanations in the paper by Wang et al. (2014) were more detailed than in the paper by Benedetti et al. (2009). For this reason, we will add a citation to this paper.

Finally, the paper by Wang, Sartelet, Bocquet, Chazette, et al. (2014) presents a very impressive and comprehensive evaluation work of the potential of assimilating lidar measurements from the EARLINET network into an aerosol transport model. Since it is an application rather than methodology paper, we will not cite it here; but we will be sure to cite it when we have come that far and submit a paper on the operational evaluation of our lidar assimilation system.
**References**

Benedetti, A., Morcrette, M. J.-J., Boucher, O., Dethof, A., Engelen, R. J., Huneeus, M. F. H. F. N., Jones, L., andS. Kinne, J. W. K., Mangold, A., Razinger, M., Simmons, A. J., and Suttie, M.: Aerosol analysis and forecast in the European Centre for Medium-Range Weather Forecasts Integrated Forecast System: 2. Data assimilation, J. Geophys. Res., 114, D13 205, 2009.

---

## Author Comment (AC7) · 2 Feb 2017

Below the reviewer comments are marked in blue, our response is marked in black.

Summary
The inversion of aerosol optical properties into the aerosol chemical composition is a ill posed problem. The authors use information theory techniques to estimate the amount of information contained in LIDAR observations. They present different methods to make use of it as contains in a 3DVAR algorithm. This is meant to avoid assimilating noise inherent to the observations. To evaluate their constrain methods,

they create synthetic observations from CTM simulations and assimilate them back into the CTM.

Recommendation
The paper is well written and should be published. The methodology proposed is novel and can be applied to different observations within the variational assimilation framework.
We thank the reviewer for this positive evaluation of our paper.

Main comments
The authors choose to place all equations and their derivations into different appendixes. This hindered slightly the reading of sections 2.4, 3.1 and 3.2. However, the overall readability of the manuscript is improved by the focus on the description and evaluation of the method in the main text.
We agree. This point has been brought up by the other reviewers as well. We will follow the recommendations given by reviewer 2 and include the key equations with explanations in the main text, while providing the more detailed derivations in the appendix. This is a good compromise that will keep the paper accessible to non-theorists, while providing all the necessary details in the appendix for the interested readers.

Minor comments
Figure 1 is hard to read, specially the colour bar. Otherwise, the previous Referees have a number of valid suggestions for improvement, and I have nothing to add.
We will remove this figure in the revised manuscript. Since the paper is not concerned with those aspects specific to regional modelling, this regional plot conveys no useful information in the context of this paper.
* * *

---

## Author Response (AR1)

**Author response regarding the manuscript 'How much information do extinction and backscattering measurements contain about the chemical composition of atmospheric aerosol?'**

M. Kahnert[1,2] and E. Andersson[1]

[1]Swedish Meteorological and Hydrological institute (SMHI), SE-60176 Norrköping, Sweden
[1]Chalmers University of Technology, Department of Earth and Space Sciences, SE-41296 Göteborg, Sweden

*Correspondence to:* Michael Kahnert (michael.kahnert@smhi.se)

Dear Matthias, we were very happy to receive comments from reviewers who seemed to have many different backgrounds, ranging from data assimilation to lidar instrumentation and remote sensing. It was our hope and intention to write this manuscript in such a way that it would be interesting to a broad readership, which has not been quite easy. But the discussion showed that it is not impossible, and we received many good suggestions to better accomodate the expectations of each of these communities. Although most of the reviewer comments were rather straight forward to answer and implement, the sum of them (5 reviews plus one extra comment) amounted to quite substantial changes in the structure and content of the manuscript. We changed all figures, added two new ones, and removed one of the old figures. This also resulted in some changes in the abstract and conclusion section. Below we answer the comments by the reviewers and describe our changes in the manuscript. A manuscript version with the tracked changes is appended at the end of this document. Since this looks a bit messy, we also submitted (in an extra file) a clean version of the revised manuscript without any markings of the changes we did.

Below the reviewer comments are marked in blue, our response is marked in black.

**1 Reviewer 1**

The ACPD paper by Kahnert and Andersson deals with the assimilation of lidar observations into a chemical transport model. They investigate how much information about the chemical composition can be extracted from backscatter and extinction measurements and how this information is best assimilated into a chemical transport model.

Overall the paper is very well written and should be published as it is an interesting and important contribution to aerosol research. I only have a few minor comments which the authors may consider for their final paper. I have to say that my experience lies more on the lidar and aerosol optics side than on the information theory / mathematical side, thus I was not really able to review all theory details described in the appendix.

We very much appreciate receiving comments from the lidar and aerosol optics community. The parts that deal with theoretical developments and chemical data assimilation have been very well covered by reviewers 2 and 4. We thank the reviewer for his supportive review and helpful comments!

Comments:

1. It may be beneficial to say a few words about the refractive index and the size bins of the individual species of MATCH.
I suggest to add a table with the refractive index of these species at the lidar wavelengths.
We added a new table to Sect. 2 providing the refractive indices, and we added an itemised list of the size bins and the
corresponding size ranges.

2. Line 105-109: The description of the MATCH aerosol microphysics module could be shortened as it is not used in this
paper.
It is difficult to shorten these 5 lines. We could only remove them. Then again, we would like the reader to understand
that there do exist more realistic optics models, but they are not so straight forward to test, owing to their nonlinearity.
Thus the present study is meant as a first step in a larger project, which we will, hopefully, be able to follow up with an
investigation of information content based on a more sophisticated description of aerosol optics.

3. Line 118: What about the emissions of the other species? Are they also from EMEP?
Yes. But EMEP does not deliver gridded emission for black carbon and elemental carbon, only for total primary particu-
late matter. The sentence in question was meant to explain how we converted these into gridded emission data for black
carbon and elemental carbon. We reformulated this to make it clear that the emissions of *all* aerosol species are taken
from EMEP.

4. Line 134: "an" -> "and"
Yes.

5. Line 147 "we constrain to better than observation error": It is not clear to me what this means.
This formulation was also criticised by another reviewer. We reformulated this part as follows: "Suppose we have an
$n$ dimensional model space. Given $m$ observations (e.g., $m_1$ different parameters at $m_2$ different wavelengths, so that
$m_1 \cdot m_2 = m$), how many independent model variables $N \leq n$ can we constrain with the observations?"

6. Line 151: Remove "the".
OK. However, we reformulated this entire section to accommodate the comments by reviewer 2.

7. Line 177: "To be specific" could be removed.
Agreed.

8. Line 177: Do the results ($N_s$ and $H$) presented in this section depend on the order of the parameters? If yes, are the
changes significant?
We do not quite understand this question, especially not what the reviewer means by "order". Is the reviewer inquiring
about the *ordering* and *grouping*, or about the *magnitude*? In the latter case, the answer is no, because $N_s$ and $H$

are computed from the scaled Jacobian of the observation operator, which does not depend on the magnitude of the parameters. In the former case, the results do depend on which parameters are being measured, but, of course, not on the ordering.

9. Line 185: "around 7.4 for a single wavelength to around 10-12 for two wavelengths" would be more precise.

OK. This text has changed significantly in the revised version, owing to comments by reviewer 2, who asked us to consider different and technically more realistic combinations of observables in table 1.

10. Line 203: I was not aware about the difference between "observation error" and "measurement error". Is this generally accepted terminology? Maybe you can add a reference here so that the reader not familiar with this terminology can see that is used also elsewhere or was introduced by someone (maybe Rodgers?).

We added the formal definition for the observation error as $\epsilon_o = \epsilon_f + \epsilon_m$ and a reference to Rabier et al. (2002). They use the same terminology as we do, and they also denote the forward model error by $\epsilon_f$. However, they use the symbol $\epsilon_o$

for the measurement error, which is potentially confusing. We find it less confusing to denote the measurement error by

$\epsilon_m$, and to reserve the symbol $\epsilon_o$ for the observation error. We also mentioned that there can be other contributions to the observation error, such as representativity error. These concepts are well understood both in the data assimilation and in the satellite remote sensing/retrieval community, but not necessarily among instrument developers, who tend to identify

$\epsilon_o$ with $\epsilon_m$, while forgetting about $\epsilon_f$. This can be a serious mistake in cases where $\epsilon_f \gg \epsilon_m$, as is the case, e.g., in lidar depolarisation measurements. We find this point sufficiently significant to repeat it, in rephrased form, in appendix B.

11. Fig. 1: The difference between the middle and the right sub-plot is hardly visible. Perhaps you find a better way to visualize it.

We removed this figure. The regional model is merely used to generate a test case, but we do not address questions of regional modelling or horizontal information spreading in 3DVAR. Therefore this figure conveys no useful information for this study.

12. Line 229 (and at other places): You use $\beta_{\mathrm{sca}}$ and $\beta_{\mathrm{bak}}$ for the backscatter coefficient. Please use only a single symbol throughout the paper.

Yes, we corrected this and consistently use $\beta_{\mathrm{sca}}$.

13. Line 241 "the secondary inorganic aerosol (SIA) species are almost completely restored by the 3DVAR": Is it under- standable why exactly SIA is restored? Because of the refractive index? Or does it have something to do with the order (index number) of the species in the model?

This question has also been brought up by other reviewers. We added a new figure to Sect. 3 in which we show the linear coefficients in the transformation of the control variables in Eq. (C16) — see the new Fig. 5. Based on this extra figure we added a discussion of the question which aerosol components in model space make the dominant contribution to the signal-related variables in the transformed space. This facilitates the interpretation of the analysis results.

14. Line 274 "there appeared ...": This was not really shown in the paper, so you might remove this sentence or write it in a
different way.
OK, we removed this sentence.

15. Fig. 3: In this figure the difference between "observations" and "analysis" is much smaller than 10 % (the assumed "ob-
servation error"). As this is somewhat unexpected (but understandable as an optimization is applied) you may add a brief
discussion about the effect a "measurement error" (noise) would have. Because of the assumed linearity this probably is
not very difficult to explain.
OK, we added the following text. "In fact, the difference between the observation-equivalent analysis and the observa-
tions deviate by even less than 10 %. However, our tests confirmed that an increase in the observation error eventually
results in analysis results of which the observation-equivalent increasingly deviates from the observations (not shown)."

16. Fig. 4: Could it be of interest to see which aerosol species (size bins) the individual variables represent? What would be
the effect of changing the order of the species?
We added an extra figure (new Fig. 1) that shows a selection of aerosol species in specific size bins. In response to
reviewer 2 and 4, we even show a comparison with an unconstrained analysis. This makes it clearer that the constrained
analysis reduces the noisiness of the analysis, since it is being constrained to assimilating signal rather than measurement
noise.
We do not understand the last question about changing the "order of the species".

17. Line 277: "to be sure" could be removed.
Agreed, but we re-wrote the whole sentence.

18. Line 314: I think some aerosol species exist for which assuming externally mixed spheres is not that wrong.
It is unclear what kind of species the reviewer refers to. Certainly not dust or black carbon (BC). Sea salt is either mixed
with water, or else it is nonspherical. Organic carbon (OC) and secondary inorganic aerosols (SIA) are rarely found
in pure form. They are often mixed with each other, with water, NaCl, and even BC and dust. Even nucleation-mode
particles are often the result of at least binary nucleation involving more than one species. In our more realistic aerosol
microphysics model there is not a single size bin in which liquid-phase (i.e., spherical) aerosols consist of a single
compound. We therefore prefer to keep the text in its present form.

## 2  Reviewer 2

### 2.1  General comments

1. This paper details an interesting way to assess the information content in lidar measurements of aerosol backscatter and
extinction with respect to model assimilation. It also demonstrates how this knowledge may be used to optimize the
incorporation of lidar measurements in the model. This is a very interesting and relevant topic. Assimilation of lidar data into models is a field that is still developing rapidly, with a few different groups using very different techniques; therefore, well designed research into how best to use lidar data is very valuable. It is also potentially informative to the lidar community, since work must begin soon to design the next satellite lidar instruments if the lidar record is to continue. The choice of which measurements and which wavelengths to include has a large bearing on cost and technological difficulty, so having quantitative information about which measurements are most useful for improving models is critical. To that end, I would like to suggest some additional cases for Table 1, please see the specific comments below.

We thank the reviewer for the considerably thorough and supportive review, which very much helped us to improve various aspects of the manuscript. Our detailed response to the review comments follows.

2. The paper is well written with very nice clarity. However, the overall organization is somewhat difficult. The current organization consists of a very streamlined and easy-to-read main text with five very technically dense appendices. While the main text is pleasantly easy to read on the first pass, there is too much information missing. While it's appropriate to include extra, more detailed information in appendices, the main text still needs to be able to stand on its own, and in my opinion, it doesn't quite. I would suggest that the main equations and brief explanations should also be included in the main text, including all the equations that a reader would need to apply to calculate the kinds of results presented in this work. The appendices also include a lot of pedagogical development; this is the kind of information that I think rightly belongs in the appendix for readers who want more details. Since the appendices are 5 different topics, I also suggest that each appendix should be exist as a separate entity, with all variables defined, so that a reader can read Appendix D to learn about the application of constraints or Appendix E for the "practical aspects" without a close reading of Appendix A,B, and C, to find the definitions of the variables.

The organisation of the paper is indeed a delicate issue that was also brought up by other reviewers. Our main goal is to make this paper accessible to a broad community, including lidar instrument developers, remote sensing groups, and data assimilation researchers. For this reason, we prefer to include most of the theoretical developments in the appendix. However, we agree that this creates a significant problem by removing essential information from the main body of the paper. In the revised paper we followed the reviewer's suggestion and re-state the most essential theoretical results from the appendix in the main text. These changes were done mainly in Sect. 2. This makes the paper more readable and self-contained, while avoiding the risk of making it too technical, which could narrow down the readership of this work.

3. The results and conclusions are also a little too abbreviated. Some key aspects are missing, like how was the specific weighting chosen and how do we know this is the best weighting? Also, as pointed out by another reviewer, the assessment (section 3.2) is really more of a demonstration. That is, although the theoretical development is compelling, the application/assessment section isn't sufficient to convince readers that this is a better way to assimilate lidar data than another way. This paper clearly reflects a lot of research on the part of the authors and I think the missing information probably exists but was left out in the effort to streamline the manuscript. I think adding this additional information should be fairly straightforward and would improve the usefulness of this research for the modeling and lidar communi- ties without adding too much complication to the nice flow of the paper.

This is also an important point, which was brought up by several reviewers. We performed additional computations using the unconstrained assimilation algorithm and compared the constrained to the unconstrained analysis. The hypothesis is that the constrained analysis is less noisy, because the unconstrained analysis is at risk of assimilating noise. The results of this comparison, which are shown in the new Figs. 1, 2, and 4, are consistent with the hypothesis. Also, we eliminated all instances of "numerical experiment" and replace it by a more appropriate term, e.g., "numerical test",

"demonstration", or "illustration". Further, we added more explanations to Sect. 2.4 about the construction of the covari- ance matrix in the constraint term. Finally, we amended the conclusion section and the abstract to incorporate the results of the comparison of the unconstrained and the constrained analysis algorithm.

**2.2  Specific comments**

1. Lines 151-158: Here is an example where I think some important things are missing from the main text which only appear in the appendices. These eight short lines are the methodology section for the key calculations that are the novel part of your research and are critically important for a reader to understand. I suggest that a way to decide what should also be included here would be to target the subset of equations that a reader would need to apply to calculate results like yours, but without their derivations. Also include enough supporting explanation to describe what the equations say and how to use them.

We agree, and we made changes following the more detailed suggestions given in the following comments.

2. L152-153: Specifically here, Eq C6 and C16 should be included in the text, since they are required to understand the meaning of the sentence. Later, at L159-160 where readers are directed to the appendix for more background information,

I think that's fine.

OK, we revised the text and included the equations (with explanations) for the observation operator, the observability matrix, and the singular-value decomposition thereof. The changes pertain to Sect. 2.

3. L155-157: The equations for signal degrees of freedom and Shannon information content should also be included in the text.

OK, this has been added with accompanying text to Sect. 2.

4. L165: "a numerical experiment". In fact, it's more of a demonstration than an experiment. It's useful as a demonstration of the results of the technique, but there's nothing in the demonstration that addresses a hypotheses. Sharing more of the background work would make the paper more compelling. For example, as another reviewer suggested, comparing to a control experiment would be necessary for convincing readers that this technique is useful. For another example, a pair of runs with different weightings in the assimilation would help answer the question of why the weighting that was ultimately chosen was the best one.

We replaced "numerical experiment" everywhere in the paper, as mentioned previously. Next, we showed a control run with the unconstrained assimilation system (new Figs. 1, 2, and 4). The hypothesis is that the constrained analysis should be less noisy than the unconstrained analysis. We revised the Figures and show both the unconstrained and the constrained analysis. Also, we added an extra figure (new Fig. 1) to show both analysis results for different aerosol species in different size bins, as these are even more sensitive than size-integrated total mass mixing ratios. Finally, the case we picked in the original manuscript was not particularly challenging, since the background state was fairly close to the reference state. In the revised paper, we picked a more challenging case in order to make the differences between both analysis runs as clear as possible. As for the different weightings, our tests, so far, indicate that the different approaches result in rather similar analysis results. So, the constrained analysis is not as strongly dependent on the weighting as one may expect. We clarified this point in a discussion added at the end of Sect. 2.4.

5. L177: Depolarization is not included in the studied parameters, yet lidar studies have shown that depolarization measurements contain some information about aerosol composition (for example, Omar et al. 2009 as referenced in the introduction, but there are many others). Do the authors have any comment on depolarization and why it isn't included in this study?

There are two major problems. The obvious practical problem is that the forward model would need to be based on nonspherical particles (as spherical particles do not depolarise). However, our simpler optics model is entirely based on spherical particles, while our newer optics model only accounts for the nonsphericity of bare black carbon, but not for that of mineral dust or dry sea salt. Thus our capabilities of modelling depolarisation are presently limited. The second problem is that the observation error for depolarisation may be very high, even though the measurement error is very low. This is because the forward-model error is likely to be quite high, since even slight variations in particle geometry (e.g. Kahnert et al. (2012)) or inhomogeneity (e.g. Kahnert (2015)) can result in large variations in the depolarisation ratio. If the forward-model error is, indeed, high, then the prospects of using depolarisation for constraining CTM model results are likely to be low. However, this question is open and will be investigated in future studies. But in order to do so, one would first need to obtain estimates of the forward-model error (e.g, by computing depolarisation ratios while varying particle morphology).

6. Table 1 and related discussion: From a lidar standpoint, some combinations of channels are more technologically affordable than others, so the discussion of which channels add significant information content is very interesting. However, the utility for the lidar community would be maximized if the combinations were ordered such that they roughly increase in technological difficulty. Also, some combinations don't really make sense from a technological standpoint. There is no lidar that measures extinction but not backscatter at the same channel (although modelers may use only the extinction). On the other hand, backscatter (actually attenuated backscatter) without a direct measurement of extinction is common. Also, since CALIPSO, CATS, EarthCARE and the $3\beta + 2\alpha$ combination of airborne HSRL2 are mentioned in the introduction and motivation sections, it would be useful if the combinations relevant to those instruments were included. CALIPSO = CATS = $\beta(\lambda 1) + \beta(\lambda 2)$. EarthCARE = $\beta(\lambda 3) + k(\lambda 3)$. HSRL2 = $\beta(\lambda 1) + \beta(\lambda 2) + \beta(\lambda 3) + k(\lambda 2) + k(\lambda 3)$. I would suggest these combinations of backscatter and extinction would be most interesting and useful to the lidar community: $\beta(\lambda 3)$

219 $\beta(\lambda 1) + \beta(\lambda 2)$

220 $\beta(\lambda 1) + \beta(\lambda 2) + \beta(\lambda 3)$

221 $\beta(\lambda 3) + k(\lambda 3)$

222 $\beta(\lambda 1) + \beta(\lambda 2) + k(\lambda 2)$

223 $\beta(\lambda 1) + \beta(\lambda 2) + \beta(\lambda 3) + k(\lambda 2) + k(\lambda 3)$

For these experiments, it appears that the observation error was always assumed to be the same in every channel. I think
it's a reasonable assumption, to first approximation, that the measurement error would be similar in every channel, but
as pointed out at L78-79, some lidar retrievals include additional non-random errors that can be much larger. This could
and should affect the choice of channels to assimilate. For example, the Raman, HSRL, and transmittance techniques are
fairly direct measures of extinction, but techniques that require an inferred lidar ratio to convert backscatter to extinction
have relatively little additional measurement information content in the extinction.

We welcome the reviewer's suggestion to take technical realisations of lidar systems into account, and we revised Tables
1 and 2 (i.e., Tables 2 and 3 in the revised manuscript) according to the reviewer's specific suggestions. We also added
a comment on the observation errors of lidar measurements, specifically on the fact that the observation errors may be
different for different channels/parameters.

7. L 197-201. Here also the discussion of incorporating soft constraints and the specifics of the three weighting schemes
should be in the main text of the paper and not just the appendix, since it is discussed here in the results section. This
section is not understandable without the equations from the appendix and most of section D3.

We removed this discussion here. Instead, we briefly discussed the construction of the constraint covariance matrix in
Sect. 2.4.

8. L 203-204. Discussion of observation error vs. measurement error. This is interesting and useful, but could be clarified
as to whether the forward model error (due to poor assumptions) is considered part of the observation error or is another
separate source of error. If it is part of the observation error, how are the forward model errors represented and how are
they transformed into the space of the measurement vector?

We extended the text to clarify that the observation error is given by $\epsilon_o = \epsilon_m + \epsilon_f$, where $\epsilon_f$ denotes the forward-model
error. We also added a citation to the paper by Rabier et al. (2002) with a hint to their Eq. (1), which explains this
terminology. A way to determine the forward-model errors theoretically is to perform light-scattering calculations while
varying various parameters, such as particle morphology, refractive index, and size distribution within typical uncertainty
ranges. This can provide us with an estimate of $\epsilon_f$. To the best of our knowledge, it would be very difficult to determine
$\epsilon_f$ with experimental methods.

We are not sure if we understand the last question. $\epsilon_f$ enters into the definition of the observation error covariance matrix,
i.e. $\mathbf{R} = \langle \epsilon_o \cdot \epsilon_o^T \rangle$, which is a matrix in the space of the measurement vector. No further transformation is necessary.

9. L 207 While there may be retrieval errors in the lidar backscatter and extinction due to assumptions, assumptions on
particle shape and size distribution are not among the assumptions used in lidar retrievals. These examples belong only to the optics model (forward model). So, perhaps delete "also". Poor assumptions in the optics model or in lidar retrievals would presumably lead to bias errors, whereas measurement errors would more typically be random. Does this make a difference in the analysis?

OK, we deleted "also". We would generally not be sure if assumptions in the optics model necessarily (mainly) lead to biases. For instance, model errors may be dependent on size and morphology of the actual particles. The errors would, correspondingly, fluctuate over time as the aerosol size and composition changes over time. The amplitude of this fluctuation may well be larger than any possible biases. However, in case that the forward-model does introduce a large bias, this would, indeed, be a problem, since analysis algorithms are typically based on the assumption that the errors are unbiased.

10. L 219. I strongly agree that estimating the uncertainties in the optics model is very important. Some discussion here seems warranted about how that can be done. Later I see that this is discussed in the summary (L281 – 292) but I think it would be better if it comes up first here in the discussion section.

Agreed. We added an explanation here, but we also mentioned it again in the conclusion section.

11. L 256 and caption to Fig 4. In both places, it would be kind to remind readers that the delta notation in $\delta x'$ means this is the difference between the value and the background value.

It is not so simple. $\delta x$ in physical space is the difference between the value and the background, while $\delta x'$ is obtained from $\delta x$ by applying the transformation $\delta \boldsymbol{x}' = \mathbf{V}_R^T \cdot \mathbf{B}^{-1/2} \cdot \delta \boldsymbol{x}$. We repeated this definition in the text with a reference to the definition (which is now found both in the main text and the appendix), and we added a reference to the defining equation both at this point in the text and in the caption to the figure. But we think it would be a bit overdone to repeat the equation in the figure caption.

12. L 259-263. The choice of D21 with its sharp drop-off in weighting appears to mean that only one transformed variable is allowed to change in a meaningful way, although the measurement scenario chosen has nearly the maximum amount of information content available, close to DOF=4. Why was D21 chosen instead of D18, which would allow the measurements to play a bigger role? The only discussion of this choice is the rather vague comment in the Appendix "it is a matter of experience to test different approaches and select the one that proves to be most suited". How and why was this approach determined to be the most suited?

We have done some additional tests and found, in fact, that the analysis is less sensitive to the choice of weighting than we expected. We explain this in the revised paper in Sect. 2.4. Also, we did the following changes to Fig. 4. First, we show $\delta x'$ for both the constrained and the unconstrained analysis. Thus the whole discussion of the figure shifts from a mere description of the behaviour of the constrained analysis to a comparative discussion. This makes it much clearer what kind of effects the weak constraints have on the analysis increments. Second, following a suggestion by reviewer 4, we show not all 20 panels, but only a subset of panels sufficient to illustrate the different behaviour of signal- and noise-related (phase-space) model variables. Third, as mentioned earlier, we picked a more challenging case in which the reference and background results differ more strongly than in the case we originally picked. So this figure has changed
considerably, and the accompanying discussion has become a lot more informative.

13. Comparison of Figure 3 and Figure 2, if I understand right, underscores the fact that there is a significant null space,
not controlled by the measurements, since essentially the same measurements in Fig 3 correspond to both the black and
red lines in Fig 2. What is not clear to me is what happens in a standard assimilation to the variables that are not well
controlled by the measurements? Do they remain close to the background values, or do they vary wildly and arbitrarily?
If the former, then the exercise of determining the singular values wouldn't help the assimilation very much (but would
still be useful in terms of building knowledge about what we can and can't actually measure). On the other hand, if
a standard assimilation arbitrarily varies state variables in the null space, then this is a very important motivation for
this technique (and maybe that motivation could be emphasized a little bit more in the introduction and conclusions).
Not being very familiar with the field of model assimilation, I guess but don't actually know that there must be other
"regularization" techniques in use to prevent an assimilation from arbitrarily varying parameters that are mostly in the
null space of the observations, although I imagine existing techniques may be more ad hoc than the method presented
here. Can you comment on other methods and demonstrate how this method performs better than other methods?
The reviewer's comment about the null space and the behaviour of the unconstrained (standard) assimilation raises an
important issue. As mentioned earlier, we have now run an additional unconstrained assimilation, and we show a com-
parison of both methods. Figure 2 has been replaced by two figures. The new Fig. 2, similarly to the old figure 2, shows
the total mass concentration of different aerosol species, but now for both the constrained and the unconstrained analysis.
The new Fig. 1 shows a similar comparison of a selection of aerosol species in specific size bins. This comparison illus-
trates that the unconstrained analysis yields more erratically varying vertical profiles (i.e., results that vary more wildly
in the null-space).

As for ad hoc methods, we did review previously reported approaches in the introduction, such as the one by Benedetti et al.
(2009) (L 53-54) based on constraining the total aerosol mass mixing ratio, and the one by Saide et al. (2013) (L 55-56)
based on constraining the mass mixing ratio per size bin. One obvious disadvantage is that these approaches are quite
inflexible. The number of constraints is fixed in these methods, so one cannot easily adapt the number of constraints to
the number of independent measurements to be assimilated, as we can in our approach. (In fact, our method automatises
this process.) Also, the available information may not be optimally exploited by these methods (L 57-59). We have not
tested such methods, so we cannot comment on their performance. However, we also believe that the burden of proof for
such a demonstration does not lie with us. We are employing a mathematically well-founded approach based on infor-
mation theory. If other groups choose to not follow us, but continue to use ad hoc methods (which, admittedly, may be
quite attractive owing to their simplicity), then it is up to them to demonstrate that such ad hoc methods yield sufficiently
accurate results while exploiting the available measurement information. Owing to the ad hoc nature of these methods,
such a demonstration would have to be repeated for any new set of measurements to be assimilated. Our method can
serve as a reference for such tests.

14. L 298-299. "It also appeared". This result is disappointingly empirical for such a well-founded theoretical study. This observation that SIC was most faithfully retrieved was made in a single case– would you expect this result to be general for all cases, and why? Answering the question is complicated since the singular variables are defined only in the transformed space and therefore the information about what variables are or are not constrained by the measurements is only in this transformed space, not the state space. Yet this statement highlights that it's desirable to have information about which chemical species and size bins are constrained by the measurements. Is there any way to provide information about this quantitatively? For example, since each state variable is a linear combination of the transformed variables, would showing the linear coefficients in a table make it more obvious which state variables are most closely related to the most significant transformed variables? Perhaps there is a way to use the coefficients to calculate a "fractional significance" that would indicate that x% of the variability in a given state parameter is orthogonal with significant transformed variables while (1-x)% is orthogonal with insignificant variables?

This is a very good suggestion. We added an extra figure (new Fig. 5) with accompanying discussion and show the magnitude of the linear coefficients for the signal-related control variables. However, the coefficients depend on the B- and R-matrices, which vary spatially. So, we cannot draw very general conclusions from a single test case. But we do think that this discussion helps the reader to understand why the analysis behaves the way it does in our specific case.

**2.3 Minor comments**

1. L37: Muller et al. 1999 and Veselovskii et al. 2002 and related papers (there are many) would be more relevant references here since they detail retrievals of refractive index, etc., from lidar. (Mishchenko et al. 2007 is an introduction to the Glory satellite and was about retrievals from a polarimeter.)

    Müller, D., U. Wandinger, and A. Ansmann (1999), Microphysical particle parameters from extinction and backscatter lidar data by inversion with regularization: theory, Appl Optics, 38(12), 2346-2357, doi: 10.1364/AO.38.002346.

    Veselovskii, I., A. Kolgotin, V. Griaznov, D. Müller, U. Wandinger, and D. N. White- man (2002), Inversion with regularization for the retrieval of tropospheric aerosol parameters from multiwavelength lidar sounding, Appl Optics, 41(18), 3685-3699, doi: 10.1364/AO.41.003685.

    Agreed. The references have been replaced.

2. L99: I infer that the ratios in the different size bins are fixed, or else there would be much more than 20 total variables. Is there a way to concisely clarify this in the sentence?

    We are not sure what the reviewer means by the "ratios in the different size bins". The concentration ratios are certainly not fixed; they can change from one grid cell to the next. The size ranges are fixed. The latter point should now be clear, since we explicitly list the size rages in Sect. 2.1 of the revised manuscript.

3. L109: maybe replace "in the present setup" with "currently in that version". "The present setup" seems to refer to "the setup used in the present study" but that is misleading, since the present study uses the 20-variable version of the model.

    Agreed.

4. L134: "an" should be "and"

   Yes.

5. L142: "Error correlations ::: are not assumed to be separable". I'm not sure what this means. What is (or is not) separable from what?

   Vertical and horizontal correlations are often assumed to be separable. We do not make such assumptions, because vertical correlations are often stronger on larger horizontal length scales. In our spectral model (where the horizontal correlations are Fourier-transformed) this means that vertical correlations are larger for smaller horizontal wavenumbers. Since this is not so essential in the context of this study (and potentially confusing), we removed this text in L 142.

6. L153: "see Eq. D16". Should this be C16?

   Yes. However, following earlier suggestions by the reviewer, this text has been revised and supplied with the main equations from the appendix. So the text in its present form has been replaced.

7. L162-164: Should this sentence perhaps be part of section 2.4, as part of the description of the new technique? The rest of this paragraph (L164-174) is more about the demonstration of the new technique and so seems like a somewhat distinct topic.

   Agreed, we have moved this text.

8. Figure 1: The caption says "note the nonlinear colour scale" Actually, the scale is hardly visible. Please expand the axis labels so they are a similar text size to the caption text.

   Actually, we think that this figure is not particularly relevant in the context of our study, since we do not consider aspects of regional modelling or horizontal information spreading in the analysis. It merely shows one out of many model variables in a single model layer, which does not convey much useful information. Also, since we consider a single profile, the analysis impacts the mass mixing ratio only at and around the observation site, which is difficult to see in a regional plot. We therefore removed this figure in the revised manuscript.

9. Figure 2: The axis labels' and inset box labels' font size should also be increased here.

   OK, we increased the font size in all figures wherever it was necessary and possible.

10. L 391. The variable n is not defined. Possibly this is the only case, but I would also request that variables be re-defined frequently when used in key equations. If a reader is directed from another part of the paper to Equation D18 or C12, for example, then it would be nice if all the information relevant to understanding that equation is given immediately after that equation, rather than having to scroll through 8 or 10 pages to relocate the definitions of key variables.

    Agreed, we have added the definition of n. Also, the problem with directing the reader to equations in the appendix is now significantly alleviated in the revised versions, since we re-stated the key equations in the main body of the paper (see our response to an earlier comment).

11. L563. The symbol lambda is used for wavelength elsewhere in the text. You might consider using a different symbol here.

OK, we have replace it by mu.

**3   Reviewer 3**

The line-number references of the reviewer seem to be offset relative to those given in the online pdf of our manuscript. But we think that we figured out each point in the text the reviewer referred to.

1. It is well known, that the problem of inversion of standard $3\beta + 2\alpha$ lidar measurements to the particle microphysics is undetermined and to constrain it, numerous techniques were considered. The authors suggest an interesting approach to assimilation of lidar measurements into chemical transport model. It looks like a promising concept to extract the information about particle parameters from lidar measurements. Paper is very well written and should be published.

   We thank the reviewer for his positive evaluation of our manuscript and for his helpful comments.

2. The structure may be questionable, because a half of material is put in appendices. These appendices are clearly written and are definitely useful for unprepared reader. I personally, had no problems with material structure.

   We agree that the structure was not optimal for all types of readers. We found that the compromise suggested by reviewer 2 adequately addresses these concerns. We refer to our detailed response to reviewer 2, which explains the changes we implemented in the revisions.

3. Additional references to the previous studies of lidar data inversion would be desirable , and other Referees have already suggested several.

   Agreed. We added a paragraph in the introduction with a brief discussion of other studies, also from numerical weather prediction data assimilation.

4. Stability of retrieval strongly depends on aerosol type. It is more challenging for aerosols with dominant coarse mode and for particles with strong absorption. The authors consider only one example (not the most challenging) in their simulation, so it is not very clear how the approach will work for other aerosol types. But this may be a subject of separate study.

   Yes. Although it is not the subject of this paper to comprehensively test all sorts of mixed aerosol populations, we do agree that the case we picked was a little bit too easy. This is mostly because the background and reference cases were very close to each other. In such a case one does not see very clear differences between a constrained and unconstrained analysis. In the revised paper we have picked a more challenging case, and we now show our test results for both the constrained and the unconstrained 3DVAR algorithm. This helps to better illustrate what practical significance the constraints can have.

**4 Reviewer 4**

**Summary**

The authors consider the case of assimilation of remote-sensing data (specifically aerosol extinction and backscattering coeffi-cients) applied to aerosols fields within a chemical transport model. They describe how an additional term can be added to the 3D-var cost-function so that the assimilation adjusts only those components (in a transformed space) for which the observations provide information. The additional term relies on the singular value decomposition of the scaled observation operator. In this way, the assimilation automates the choice of control variables in an otherwise highly under-constrained inverse problem.

**Verdict**

The paper is very well written and is surprisingly clear, given the subject matter. The manuscript introduces a potentially very powerful concept for variable selection into the field of aerosol data assimilation. The authors have probed the idea in a minimal test case, which assists in understanding the effects. I found that the shortcomings of the paper were relatively minor. I felt there was insufficient discussion of the literature of related treatment. I was unsure about whether the organisation of the material was optimal (see the "Main comments"). Finally, a counter-experiment without the addition of the new constraint in the 4D-var cost function was, in my opinion, lacking. All in all, I believe that the paper should be published, pending the minor revisions suggested below.

We are grateful for this encouraging assessment of our work, as well as for the insightful comments and suggestions. It is obvious that the reviewer has devoted considerable time into studying the manuscript and providing constructive criticism on various aspects of the content and organisation of the paper. Our detailed response to these comments follows.

**4.1 Main comments**

1. There was little or no discussion of literature on related treatments. I have not the time to read all of these myself, however I have included a list at the end of articles that may be relevant, for example those that deal with information content of observations in data assimilation or those that refer to the singular value decomposition of the observability matrix.

   We have, indeed, only cited studies on aerosol data assimilation. Most of the studies cited by the reviewer are concerned with numerical weather prediction (NWP). We have added a paragraph to the introduction to discuss related NWP studies and include the citations suggested by the reviewer.

2. I believe that a small counter-experiment was lacking. In the results presented in section 3.2, I would suggest also presenting results for the assimilation experiment which did not include the additional constraint in the 3D-var cost-function.

   This is a very valid point that was also brought up by other reviewers. We have included these results and revised the figures and discussion accordingly. In particular, we replaced Fig. 2 by two new figures. The new Fig. 2 shows, similarly to the old Fig. 2, the total mass concentration of different aerosol species, but now for both the constrained and the unconstrained analysis. The new Fig. 1 shows a similar comparison of a selection of aerosol species in specific size bins.

This comparison illustrates that the unconstrained analysis yields more erratically varying vertical profiles (i.e., results that vary more wildly in the null-space).

3. I was unsure whether the organisation of the material was optimal - I highlight this as an issue that the editor may wish to take up. The introduction concludes by urging the reader to read the Appendix before proceeding onto the rest of the methods and results section. Much of the interesting methodology is contained within the Appendix, and we agree that it would be difficult to make sense of the main part of the paper without a good understanding of the contents of the Appendix. As such, I would suggest incorporating the Appendix into the main body of the text. At one level, this is really a matter of taste, and thus I leave it to the editor.

This is a tricky point. We put some thought into this before writing the paper, and we concluded that the appendix is, indeed, most interesting for readers who are mainly interested in data assimilation methodology, and for those who are very eager to learn something about it. But other readers, e.g. lidar instrument developers, will most likely be deterred from reading the paper if we merge the entire appendix with the main body of the paper. However, the reviewer's criticism is very valid, and it has been brought up by several reviewers. We believe that reviewer 2 has suggested a very good compromise, namely, to state and explain the main results (equations) from the appendix in the methodology section of the paper, while retaining the derivations and more detailed explanations in the appendix. This alleviates the problem that parts of the main text are hard to understand without the information given in the appendix. At the same time, we avoid the risk of making the paper inaccessible (or just too boring) for those who do not mainly work with data assimilation methodology.

We therefore followed the suggestions of reviewer 2 in this point. It seems to us that this also adequately addresses the main point of criticism brought up by reviewer 4.

**4.2 Minor comments**

1. When describing observation errors, there was no reference to the component from "representativity errors" (i.e. measurements are made at a point, or over a small area in the case of remote sensing, while model grid-boxes are typically in the order of kilometres across in the horizontal dimensions). All of the discussion about observation errors was in terms of the measurement error and errors in the observation operator, both of which are relevant. However the representativity component is not insignificant in many contexts.

Agreed. We have added a discussion of the representativity error in the text accompanying table 2, where we made it clear that in this numerical test we have neglected this source of error.

2. Observation standard deviation was reported in percentage, but it was unclear what this was a percentage of. Please clarify.

It is a percentage of the observed backscattering coefficient or extinction coefficient. We changed the text in Sect. 2.5 from "We assumed an observation error standard deviation of 10 %" to "We assumed that the observation error standard deviation is 10 % of the measurement value."

3. I would suggest replacing all instances of the term "costfunction" with "cost function" (or "cost-function"). The latter is about 15 times more common (on the web, at least). Similarly, I believe that the compound word "nullmatrix" is used in German (capitalised, that is) whereas it is "null matrix" (or "zero matrix") in English.

Agreed.

4. I could not find a definition to the term "signal degrees of freedom". Please include this somewhere (preferably at first usage, or in the Appendix).

We have added a detailed explanation of the terminology to Sect. 2.4. Following the suggestions of reviewer 2, we have also provided key equations of the appendix with explanations in the main text. This also applies to Eq. C12, which is now provided in the methodology section. Thus the explanation and definition of the term "signal degrees of freedom" now appears much earlier in the revised paper. (Note that in the remote sensing community the number of signal degrees of freedom is also known as the "effective rank" of the problem.)

5. Line 48: Please replace "This is a rather bold approach that largely disregards ..." with "This is approach largely disregards ..." – please use argument rather than rhetoric to explain what is wrong with the work of others.

Agreed.

6. Line 54: The reference to Kahnert (2009) is used to show that several optical properties at multiple wavelengths may allow constraining more than just the total mass concentration. Surely other authors have looked into this. If so, please summarise other work done. If not, please say so.

We did cite the study by Burton et al. (2016) (L64), although we did so in the introduction. We now have added two more references that analyse the information content of lidar observations, namely, the papers by Veselovskii et al. (2004) and Veselovskii et al. (2005). However, these papers analyse the information content with respect to particle size and refractive index, not with respect to chemical composition. Therefore, we put these citations into the introduction.

7. Line 98: "... using 40 eta-layers with variable thickness depending on the under- lying topography" – do you just mean that this is a terrain-following coordinate? Or is there something more sophisticated about this?

OK, we have replaced this with "using 40 terrain-following coordinates".

8. Line 125: "The background error covariance matrix of the model a priori is modelled with the NMC method ..." . I checked the reference (Kahnert, 2008), in which I believe this is described. If the implementation is the same here as in the 2008 article, then I believe that it is best to say that it "follows similar principles to the NMC method" or "is inspired by the NMC method". If it is indeed the NMC method, the authors should clarify the difference to methodology laid out in Kahnert (2008).

OK, we have replaced this with "follows similar principles to the NMC method".

9. Line 129: I would suggest replacing "Given m observations of, e.g., m1 different parameters at m2 different wavelengths, so that m1 m2=m, how many..." with "Given m observations (e.g., m1 different parameters at m2 different wavelengths, so that m1 m2=m), how many...”

Agreed.

10. Line 130: “... we can constrain to better than observation error” – do you mean “model error”? If not, please explain that the transformation makes the (rescaled) observation errors and (rescaled) model variables comparable.

This was a bit confusing. We have reformulated this sentence, and we have addd a more detailed explanation of the terminology *signal degrees of freedom*.

11. Line 134: Please replace “... a singular value decomposition of the Jacobian of the observation operator ...” with “... a singular value decomposition of the Jacobian of the scaled observation operator ...” or something similar. By the way, this scaled observation operator appears to have a name: “the observability matrix”

Yes. Actually, this text has been extended with a lot more explanations, and it now provided the main equations from the appendix. We have followed the reviewer's suggestion and introduced the term *observability matrix* for the scaled

Jacobian.

12. Footnote 2, page 5: I found this distinction a bit cryptic. Please consider rephrasing.

There seem to be two fractions in the community. One that uses *data analysis* and *data assimilation* almost interchange- ably, and another that insist on keeping these two concepts apart. We are mostly guilty of belonging to the first one, but we do not want to make a big deal out of mere questions of terminology (which is why we put this into a footnote rather than into the main text). However, we did our best and clarified the text as best as we could.

13. Line 150: I realise that this is something that is clarified later on, but I would suggest saying a few words at this point about the synthetic observations; namely, what kind of observations they were and how many observation points there were.

Agreed; we have added this information in the revised manuscript.

14. Line 154: I would suggest the following change “thus providing nearly perfect observations. (We assumed an observation error standard deviation of 10 %) The only ...” becomes “thus providing nearly perfect observations (we assumed an observation error standard deviation of 10 %). The only...”. See also my comment about about describing the units for the observation error standard deviation.

Agreed (replacing “observations. (We” by “observations (we”. In addition, in response to an earlier request to be more specific what me mean by “10 %” (percent of what?), we have replaced the text in parenthesis with “(we assumed that the observation error standard deviation is 10 % of the measurement value)”.

15. Line 162: What is “Nd:YAG”? Please clarify. I suspect that this is some error with the bibliography manager.

It is no error. “Nd:YAG” is the standard abbreviation for “neodymium-doped yttrium aluminium garnet” laser, one of the most commonly used solid-state lasers in remote sensing. We have now added this information at the first instance, which is in the introduction section.

16. Line 168: I would suggest the following change: "... two wavelengths. (Compare, e.g., cases 1., 2., and 3. to cases 4., 5., and 6.) Hence ..." becomes "... two wavelengths (compare, e.g., cases 1., 2., and 3. to cases 4., 5., and 6.). Hence ..."

Agreed. However, reviewer 2 has suggested to replace the cases considered in Table 1 with different cases that are more closely associated to combinations of wavelengths and parameters that are technologically feasible and common. Thus the text accompanying Table 1 (now Table 2 in the revised manuscript) has changed considerably.

17. Line 171: a missing full stop after the right parenthesis.

Agreed.

18. Table 1, caption: the "Nd:YAG" term appears again.

See our earlier response.

19. Line 181: I believe that "weak constrains" should be "weak constraints".

Yes.

20. Line 189: See my comment above about the representativity component to the observation error.

Agreed, see our earlier response.

21. Figure 1: I think it would be interesting to see the increment as an additional panel in this figure.

Figure 1 has been criticised by several reviewers. In fact, this figure is not particularly useful in the context of our paper. We are not discussing any aspects of regional modelling or horizontal information spreading in the assimilation algorithm. The model merely serves us to provide us with a test case. So, we have removed this figure in the revised manuscript (see also our response to reviewer 2).

22. Figure 1: The text on the scale is a bit too small. I would suggest having one scale, rather than three, and enlarging the scale so that the labels can be read.

See the previous item.

23. Figure 2: The units appear to be "mixing ratio [ppb-m]". Do you mean mass mixing ratio? Please clarify.

Yes. This has been corrected.

24. Figure 4: Do we need all panels? Why not just show the first three or four, and then a selection of the remaining terms.

Agreed, we now show 10 instead of 20 panels. Following the comment by reviewer 2, we have run a $3\alpha + 2\beta$ test case, in which case we have 5 signal degrees of freedom. Thus we now show the first 5 signal-related transformed increments, and 5 out of the 15 noise-related increments (Fig. 4).

25. Line 262: I would suggest the following change " ... dramatic decrease in both the entropy and signal degrees of freedom ..." becomes " ... dramatic decrease in both the entropy-change and signal degrees of freedom ..."

Agreed; however, owing to the changes in Sect. 3 this part of the text in the conclusions has now also changed.

26. Line 282: "It also appeared that among the original model variables, secondary inorganic aerosol components were most faithfully retrieved by the inverse modelling solution" – why is this? why SIA? Do they have specific optical properties to make them more observable by such LIDAR pseudo-observations?

This question has been brought up by several reviewers. We follow the suggestion of reviewer 2 and add an analysis of the linear coefficients that transform the elements in model space to the signal-related control variables. We have added a new figure (Fig. 5) and a discussion — see our detailed response to reviewer 2.

27. Line 293: I would suggest the following change: "The present study should be extended..." becomes "The present study could be extended..."

Agreed.

28. Line 295: I believe that the expression "highly underrated" is somewhat dramatic and relatively colloquial, and does not fit with the tone in the rest of the paper. The authors are encouraged to use argument rather than rhetoric to make their point.

OK, we have replaced the text with "Another important issue concerns the choice of ...".

29. Line 297: Regarding the statement "There is little one can put forward in defence of this model other than pure convenience". Some justification is required (e.g. some references) to demonstrate why this model is untenable. There's a saying (attributed to George Box) "All models are wrong, some models are useful". Does this model give significantly worse results than representations, or is it just inaccurate in its assumptions?

Worst of all, this model is rather unpredictable, since its accuracy depends on the size, refractive index, and shape of the aerosols. Also, it may, in some cases, give reasonable results at one wavelength and for one specific parameter, and fail at other wavelength or for other optical parameters.

There is a large body of work concerned with aerosol optics and the shortcomings of simplified model particles. Some of these studies focus on specific types of aerosols, others on specific morphological properties, such as non-sphericity, inhomogeneity, surface roughness, or chemical heterogeneity. It is difficult to pick just a few of such studies as representative citations. So, we found that the best solution was to cite a recent review paper on aerosol optics modelling that discusses the strengths and shortcomings of various morphological models (Kahnert et al., 2014).

30. Paragraph beginning at line 307: It may be worth making it clear that y is not observed, but a model equivalent of the observations

We have inserted the following sentence: "The operator $\hat{H}$ maps from model space into observation space, which allows us to compare model output and observations."

31. Lines 324 and 326: I would suggest replacing all instances of "3-dimensional" with "three-dimensional"

Agreed.

32. Paragraph beginning 336: I would suggest mentioning that the assumption of unbiased background and observation errors

Agreed.

33. Footnote 6, page 15: See my comments above about the representativity component of the observation error.

OK; see our earlier response.

34. Footnote 7, page 16: I would suggest the following change: "The observation errors are often uncorrelated" becomes "The observation errors are often assumed to be uncorrelated (this is not always true)"

Agreed.

35. Paragraph beginning at line 368: Please comment on the role of spatial and inter- species correlations, particularly in light of the comment "if we allow all model variables to be freely adjusted" (line 374).

OK. We have added the following footnote (after "(within the given error bounds)."): By solving the equation $\nabla J|_{\boldsymbol{x}=\boldsymbol{x}_a} = \boldsymbol{0}$ for the analysed state $\boldsymbol{x}_a$ it can be shown that the solution to the inverse problem is given by $\boldsymbol{x}_a = \boldsymbol{x}_b + \mathbf{K} \cdot (\boldsymbol{y} - \hat{H}(\boldsymbol{x}_b))$, where $\mathbf{K} = \mathbf{B} \cdot \mathbf{H}^T \cdot (\mathbf{H} \cdot \mathbf{B} \cdot \mathbf{H}^T + \mathbf{R})^{-1}$ is known as the gain matrix. This illustrates that the analysis updates the background estimate $\boldsymbol{x}_b$ by mapping the increment $(\boldsymbol{y} - \hat{H}(\boldsymbol{x}_b))$ from observation space to model space by use of the gain matrix. The correlations among the model variables enter into the gain matrix through the matrix $\mathbf{B}$. In our case the vertical correlations are rather weak in comparison to correlations among different aerosol species.

36. Line 369: It might be worth noting that $\delta x$ is not constrained to ensure that all components of x remain positive in the analysis.

There is no such constraint in the minimisation process itself, but we do post-process the results for $\delta x$ such that negative concentrations would be set to zero. In practice, this rarely ever happens.

37. Line 386: The phrase "rather tricky" strikes me as somewhat colloquial. I would suggest the following change: "However, to actually make such a comparison israther tricky" becomes "However, to actually make such a comparison poses two problems."

Agreed.

38. Paragraph beginning at line 390: Please introduce the meaning of the angle- bracket notation. I believe that this is common in physics, but other disciplines (e.g. statistics) often use different notation for the expectation.

OK, we have added a formal definition of the expectation value for discrete variables in a footnote.

39. Footnote 8, page 17: Should $A \cdot A = B$ be $A^T \cdot A = B$?

Yes!

40. Line 425: Should "(C7)-(C9)" not be "(C6)-(C9)"? As far as I can see, Eq. (C6) is required here.

Yes.

41. Line 434-435: Please state which particular sections/chapters of Rodger (2000) the reader is referred to.

Agreed.

42. Equations C12, C15: I would suggest showing the range of the summation to indicate that it is a summation over observations (i.e. i ranges from 1 to m)

This is not generally true. The summation goes from 1 to $\min\{m,n\}$, where $n$ is the dimension of model space, and $m$ is the dimension of observation space. We have added these summation limits to the sums.

43. Line 479: "Naively, one may have expected that the dimension would, on the contrary, be reduced to $n-k$" – why? is this because the number of unknowns remains the same but the number of equations to be solved has increased by k?

In physics one usually learns about holomorphic constraints in theoretical mechanics, often by considering a point mass moving on a hypersurface. So, this is often the mental picture one invokes when dealing with constrained problems. For instance, a point mass in three-dimensional Euclidean space with a single holomorphic (i.e. strong) constraint can be pictured as moving on a two-dimensional surface. Thus this constraint reduces the dimension of the manifold on which the the point mass can move from three to two. One would therefore *naively* expect that one is now dealing with a two dimensional problem. The reason why this is naive is because a nonlinear constraint will correspond to a *curved* manifold. To characterise this manifold requires additional equations. Only if we have *linear* constraints, then the hypersurface is simply a tilted plane, which, by a suitable rotation-translation, can be brought into coincidence with, e.g., the xy plane. In such cases, and only in such cases, can the dimension of the problem actually be reduced, as one would naively have expected.

44. Line 486: I would suggest the following change: "(Note that the covariance matrices and their inverses are symmetric, i.e., $R^T = R$, etc.)" becomes "Note that the covariance matrices and their inverses are symmetric (i.e. $R^T = R$, etc.)."

Agreed.

45. Appendix: For all unit and zero matrices (and vectors), I would suggest indicating the dimension as a sub-script.

Agreed. We have changed this throughout the manuscript.

46. Line 498: I would suggest adding a subscript to clarify with respect to what the differentiation refers (i.e. replace $\nabla$ with $\nabla_\xi$).

Agreed.

47. Paragraph beginning line 515: how was this tuning done in practice?

As it is explained in the text. When the error variance is too large, one can see that the analysis is close to the un-constrained one. When it is too small, the analysis lies very close to the background estimate. One varies the variance until one obtains an analysis that departs from the background without drifting over to the (often noisy) unconstrained analysis.

 48. Line 549: "It turns out that Eq. (D18) gives a relatively sharp transition from unconstrained to constrained model vari-
ables, while Eq. (D19) gives a very gentle transition" – this can be seen from the equations. I would suggest replacing
the sentence with "It can be seen that Eq. (D18) gives a relatively sharp transition from unconstrained to constrained
model variables, while Eq. (D19) gives a very gentle transition"

We have replaced the text with "We tested all three approaches . These tests showed that the different approaches of-
ten yield analysis results that are quite similar. However, in each approach the free parameters $\sigma_G$ and $c$ are tuned to
different values. If they are not well tuned, then the analysis tends either toward the background estimate or toward the
unconstrained analysis, as explained earlier in the text following Eq. (D15)." Our tests, so far, showed that the differences
between these approaches are not quite as dramatic as we expected.

49. Paragraph beginning line 567: I found that this went too fast and skipped a bit too much detail, after what was otherwise
a very well-written paper that included a fair bit of theory. In particular, can you please explain in further detail the
reduced matrices. The phrase "we are primarily interested in constraining the chemical components" was surprising,
since I thought the authors were mainly interested in the aerosol components. What does it mean to "restrict ourselves
to the chemical subspace"?

This seems to be a misunderstanding. What we mean by "chemical components" is "chemical components in the aerosol
phase". Since our paper is exclusively concerned with aerosols, we thought that there was no risk of misunderstanding.
Thus, by "chemical subspace" we mean "subspace of aerosol components". We have revised the text accordingly and
replaced all instances of "chemical components" by "aerosol components", and similarly for "chemical subspace". Also,
we have revised the text in response to point 51 (see below).

50. Line 570: Full stop missing after $N_c$.

OK.

51. Paragraph beginning 574: similar to the above comment, I found that this skipped over too much detail. Please add fur-
ther explanation. The authors state that in their present study, they use a Cholesky decomposition of the B-matrix. Is this
what was used in Kahnert (2008), or is this described as the "spectral formulation"? If it is different, it may be relevant
to understand why the Cholesky decomposition was preferable to the author's previously presented methodology. This
is mainly to understand the requirements and limitations of the proposed methodology.

We *are* using the spectral formulation for the minimisation of the cost function. However, we formulate the weak con-
straints in a subspace of physical space, as explained above. The Cholesky decomposition is only applied to the reduced
B-matrix in the formulation of the weak constraints. We do not go into the details of spectral data assimilation, since
these questions are rather specific to our particular implementation, while the paper is not restricted to spectral methods.
However, we have rewritten this entire subsection and explained the reduced subspace approach in much more detail.
We have also added a short footnote on how to incorporate this into the spectral formulation.

**4.3 Minor formatting issues**

1. References with parentheses inside parentheses: lines 33, 268, 269, 376

This has been corrected.

2. Some of the in-line equations appeared to be missing spaces on one or both sides of the equals sign – this only appeared in the appendix. See lines: 391, 403, 404, 425, 515. I might just be imagining it. The paper was otherwise very well laid out.

Our latex program seems to insert spaces when using the eqnarray environment, but not when using the equation environment. We trust that the copy editor will take care of this problem.

**4.4 References the authors may wish to consider**

– Qin, X. Measuring information content from observations for data assimilation: relative entropy versus shannon entropy difference. Tellus: Series A. 59, 2, 198- 209, 2007.

– J. Joiner, A. M. da Silva. Efficient methods to assimilate remotely sensed data based on information content. Q. J. R. Meteorol, SOC. (1998), 124, pp. 1669- 1694

– C Cardinali, S Pezzulli, E Andersson. Influence-matrix diagnostic of a data as- similation system. Q. J. R. Meteorol. Soc. (2004), 130, pp. 2767-2786. doi: 10.1256/qj.03.205

– C. Johnson, N. K. Nichols; B. J. Hoskins. Very large inverse problems in atmo- sphere and ocean modelling. Int. J. Numer. Meth. Fluids 2005; 47:759-771.

– M Bocquet, 2009: Toward Optimal Choices of Control Space Representation for Geophysical Data Assimilation. Mon. Wea. Rev., 137, 2331-2348, doi: 10.1175/2009MWR2789.1.

– F Rabier, N Fourrie, D Chafai, P Prunet. Channel selection methods for Infrared Atmospheric Sounding Interferometer radiances. Q. J. R. Meteorol. Soc. (2002), 128, pp. 1011-1027

– C Johnson, B. J. Hoskins, N. K. Nichols. A singular vector perspective of 4D-Var: Filtering and interpolation. Q. J. R. Meteorol. Soc. (2005), 131, pp. 1-19 doi: 10.1256/qj.03.231

Agreed; these have been added to and discussed in the introduction.

**5 Reviewer 5**

Summary

The inversion of aerosol optical properties into the aerosol chemical composition is a ill posed problem. The authors use information theory techniques to estimate the amount of information contained in LIDAR observations. They present different methods to make use of it as contains in a 3DVAR algorithm. This is meant to avoid assimilating noise inherent to the observations. To evaluate their constrain methods, they create synthetic observations from CTM simulations and assimilate them back into the CTM.

Recommendation
The paper is well written and should be published. The methodology proposed is novel and can be applied to different obser-
vations within the variational assimilation framework.

We thank the reviewer for this positive evaluation of our paper.

Main comments
The authors choose to place all equations and their derivations into different appendixes. This hindered slightly the reading of
sections 2.4, 3.1 and 3.2. However, the overall readability of the manuscript is improved by the focus on the description and
evaluation of the method in the main text.

We agree. This point has been brought up by the other reviewers as well. We have followed the recommendations given by
reviewer 2 and included the key equations with explanations in the main text, while providing the more detailed derivations
in the appendix. This is a good compromise that keeps the paper accessible to non-theorists, while providing all the necessary
details in the appendix for the interested readers.

Minor comments
Figure 1 is hard to read, specially the colour bar. Otherwise, the previous Referees have a number of valid suggestions for
improvement, and I have nothing to add.

We have removed this figure in the revised manuscript. Since the paper is not concerned with those aspects specific to regional
modelling, this regional plot conveys no useful information in the context of this paper.

**6   Comment by P. Chazette**

May be you can check the papers of Wang at al. in ACP (2013, 2014a and b), where lidar assimilation is tested.

We thank Patrick Chazette for bringing these three papers, which he co-authored, to our attention. The results reported in
these articles are very interesting. The paper by Wang, Sartelet, Bocquet, and Chazette (2013) is particularly impressive. It
investigated assimilation of lidar and ground observations of PM10 and performed an observing system simulation experiment.
The results demonstrate that a relatively small lidar network can give analyses and forecasts of similar, and in some cases even
higher accuracy than corresponding results obtained with an extensive network of ground stations, such as AirBase. This clearly
demonstrates the potential of lidar observations. However, this study is only marginally relevant in the context of our paper,
because it considers assimilation of lidar measurements for determining PM10, not for determining the concentrations of each
aerosol component. It does not discuss the question of how to constrain the assimilation algorithm in order not to assimilate
noise. For this reason, we did not feel compelled to add a citation to this article.
The paper by Wang, Sartelet, Bocquet, and Chazette (2014) presents a comparison of modelled and measured backscattering
profiles, where the measurements were taken by a mobile lidar in the vicinity of Paris. The results of this comparison are highly
encouraging. They also describe their assimilation methodology. If we understand it correctly, they set up the assimilation to correct PM10, and they distribute the analysis increment back to the various aerosol components in each size class according
to the a priori distribution. In the context of our study, this is the most relevant fact in this paper, since it describes an ad
hoc method for specifying constraints. Essentially, this approach seems to be based on the same idea as that described in
Benedetti et al. (2009). However, we found that the explanations in the paper by Wang et al. (2014) were more detailed than in
the paper by Benedetti et al. (2009). For this reason, we have added a citation to this paper.

Finally, the paper by Wang, Sartelet, Bocquet, Chazette, et al. (2014) presents a very impressive and comprehensive evalu-
ation work of the potential of assimilating lidar measurements from the EARLINET network into an aerosol transport model.
Since it is an application rather than methodology paper, we did not cite it here; but we will be sure to cite it when we have
come that far and submit a paper on the operational evaluation of our lidar assimilation system.

**References**

[revised manuscript text omitted]